# Cost-Effective Online Multi-LLM Selection with Versatile Reward Models

## Abstract

With the rapid advancement of large language models (LLMs), the diversity of multi-LLM tasks and the variability in their pricing structures have become increasingly important, as costs can vary greatly between different LLMs. To tackle these challenges, we introduce the *C2MAB-V*, a Cost-effective Combinatorial Multi-armed Bandit with Versatile reward models for optimal LLM selection and usage. This online model differs from traditional static approaches or those reliant on a single LLM without cost consideration. With multiple LLMs deployed on a scheduling cloud and a local server dedicated to handling user queries, *C2MAB-V* facilitates the selection of multiple LLMs over a combinatorial search space, specifically tailored for various collaborative task types with different reward models. Based on our designed online feedback mechanism and confidence bound technique, *C2MAB-V* can effectively address the multi-LLM selection challenge by managing the exploration-exploitation trade-off across different models, while also balancing cost and reward for diverse tasks. The NP-hard integer linear programming problem for selecting multiple LLMs with trade-off dilemmas is addressed by: i) decomposing the integer problem into a relaxed form by the local server, ii) utilizing a discretization rounding scheme that provides optimal LLM combinations by the scheduling cloud, and iii) continual online updates based on feedback. Theoretically, we prove that *C2MAB-V* offers strict guarantees over versatile reward models, matching state-of-the-art results for regret and violations in some degenerate cases. Empirically, we show that *C2MAB-V* effectively balances performance and cost-efficiency with nine LLMs for three application scenarios.

## 1 Introduction

In the digital era of today, large language models (LLMs) such as ChatGPT lead innovations in computational linguistics and cognitive processing Tian et al. (2023). The emergence of numerous high-performance LLMs has sparked significant interest in the challenge of model selection Foster et al. (2019); Wang et al. (2023). Typically, current schemes for selecting LLMs often rely on identifying the best-performing model in a static setting, such as selecting the LLM that achieves the lowest perplexity score Salazar et al. (2019); Peng et al. (2023). However, the diverse capabilities of various LLMs present an opportunity to adopt a task-specific selection approach, where different LLMs have their own strengths and weaknesses, e.g., Investlm Yang et al. (2023) is specifically designed for the financial sector and may better suit queries related to investments. Moreover, the limitations of static selection methods become more pronounced due to factors like "*generation diversity*", where a less expensive LLM may perform better in certain scenarios Chen et al. (2023), and "*data drift*", which refers to changes in the characteristics of answers generated in real-time compared to those of training data Bhardwaj et al. (2022). Consequently, beyond large-scale pre-training methods, these issues highlight the need for an "*online*" approach to optimize decision-making in selecting the "*appropriate*" LLMs, which leverage *continuous feedback* to adapt to the varying performance levels of models and diverse application needs through *ongoing user interaction*.

Furthermore, scenarios combining multiple LLMs (or agents) to complete tasks have commonly emerged, moving beyond using a single LLM. Platforms like Poe (2024) have spearheaded functionalities that integrate several bots within a single chat session. Liu et al. (2023c) introduces a dynamic LLM-agent network through dynamic interaction architecture and intelligent agent team optimization. Hong et al. (2023) proposes a meta-programming framework that enhances the col-

laboration of multiple LLMs. Gupta et al. (2024) explores the implementation of LLM cascades for generative tasks. However, previous works have not considered optimizations tailored to the characteristics of different tasks, which may feature different forms of rewards. Below, we present three streamlined collaborative combinations of LLMs for different tasks: 1. For user experience enhancement, multiple LLMs may be deployed to ensure satisfactory outcomes. 2. In educational tutoring, subject-specific LLMs operate in parallel, with failures in one not severely impacting others. 3. In project development, LLMs manage different sub-modules, where any module's failure could jeopardize the entire project. These three examples underscore the importance of combining multiple suitable LLMs based on the specific "*task structure*".

Additionally, it is crucial to recognize that the emergence of LLMs with diverse performance levels introduces varying costs in practical use, an important factor neglected in existing research. The operating expenses for LLMs are high; for example, it is estimated that running ChatGPT costs over $700,000 daily, and deploying GPT-4 for customer service could cost a small business upwards of $21,000 per month Chen et al. (2023). This implies the necessity of incorporating *cost considerations* into the selection and utilization strategies for LLMs.

Based on the discussions above, we introduce the Cost-effective Combinatorial Multi-armed Bandit with Versatile reward models (*C2MAB-V*), designed to synergize the integration of diverse LLMs across different task types. *C2MAB-V* manages the dual challenges of selecting LLMs that both achieve high performance and meet cost constraints. Additionally, *C2MAB-V* adapts to various multi-LLM collaborative tasks by utilizing a "*combinatorial model selection strategy*" which extends beyond traditional single-model limitations by encompassing a broad spectrum of LLM candidates. Finally, to address the NP-hard complexities of combinatorial LLM selection under cost considerations, we transform the initial problem into a continuous form. This process is executed on the local server, which is required to handle user queries. Meanwhile, our discretization method, which guarantees precise evaluation of LLM performance post-relaxation, is implemented on the scheduling cloud where multiple LLMs are deployed. This two-tiered approach can accommodate both the limited resources of the local server and strategically mitigate the scheduling cloud's workload. We emphasize that this paper focuses on a fully online learning approach that operates without initial information and relies solely on user feedback with robust interpretation. Our method is designed to complement, not compete with, current approaches that utilize GPU resources for offline training, where offline training can streamline the search space and enhance the convergence speed of online learning.

In summary, our contributions are as follows.

**Novel Multi-LLM Selection Formulation.** We introduce a novel formulation of the *cost-effective multi-LLM selection*, designed for tasks requiring collaboration among multiple LLMs with *versatile reward structures*. This formulation emphasizes the essential balance between exploring new models and exploiting proven effective models, while adhering to long-term cost budget considerations and securing high rewards across diverse multi-LLM tasks.

**Unique Online Algorithmic Framework.** We developed the *C2MAB-V* framework for online multi-LLM selection, managing diverse collaborative LLM tasks and performance variability due to *generation diversity* and *data drift*. Our approach utilizes a "*combinatorial bandit selection strategy*" with a cost-conscious versatile reward structure. Based on a natural local-cloud architecture, the local server, with its limited resources, simplifies the selection process and alleviates the computational load on the scheduling cloud. The cloud coordinates and selects LLMs based on continualized feedback data from the local server, which also gathers user feedback to enhance LLM evaluations.

**In-Depth Regret and Violation Analysis.** We conduct a thorough theoretical analysis of our online *C2MAB-V* framework, covering three distinct reward models. This analysis addresses the trade-offs between reward and violation, and between exploration and exploitation. We identify common properties across different reward models and employ "*martingale construction techniques*" to examine the stochastic properties of our model under varying collaborative tasks. Notably, our results on regret and violation analysis match the state-of-the-art results in several degenerate cases.

**Comprehensive Empirical Validation.** Our *C2MAB-V* framework has been empirically validated with superior performance outcomes across evaluations involving nine distinct LLMs. These tests consistently confirm *C2MAB-V*'s capacity to adaptively navigate the trade-off dilemmas, resulting in amplified rewards or decreased costs. Moreover, detailed analysis from exploratory experiments provides deeper insights into the strategic design and utilization of the multi-LLM approach.

## 2 RELATED WORK AND MOTIVATION

### 2.1 RELATED WORK

**Combinatorial Multi-Armed Bandit.** The field of online learning problems under the multi-armed bandit (MAB) model has been extensively studied. The MAB model was first introduced by the seminal work Robbins (1952) and has been expanded upon by many other researchers ( Liu et al. (2021c); Slivkins et al. (2019); Liu et al. (2023a)). Traditional MAB models focus on selecting a single arm per trial; however, the more complex scenario of combinatorial MAB (CMAB) involves selecting multiple arms simultaneously. The stochastic CMAB has received much attention recently Chen et al. (2016); Kveton et al. (2015b); Wang & Chen (2017); Merlis & Mannor (2019; 2020). Initial research on stochastic CMAB is spearheaded by Gai et al. (2012), with subsequent improvements in regret bounds offered by Kveton et al. (2015c); Combes et al. (2015). Later on, Chen et al. (2016); Wang & Chen (2017) considers probabilistic feedback to extend the feedback model. Recently, Liu et al. (2022) proposes the variance-adaptive algorithm BCUCB-T. Liu et al. (2023a) incorporates the contextual information in CUCB. Our research builds on the CMAB settings but diverges significantly by addressing more complicated trade-off issues. Furthermore, to address various cooperative task types, we explore three different combinatorial reward formulations within versatile reward models. Please refer to the additional literature in Appendix A.2, and the unique challenges in Appendix A.3.

**Multi-LLM Combination.** The combination of multiple LLM models for enhanced performance has received considerable attention, aimed at bolstering output quality Kim et al. (2023). Techniques such as "*knowledge distillation*" Hinton et al. (2015); Sanh et al. (2019) facilitate the training of compact models to mimic the behavior of larger and more complex models, thus optimizing resource utilization. Meanwhile, "*ensemble learning*" Hokamp et al. (2020); Huy et al. (2022) leverages the collective predictions of independently trained models for superior results. Nonetheless, the prevalent practice of withholding model internals from commercial LLM, such as OpenAI (2023), restricts knowledge distillation by obscuring the "teacher" model, which can decrease the replication efficacy of the "student" model. Concurrently, ensemble learning faces increased financial and procedural burdens due to the necessity of amalgamating various model outputs, a task complicated by the absence of open-source models. Madaan et al. (2023); Ding et al. (2024); Chen et al. (2023) primarily address one of the scenarios we have outlined, i.e., selecting a single LLM that best satisfies the user from multiple options. Moreover, these approaches do not incorporate user feedback. Our approach distinctively accounts for the online aspects of model selection, the diversity of multi-LLM task types, and the associated costs, setting it apart from previous methods.

### 2.2 MOTIVATION

We introduce the rationale behind our cost-effective combinatorial LLM online selection strategy.

**Limitations of Single and Static Model Policy.** Table 1 presents the commercial costs of various LLMs Chen et al. (2023). Economically, it may not be viable for businesses or government entities to consistently choose the most expensive option, such as GPT-4, for all applications. Therefore, a careful assessment of the trade-offs between cost and performance (termed as 'reward') is essential, as it allows for more strategic budget management across various tasks.

Table 1: Cost comparison of various LLMs based on 10 million output tokens from Chen et al. (2023).

| LLM | ChatGPT | GPT-3 | GPT-4 | ForeFrontAI | J1-Large | J1-Jumbo |
|---|---|---|---|---|---|---|
| **Cost** ($) | 2 | 20 | 60 | 5.8 | 30 | 250 |
| **LLM** | **Xlarge** | **GPT-Neox** | **GPT-J** | **GPT-Curie** | **FAIRSEQ** | **J1-Grande** |
| **Cost** ($) | 10 | 35 | 5 | 2 | 15 | 80 |

Moreover, we evaluate the performance of various LLM, including GPT-4, GPT-3.5, Claude 1.2, Claude 2, Forefront OpenAI (2023); Forefront-AI (2023); Claude (2023), using the mathematics dataset Saxton et al. (2019) and SciQ dataset Welbl et al. (2017), which cover multiple topics including physics, chemistry, and biology. As shown in Fig. 1, comparing these LLMs across three randomly selected scenarios with problem sets (each containing 200 samples) reveals the inherent limitations of relying on a single LLM. This measurement highlights the limitations of advanced models like GPT-4 in various contexts, emphasizing the phenomena of "generation diversity" mentioned in Section 1

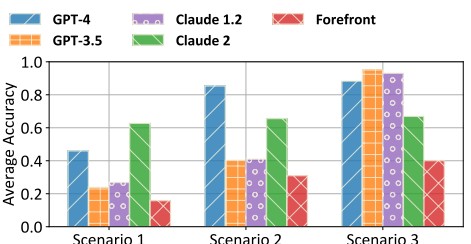

Figure 1: Accuracy of different LLMs across varied problem samples.

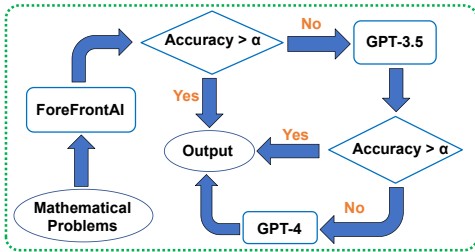

Figure 2: Simple example of combinatorial LLMs in a cascading form.

and the necessity of continual online learning to select the "appropriate" LLMs for different queries. Moreover, challenges like "distribution shift" Prudencio et al. (2023) and the "Matthew effect" Gao et al. (2023) can cause deviations in the performance of models presumed optimal in the offline settings. Subsequently, the adoption of an online learning framework becomes essential, with ongoing interaction utilized to refine LLM selection.

**Benefits of Combinatorial LLMs.** We select the multi-LLM collaboration task on *ensuring user experience while minimizing costs* as an example. As shown in Fig. 2, ForeFrontAI is the first option. If its response's accuracy matches the right choice question label from the datasets Welbl et al. (2017), the query is routed to GPT-3.5. For further refinement, GPT-4 is invoked. This combination of multi-LLMs in a cascading form is compared with the exclusive use of GPT-4 for identical queries. Cost evaluation, based on the statistical data of Chen et al. (2023), reveals that such combinatorial LLMs incurs only 60.1% of the expenses associated with relying solely on GPT-4. Accuracy assessments further highlight the merits of the combinatorial approach, achieving an accuracy of 0.824 on the dataset, surpassing the 0.732 accuracy obtained by using GPT-4 alone. Consequently, the strategy of using combinatorial LLMs demonstrates a promising and compelling alternative.

## 3  PROBLEM FORMULATION

In this section, we introduce our online framework of a cost-effective combinatorial bandit for multi-LLM selection with versatile reward models, namely, *C2MAB-V* (Please refer to Fig. 3 for details). A summary of the main symbols is provided in Appendix A.1.

**Local-Cloud Architecture.** Given the large number of parameters and significant storage overhead of LLMs, a typical approach, if not opting for a streamlined but less capable version of LLMs, involves deploying multiple LLMs on a resource-abundant scheduling cloud. When responding to user queries, the local server first handles the requests and synchronizes communication with the cloud to initiate tasks (for discussions on asynchronous handling, see Appendix E.3). Subsequently, user feedback is stored on the local server.[1] While the scheduling cloud serves multiple local servers, for ease of presentation, we focus on describing the relationship between a single local server and the cloud.

**Combinatorial LLM Instance.** The scheduling cloud orchestrates multiple independent LLMs, to effectively fulfill requests from the local server. Let $\mathcal{K} = \{1, \ldots, K\}$ represent the set of all LLMs, where each index $k \in \mathcal{K}$ corresponds to a specific LLM (i.e., base arm), comprising a total of $K$ LLMs. The system operates in a time-slotted manner, delineated by discrete intervals $\mathcal{T} = \{1, 2, 3, \ldots, T\}$. During each round $t \in \mathcal{T}$, the cloud coordinates and selects a subset $S_t$ of the available LLMs from $\mathcal{K}$. This selection process, termed an "action", adapts dynamically based on real-time constraints and availability, with the cardinality $|S_t| \leq K$ indicating the number of LLMs selected. Let $\mathcal{S}$ represent the set of all possible combinations of actions.[2] For example, high-demand workloads may cause GPT-4 to reach its usage limit, temporarily preventing its selection. Let $N = \max_{S \in \mathcal{S}} |S|$, denoting the maximum number of LLMs that can be simultaneously active.

**Online Learning Protocol.** A combinatorial LLM instance involves the sequential interaction between the local server, the scheduling cloud, and user queries within our local-cloud architecture.

---

[1]Feedback can include both direct user input and data from techniques that quantify user behavior or responses, as well as preference simulators Dwaracherla et al. (2024). Regardless of the methods, we refer to all these collectively as "feedback" for simplicity.

[2]Action $S_t$ is actually a set. Similarly to previous CMAB works, we also do not use script font here.

The local server processes user activity and feedback, particularly focusing on the users' performance feedback regarding the utilized LLMs. This process involves locally recording and updating the performance evaluations of the LLM. Then, the local server will transmit these information to the scheduling cloud. The scheduling cloud undertakes the new coordination and selection of an action $S_t$ (i.e., selecting a subset of LLMs for service) after receiving the newly updated information from the local server. Such performance evaluations, termed as rewards, could, for instance, be based on the ROUGE-2 score for automatic summarization tasks Lin (2004), and are represented by a random variable vector $\boldsymbol{X}_t = (X_{t,1}, \ldots, X_{t,|S_t|})$. For simplification, we posit that $\boldsymbol{X}_t \in [0, 1]^{|S_t|}$. Furthermore, for each LLM $k \in \mathcal{K}$ included in the action $S_t$, its associated cost $y_{t,k}$ is observable at round $t$. A detailed definition of this cost will be provided later.

Note that we demonstrate a fully online learning process here without any prior information for a clear description. In real-world scenarios, online learning can often be accelerated by incorporating offline pre-training. For example, for mathematical queries, narrowing the action space by offering fine-tuned LLMs for specific mathematical tasks, rather than selecting from a broad set, can significantly streamline the online selection process.

**Versatile Reward Models.** Let $\boldsymbol{\mu} = (\mu_1, ..., \mu_K)$ represent the "*initially unknown*" mean vector of outcomes for each LLM with $\mu_k = \mathbb{E}[X_{t,k}]$. We consider versatile reward models for different combinations of LLMs in multi-LLM tasks below.
- **Any Win Combination (AWC):** $r(S; \boldsymbol{\mu}) = \left(1 - \prod_{k \in S}(1 - \mu_k)\right)$. As illustrated in Fig. 2, this reward model is designed to safeguard user experience by selecting multiple LLMs to generate answers, with success defined as any LLM's answer satisfying the user. This reward model aims to maximize user satisfaction by providing a range of possible solutions.
- **Sum Up Combination (SUC):** $r(S; \boldsymbol{\mu}) = \sum_{k \in S} \mu_k$. In this setup, domain-specific LLMs independently tackle tasks in parallel. Each LLM earns rewards for correctly answering questions in its field. This reward model aims to speed up task completion and reduces the workload on each LLM, enhancing overall task effectiveness.
- **All In Combination (AIC):** $r(S; \boldsymbol{\mu}) = \prod_{k \in S} \mu_k$. This reward model is exemplified in development tasks, where each LLM is responsible for developing sub-modules simultaneously. The key aspect of this reward model is that the failure of any LLM leads to the failure of the entire development task, thus ensuring the success of the whole collaborative work.

For more discussions (e.g., specify LLMs under the SUC and AIC reward models), please refer to Appendix C.1. Compared to previous works Ding et al. (2024); Chen et al. (2023); Madaan et al. (2023); Zhu et al. (2023), which primarily focus on selecting a single LLM to satisfy the user, our versatile reward model accounts for the diversity of tasks, addressing scenarios where multi-LLMs collaborate to complete tasks, besides the distinction being our focus on an online learning setting.

**Partial LLM Feedback.** In the process of selecting and querying LLM in action $S_t$, not all LLMs may actually be queried in the muti-LLM tasks. Consequently, the local server often only observes outcomes from a subset of the selected LLMs. Formally, the local server observes feedback from the LLM subset $\mathcal{F}_t$ in action $S_t$, where the subset's cardinality is denoted as $F_t$. If all LLMs in $S_t$ are queried, then $F_t = |S_t|$, which is worst-case scenario under the AWC reward model.

**Statistically-Based Cost Model.** Drawing inspiration from Alibabacloud-Opensearch (2024), which charges based on statistical computational resource utilization (with 1 CU supporting an average of 10 interactions), Awan-LLM (2024), which offers unlimited token pricing without the consideration of token limits, and JD-Cloud-Yanxi-AI (2024), which bills based on exclusive resource groups, we propose a statistically-based cost model. Specifically, for any query $q$ from a set $\mathcal{Q}$, LLM $k \in \mathcal{K}$ processes the query using a deterministic number of input tokens, $l_k^{\text{in}}(q)$, and generates a random number of output tokens, denoted as $l_k^{\text{out}}(q) \sim D_k^{\text{out}}(q)$, representing the probabilistic nature of LLMs Xie et al. (2021); Dalal & Misra (2024). Given a distribution $D_q$ over $\mathcal{Q}$, a user selects a query $q_t \sim D_q$ in each round, and the cost for LLM $k$ is $y_{t,k} = (l_k^{\text{in}}(q_t) + l_k^{\text{out}}(q_t))C_k$, where $C_k$ is the cost per token. The goal is to estimate the *unknown* expected cost $c_k = \mathbb{E}[y_{t,k}]$.

**Budget Violation Consideration.** The total cost of executing action $S_t$ on the utilized LLM subset $\mathcal{F}_t$, selected by the scheduling cloud at round $t$, is given by $\sum_{k \in \mathcal{F}_t} y_{t,k}$. Furthermore, there is a predetermined budget guarantee threshold $\rho > 0$ exists for the combinatorial set of LLM, requiring the cumulative cost of the selected action to remain below this threshold $\rho$ for the long term. This mechanism allows organizations, including enterprises and governmental bodies, to manage LLM usage efficiently and align with budgetary constraints. To assess compliance with this

budgetary constraint, the concept of constraint violation Cai et al. (2022) is introduced as follows, with $[x]^+ := \max(x, 0)$:

$$V(T) = \left[ \frac{1}{T} \sum_{t=1}^{T} \sum_{k \in \mathcal{F}_t} y_{t,k} - \rho \right]^+, \tag{1}$$

Although it is conceivable to occasionally exceed this cost constraint during the online learning process, significant overruns are not permissible, thus necessitating the violation metric. A violation rate decreasing as $\tilde{O}\left(T^{-\gamma}\right)$ signifies effective constraint adherence, indicating that $V(T)$ diminishes at this rate, where $\gamma > 0$ illustrates the rate of reduction in violations with $T$ Chen et al. (2018).

$\alpha$-**Approximate Regret.** The performance of an online learning algorithm $A$ is evaluated by its *"regret"*, which is the discrepancy between the expected cumulative reward of consistently choosing the optimal action $S_t^* \triangleq \arg\max_{S \in \mathcal{S}} r(S; \boldsymbol{\mu})$ at each round $t$, with $\mathcal{S}$ representing the set of all viable actions, and the expected cumulative reward resulting from the actions selected by algorithm $A$. The challenge, however, lies in the computational difficulty of determining the exact $S_t^*$, even when $\boldsymbol{\mu}$ is known, as this can be NP-hard Hochba (1997). Thus, following Li et al. (2016); Wang & Chen (2017); Liu et al. (2022; 2023a), we presume that algorithm $A$ utilizes an offline $\alpha$-approximation oracle. This oracle, upon receiving a mean vector $\boldsymbol{\mu}$, outputs an action $S$ that guarantees $r(S; \boldsymbol{\mu}) \geq \alpha \cdot r(S_t^*; \boldsymbol{\mu})$. With the $\alpha$-approximation oracle, the $\alpha$-approximate regret over $T$ rounds is defined as:

$$R(T) = \mathbb{E}\left[ \sum_{t=1}^{T} \left( \alpha \cdot r(S_t^*; \boldsymbol{\mu}) - r(S_t; \boldsymbol{\mu}) \right) \right], \tag{2}$$

where the expectation accounts for the randomness in outcomes $\boldsymbol{X}_1, ..., \boldsymbol{X}_T$, and algorithm $A$ itself.

**Goal of Multi-LLM Selection.** The performance of an online algorithm $A$ is critically evaluated through both violation and regret metrics, as defined in Eq. (1) and Eq. (2), respectively. A diminishing regret implies an increasingly oracle-like performance by algorithm $A$, indicative of its efficiency in selecting optimal LLM combinations. Conversely, a reduction in violation highlights improved compliance with predefined cost constraints over time. The dual objectives for algorithm $A$ involve simultaneously minimizing regret and violation under different reward models.[3]

# 4 ALGORITHM DESIGN

We present the design of the online *C2MAB-V* framework, as depicted in Algorithm 1. Our method addresses the challenge of estimating unknown rewards and costs associated with using corresponding LLMs. As shown in Fig. 3, given the NP-hardness complexity of selecting a combination of LLMs under cost constraints, we transform the original integer problem into a continuous space. Utilizing the limited resources of a local server, we solve this relaxed optimization problem, which can also ease the computational load on the scheduling cloud

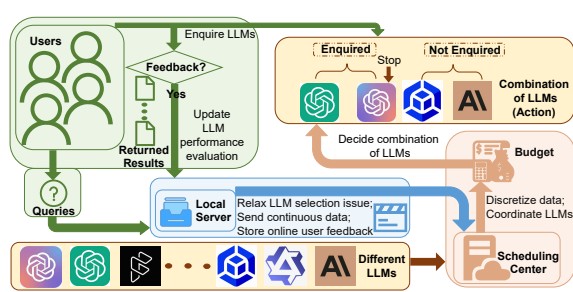

Figure 3: Design of *C2MAB-V* workflow, with detailed process descriptions provided on the left main text.

when supporting multiple local-server query requests. A scheduling cloud with multiple LLMs coordinates and selects based on continuous data transmitted from the local server. Meanwhile, the local server collects user feedback to enhance the online evaluations of LLMs. Additionally, the cloud does not have access to the original sensitive user data, enhancing future privacy protection prospects. Next, we describe the processes undertaken by the local server and the scheduling cloud.

## 4.1 PROCEDURES BY LOCAL SERVER

**Confidence Bound for Reward and Cost.** To avoid the limitations of a greedy LLM selection strategy, i.e., $\arg\max_{k \in \mathcal{K}} \hat{\mu}_{t,k}$ with $\hat{\mu}_{t,k}$ denoting the empirical estimate of the true mean of LLM $k$, which might result in the risk of overlooking superior LLM options, we implement the confidence bound (CB) method Lattimore & Szepesvári (2020); Liu et al. (2022). This "optimistic" strategy

---

[3]Without loss of generality, we assume $\mu_k, c_k \in [0, 1]$ for all $k \in \mathcal{K}$.

---

**Algorithm 1** Online Update of *C2MAB-V* with Feedback

---

**Input:** Set of all LLMs $\mathcal{K}$, cost constraint $\rho$, probability parameter $\delta \in (0, 1]$.

 1: **Initialize:** $\forall k \in \mathcal{K}$, $\hat{\mu}_{1,k} = 0$, $\hat{c}_{1,k} = 0$, $T_{1,\mu_k} = 0$, $T_{1,c_k} = 0$.
 2: **for** $t = 1, 2, \cdots, T$ **do**
 3:     Predict reward of $k$-th LLM $\bar{\mu}_{t,k} = \min\{\hat{\mu}_{t,k} + \alpha_\mu \rho_{t,\mu_k}, 1\}$ based on "optimistic" strategy;
 4:     Estimate cost of $k$-th LLM $\underline{c}_{t,k} = \max\{\hat{c}_{t,k} - \alpha_c \rho_{t,c_k}, 0\}$ based on "pessimistic" strategy;
 5:     Utilize the greedy algorithm to solve the relaxed constrained optimization problem in Eq. (3);
 6:     Obtain LLM action $S_t$ by discretization rounding (Algorithm 2 or Algorithm 3);
 7:     Observe the corresponding reward of LLMs and the cost of LLMs in for $k \in \mathcal{F}_t$ ;
 8:     Update $\hat{\mu}_{t,k}$ , $\hat{c}_{t,k}$ $\forall k \in \mathcal{K}$ according to Eq. (6).
 9: **end for**

---

promotes exploration among various LLM alternatives, thereby reducing the likelihood of consistently choosing sub-optimal options without considering potentially better LLMs. We define the confidence radius of CB as $\rho_{t,\mu_k} = \sqrt{\ln\left(\frac{2\pi^2 K t^3}{3\delta}\right)/2T_{t,\mu_k}}$, which quantifies the exploration potential for the reward of LLM $k$ at round $t$, with $\delta$ in the range $(0, 1]$. The term $T_{t,\mu_k}$ represents the number of times LLM $k$ has been selected in action $S_t$. Accordingly, the adjusted reward prediction for LLM $k$ is defined as $\bar{\mu}_{t,k} = \min\{\hat{\mu}_{t,k} + \alpha_\mu \rho_{t,\mu_k}, 1\}$, with $\alpha_\mu$ denoted as a positive control parameter. Furthermore, we take into account the cost associated with each LLM. In light of the uncertainty introduced by complex queries, we adopt a cautious approach based on "pessimistic" strategy to ensure adherence to the cost budget. Specifically, the adjusted cost estimate for LLM $k$ at round $t$, $\underline{c}_{t,k}$, is determined by $\underline{c}_{t,k} = \max\{\hat{c}_{t,k} - \alpha_c \rho_{t,c_k}, 0\}$. Here, $\hat{c}_{t,k}$ is the empirical cost, $\alpha_c$ is a positive adjustment parameter, and confidence radiu $\rho_{t,c_k} = \sqrt{\ln\left(\frac{2\pi^2 K t^3}{3\delta}\right)/2T_{t,c_k}}$, with $T_{t,c_k}$ denoting the count of LLM $k$ selection up to round $t$. Using this CB approach, we have the following lemma.

**Lemma 1.** *For each round $t$ and LLM $k \in \mathcal{K}$, define $\mathcal{N}_\mu$ as the event where $|\hat{\mu}_{t,k} - \mu_k| < \rho_{t,\mu_k}$, and $\mathcal{N}_c$ as $|\hat{c}_{t,k} - c_k| < \rho_{t,c_k}$. Then, the probability of $\mathcal{N}_\mu, \mathcal{N}_c$ occurring is at least $1 - \delta/2$, i.e., $\Pr\{\mathcal{N}_\mu\} \geq 1 - \delta/2$, $\Pr\{\mathcal{N}_c\} \geq 1 - \delta/2$.*

Please refer to Appendix D.1 for the proof. Lemma 1 underscores the high-probability events wherein the empirical estimates for both reward and cost closely align with their true means of LLMs.

**Relaxation Strategy for LLM Combination Selection.** To mitigate the computational hardness of the LLM selection problem, the local server adopts a relaxed strategy while also ensuring that original sensitive user information is not transmitted to the scheduling cloud. Specifically, let indicator variable $\mathbb{I}_S = \{z_1, z_2, \cdots, z_K\} \in \{0, 1\}^K$ denote the selection status of LLM in $\mathcal{K}$, where $z_k = 1$ indicates LLM $k$ is selected, and $z_k = 0$ otherwise. $z_k \in \mathbb{I}_S$ is regarded as a continuous variable $\tilde{z}_k \in [0, 1]$, with $\tilde{\boldsymbol{Z}} = \{\tilde{z}_1, \tilde{z}_2, \cdots, \tilde{z}_K\}$. We then introduce the three different relaxation strategies.

• **Any Win Combination (AWC):** Treating $r(S; \boldsymbol{\mu}) = \left(1 - \prod_{k \in S}(1 - \mu_k)\right)$ as a submodular function, we apply its multi-linear extension to accommodate the relaxed problem form: $\tilde{r}\left(\tilde{\boldsymbol{Z}}, \bar{\boldsymbol{\mu}}\right) = \sum_{S \subseteq \mathcal{K}} \prod_{k \in S} \tilde{z}_k \prod_{k \notin S}(1 - \tilde{z}_k)$, ensuring convexity across any direction $\mathbb{I}_{\{i\}} - \mathbb{I}_{\{j\}}$ for distinct $i, j \in \mathcal{K}$ Calinescu et al. (2007). The closed form, $\tilde{r}\left(\tilde{\boldsymbol{Z}}, \bar{\boldsymbol{\mu}}\right) = \left(1 - \prod_{k \in \mathcal{K}}(1 - \bar{\mu}_k \tilde{z}_k)\right)$, of this extension leads to the following relaxed optimization problem:

$$\left\{\max\left(1 - \prod_{k \in \mathcal{K}}(1 - \bar{\mu}_k \tilde{z}_k)\right) : \sum_{k \in \mathcal{K}} \tilde{z}_k \leq N, \sum_{k \in \mathcal{K}} \underline{c}_{t,k} \tilde{z}_k \leq \rho, 0 \leq \tilde{z}_k \leq 1, \forall k \in \mathcal{K}\right\}. \quad (3)$$

The common greedy algorithms, apt for such constrained continuous problems, can efficiently select $\tilde{z}_k$ values that optimize $1 - \bar{\mu}_k \tilde{z}_k$ within constraints Boyd & Vandenberghe (2004). Subsequently, this optimization problem can be easily and effectively solved.

• **Sum Up Combination (SUC):** For $r(S; \boldsymbol{\mu}) = \sum_{k \in S} \mu_k$, we choose the relaxed reward function to be $\tilde{r}(\tilde{\boldsymbol{Z}}, \bar{\boldsymbol{\mu}}) = \sum_{k \in \mathcal{K}} \bar{\mu}_k \tilde{z}_k$ and the relaxed constraint optimization problem is:

$$\left\{\max \sum_{k \in \mathcal{K}} \bar{\mu}_k \tilde{z}_k : \sum_{k \in \mathcal{K}} \tilde{z}_k = N, \sum_{k \in \mathcal{K}} \underline{c}_{t,k} \tilde{z}_k \leq \rho, 0 \leq \tilde{z}_k \leq 1, \forall k \in \mathcal{K}\right\}. \quad (4)$$

For such relaxed linear programming, it can be easily solved in polynomial time Chan (2018).

- **All In Combination (AIC):** Since the reward function $r(S; \boldsymbol{\mu}) = \prod_{k \in S} \mu_k$ is the conjunctive reward function and the feasible actions are $\mathcal{S}_2 = \{S \subseteq \mathcal{K} : |S| = N\}$, we select the relaxed function to be $\tilde{r}(\tilde{\boldsymbol{Z}}, \bar{\boldsymbol{\mu}}) = \prod_{k \in \mathcal{K}} \bar{\mu}_k^{\tilde{z}_k}$, with the following optimization problem:

$$\max\{\prod_{k \in \mathcal{K}} \bar{\mu}_k^{\tilde{z}_k} : \sum_{k \in \mathcal{K}} \tilde{z}_k = N, \sum_{k \in \mathcal{K}} \underline{c}_{t,k}\tilde{z}_k \leq \rho, \ 0 \leq \tilde{z}_k \leq 1, \forall k \in \mathcal{K}\}. \quad (5)$$

The optimal solution in Eq. (5) is equivalent to solving the logarithmic linear programming $\arg\max\{\sum_{k \in \mathcal{K}} \tilde{z}_k \ln \bar{\mu}_k : \sum_{k \in \mathcal{K}} \tilde{z}_k = N, \sum_{k \in \mathcal{K}} \underline{c}_{t,k}\tilde{z}_k \leq \rho, \ 0 \leq \tilde{z}_k \leq 1\}$ by taking $\ln r(S; \bar{\boldsymbol{\mu}})$ as our objective, thus making the optimization problem more tractable.

The commonalities in the above three types of reward forms are in Appendix C.2. We will further describe the selection of multiple LLMs based on the continuous variable $\tilde{\boldsymbol{Z}}_t = \{\tilde{z}_{t,1}, \tilde{z}_{t,2}, \cdots, \tilde{z}_{t,K}\} \in [0, 1]^K$ at round $t$ in Section 4.2 on *Procedures by Scheduling Cloud*.

**Online Update for Combinatorial LLMs.** In contrast to traditional offline approaches that use relaxation methods to address constrained optimization problems Gandhi et al. (2006); Chekuri et al. (2009), our strategy enables the local server to dynamically adapt LLM performance estimations for both reward and cost, leveraging continual feedback from the combinatorial LLM selection process.

Specifically, the partial combinatorial feedback model is employed to enhance reward prediction and cost estimation based on the LLM in the chosen actions, which are updated as follows:

$$\hat{\mu}_{t+1,k} = \frac{T_{t,\mu_k}\hat{\mu}_{t,k} + X_{t,k}}{T_{t+1,\mu_k}}, \quad \hat{c}_{t+1,k} = \frac{T_{t,c_k}\hat{c}_{t,k} + y_{t,k}}{T_{t+1,c_k}}, \quad k \in \mathcal{F}_t. \quad (6)$$

Note that for an action $S_t$, the local server only monitors the performance of the LLMs that are actually used in $S_t$, as not all LLMs are utilized for every type of task. For example, in the case of an AWC task type, if one LLM provides a satisfactory answer, the remaining LLMs are not utilized. In the AWC scenario, we use a conservative approach by ensuring that the budget threshold is met even if all selected LLMs $S_t$ are used, representing a "cautious" economic strategy.

### 4.2 Procedures by Scheduling Cloud

**Discretization Rounding for LLM Selection.** In our architecture, the scheduling cloud can communicate with each local server, while the local servers do not directly communicate with each other. As described in Section 3, our focus is on elucidating the one-to-one relationship between a local server and the cloud. With fully original user data stored locally, the local server sends the relaxed continuous data $\tilde{\boldsymbol{Z}}_t$ to the scheduling cloud, which then coordinates various LLMs and selects a new action in response to requests. Inspired by the works of Chekuri et al. (2009); Gandhi et al. (2006), the synchronized cloud employs specialized discretization rounding algorithms based on reward models to convert the relaxed continuous data back into discrete form. By discretizing $\tilde{\boldsymbol{Z}}_t$, the scheduling cloud identifies a feasible set of LLMs $S_t$ to schedule and select for the current round $t$ (line 6 in Algorithm 1). *The corresponding algorithms are tailored solutions for our versatile reward model and represent a significant effort to find suitable methods for our specific setup.* However, since the design of these algorithms is not our core contribution, details are provided in Appendix B. Note that discretization rounding can reduce the complexity of LLM combination selection, as detailed in the computational efficiency comparison with direct discrete constrained optimization in Appendix E.3.

## 5 Performance Analysis

We conduct a comprehensive analysis on regret in Eq. (2), and violation in Eq. (1) of *C2MAB-V*. Due to the page limit, the proof and analysis of the instance-dependent bound are provided in Appendix D.

To state the analysis, we firstly give some definitions. Let $o_{t,S}$ represent the probability that all LLMs within a selected action $S$ are observed at round $t$, and $r^* = \max_{S \in \mathcal{S}} r(S; \boldsymbol{\mu}) \leq NL$ be the maximum reward for the $L$-Lipschitz reward function $r(S; \boldsymbol{\mu})$.[4] Following Kveton et al. (2015a); Li et al. (2016), we define $o^* = \min_{t \in \mathcal{T}, S \in \mathcal{S}} o_{t,S}$ as the minimum observation probability across all

---

[4] All three reward functions satisfy 1-Lipschitz continuity, allowing a more general extension to the $L$-Lipschitz case (see Appendix C.2).

feasible combinations of LLM selection, within the context of partial feedback mechanisms. For the AWC scenario, as previously discussed, a subset may be chosen from multiple available options for an action $S_t$. In such cases, we provide the violation upper bound under the *worst-case* scenario, where each LLM in action $S_t$ fails to satisfy the user until the last one, i.e., $S_t = \mathcal{F}_t$ for all $t \in \mathcal{T}$.

**Theorem 1** (Regret Bound). *With $\delta = 1/T$ in the confidence radius, the $\alpha$-approximate regret for the multi-LLM selection problem is bounded as follows, with probability at least $1 - 1/T$:*

$$R(T) \leq \frac{2L}{o^*}\sqrt{2NKT\ln\left(\frac{2\pi^2 KT}{3}\right)} + (K+1)\,r^*. \tag{7}$$

**Remark 1.** *The C2MAB-V framework extends the capabilities of existing CMAB models by incorporating long-term cost considerations, which is typically not present in previous works Chen et al. (2016); Kveton et al. (2015c;b;a). Comparing with the linear CMAB model in Kveton et al. (2015c) which focuses on full rather than partial feedback, by setting $L = 1$ and $o^* = 1$, our regret bound aligns with theirs, adhering to the lower bound $\Omega(\sqrt{NKT})$ in Kveton et al. (2015c), up to a $\sqrt{\ln T}$ factor. The comparability of our regret bound with other CMAB frameworks primarily stems from the efficient management of uncertainty in parameter estimation. A unique difference in our approach, however, is the implementation of a discretization process for the relaxed NP-hard problem. Despite this, due to our efforts on the shared attributes of different reward functions (which is not an easy task), the regret of C2MAB-V maintains the same order as those observed in other CMAB studies.*

**Theorem 2** (Violation Bound). *With $\delta = 1/T$ in the confidence radius, the constraint violation under the worst case is bounded as follows, with probability at least $1 - 1/T$:*

$$V(T) \leq \sqrt{NK/T}\left(2\sqrt{2\ln\left(\frac{2\pi^2 KT}{3}\right)} + \sqrt{NK/T}\right). \tag{8}$$

**Remark 2.** *Our analysis reveals that the violation decreases at a rate of $\tilde{O}(\sqrt{\frac{K}{T}})$. As $T$ grows large, $V(T)$ approaches zero, suggesting an eventual elimination of violation. Furthermore, the overall violation is shown to be $\tilde{O}(\sqrt{KT})$, which is comparable to the order of regret. An interesting comparison arises with the work of Takemori et al. (2020), which studies non-linear submodular rewards and linear costs within the context of knapsack constraints, specifically focusing on scenarios with known and fixed costs. This contrasts with our exploration of scenarios characterized by unknown stochastic costs. Despite the inherent challenges posed by unknown costs, our framework manages to attain an $\tilde{O}(\sqrt{T})$ approximate regret with an approximation ratio of $\alpha_1 = 1 - \frac{1}{e} \approx 0.632$ under knapsack constraints. In contrast, the method from Takemori et al. (2020) achieves an $\tilde{O}(\sqrt{T})$ approximate regret with a lower approximation ratio $\alpha_2 \leq 0.5$. Thus, our approach secures a regret improvement of at least $(\alpha_1 - \alpha_2)T \geq 0.132T$. Compared to Sankararaman & Slivkins (2018), which focuses on addressing constraints, our work expands to include non-linear rewards.*

## 6 PERFORMANCE EVALUATION

**Experiment Settings.** We evaluate the three multi-LLM reward models (AWC, SUC, AIC) to represent different task types using the SciQ dataset Welbl et al. (2017) across nine LLMs: ChatGPT-3.5, ERNIE 3.5, ChatGPT-4, ChatGLM2, Llama 2-7B, Llama 2-13B, Llama 2-70B, Mixtral-8x7B, and Claude 2 OpenAI (2023); Claude (2023); Mistral-AI (2024); Baidu (2024); Ollama (2023).[5] Costs are determined based on the official pricing. Results are averaged over 10 seeds with a 95% confidence interval; for more setting details, refer to Appendix E.1. The maximum number $N$ is set uniformly to 4, according to the size of the LLM set. Budget threshold $\rho$ is 0.45 for AWC, 0.5 for SUC, and 0.3 for AIC, according to reward models and official LLM pricing. More experiments (e.g., varying budget threshold, impacts of maximum number, comparison to scenarios without relaxation, and evaluating results on more public datasets) are available in Appendix E.

**Comparison Benchmarks.** Comparisons include consistently utilizing the expensive ChatGPT-4 OpenAI (2023) and the cheap ChatGLM2 OpenAI (2023), the online CMAB algorithm *CUCB*, Wang

---

[5]Our evaluation focuses exclusively on online learning from continuous feedback, incorporating offline pre-training, requiring offline training or GPU resources, or addressing cold-start issues. Studies, such as Zhang et al. (2020), on accelerating online learning to overcome cold-start challenges, can be seamlessly integrated.

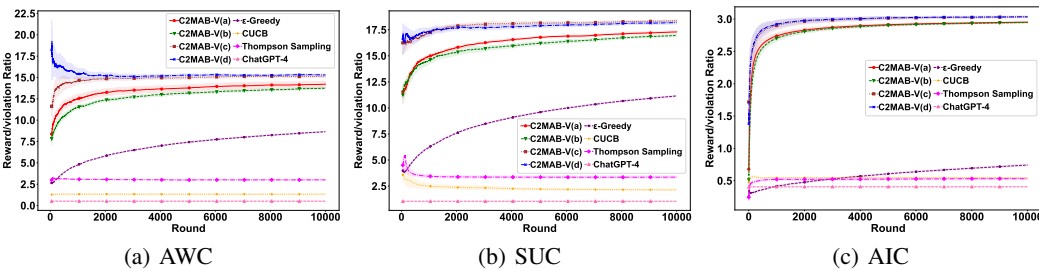

(a) AWC        (b) SUC        (c) AIC

Figure 4: Reward/violation ratio of three task types with nine different LLMs.

& Chen (2017); Liu et al. (2021c), which ignores constraints, Thompson Sampling, a mainstream probabilistic Bayesian online learning approach, Lattimore & Szepesvári (2020) and $\epsilon$-*Greedy* Auer et al. (2002), which alternates between using empirical means and selecting uniformly based on the adaptive $\epsilon_t = \min\{1, \frac{2\sqrt{K}}{\sqrt{t}}\}$ (Here, works of Sankararaman & Slivkins (2018); Takemori et al. (2020) are not compared due to being restricted to fixed costs or linear reward). The robustness of *C2MAB-V* is validated by varying the parameters $\alpha_\mu, \alpha_c$ with values of (0.3, 0.05), (1, 0.05), (0.3, 0.01), and (1, 0.01), referred to as (a), (b), (c), (d). Although the setting in Ding et al. (2024); Chen et al. (2023); Madaan et al. (2023); Zhu et al. (2023) totally differs from our online learning setting, we also pre-learned a fixed combination of multi-LLMs offline, which we applied to online queries. This enables to explore how online learning based on feedback can complement the adjustment of LLM selections in the offline domain. The results are provided in Appendix E.3 due to space constraints.

**Performance Metric.** To balance both reward and cost considerations, we assess performance using a reward/violation ratio, defined as the average per-round reward divided by the average per-round violation, i.e., $\frac{\sum_{\tau=1}^{t} r(S_\tau, \boldsymbol{\mu})/t}{\sum_{\tau=1}^{t} V(\tau)/t}$, with higher ratios indicating superior performance. More extended experimental results with the varying budget threshold and a greater emphasis on performance, along with detailed results for individual reward and violation, are provided in Appendix E.2.

**Evaluation Results.** The performance of the reward/ violation ratio is illustrated in Fig. 4. *ChatGLM2 is excluded since its rewards are significantly low*, below 0.18, 0.10, and 0.0001 in the AWC, SUC, AIC models, despite no violations. In the AWC model (Fig. 4(a)), *C2MAB-V* consistently outperforms other algorithms in all four parameter settings, achieving the highest reward/violation ratio. This underscores *C2MAB-V*'s superior ability to balance rewards against violations. Our algorithms converge in fewer than 1,000 rounds, significantly faster than the $\epsilon$-greedy algorithm, which requires nearly 10,000 rounds, even without incorporating offline information and relying entirely on online learning from scratch. The patterns observed for the SUC and AIC models (Fig. 4(b) and Fig. 4(c)) mirror those of the AWC model, further showing the robustness of *C2MAB-V* across different settings. When randomly evaluating *C2MAB-V(c)*, it improves by at least 64.72% over $\epsilon$-*Greedy*, performs 3.9× better than *Thompson Sampling*, 4×that of *CUCB*, and 6× that of *Always using ChatGPT-4* across three tasks. In summary, *C2MAB-V* maintains the highest reward/violation ratio under different $(\alpha_\mu, \alpha_c)$ for the ablation study on multi-LLM tasks aligning with our theoretical analysis.

## 7 CONCLUSION

In this paper, we introduce *C2MAB-V*, a cost-effective combinatorial online model with a versatile reward structure, designed to efficiently select multiple LLMs based on specific task requirements while adhering to budget constraints. *C2MAB-V* incorporates continual online feedback, and transform the NP-hard multi-LLM selection problem into a manageable relaxed form, with specified discretization rounding schemes designed to coordinate LLMs within a local-cloud architecture. Theoretical analysis of *C2MAB-V* provides novel and robust guarantees for the efficacy of the framework, including sublinear regret and a rapidly decreasing violation. Moreover, empirical evaluations with nine LLMs demonstrate *C2MAB-V*'s capability to balance performance with cost efficiency across three distinct types of collaborative multi-LLM tasks.

Looking ahead, there are several compelling directions for future research. For example, further development could focus on enhancing privacy protection and improving communication between multiple local servers within our local-cloud architecture. Additionally, incorporating contextual combinatorial multi-armed bandit approaches could more effectively capture user preferences.

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

# A CONTENT SUPPLEMENT

## A.1 SUMMARY OF MAIN SYMBOLS

Table 2: Summary of the main symbols.

| Symbol | Description |
|--------|-------------|
| $\mathcal{K}$ | Set of all LLMs, where $k$ indexes an LLM |
| $\mathcal{T}$ | Set of discrete time intervals or rounds |
| $S_t$ | Set of LLMs selected at round $t$, i.e., an action |
| $\mathcal{S}$ | Set of all possible combinations of LLM actions |
| $N$ | Maximum number of LLMs active simultaneously |
| $X_{t,k}$ | Reward associated with LLM $k$ at round $t$ |
| $y_{t,k}$ | Cost associated with LLM $k$ at round $t$ |
| $\mu_k$ | Unknown expected performance reward for LLM $k$ |
| $c_k$ | Unknown expected cost for LLM $k$ |
| $D_q$ | Distribution over queries |
| $l_k^{\text{in}}(q)$ | Number of input tokens by LLM $k$ for query $q$ |
| $l_k^{\text{out}}(q)$ | Random number of output tokens by LLM $k$ for query $q$ |
| $C_k$ | Cost per token for LLM $k$ |
| $D_k^{\text{out}}(q)$ | Distribution of output tokens for LLM $k$ given query $q$ |
| $\mathcal{Q}$ | Set of all queries |
| $\mathcal{F}_t$ | Subset of LLMs from which feedback is observed at round $t$ |
| $F_t$ | Cardinality of the subset $\mathcal{F}_t$ |
| $V(T)$ | Total budget violation over $T$ rounds |
| $R(T)$ | Total regret over $T$ rounds |
| $\rho$ | Budget guarantee threshold |
| $\delta$ | Probability parameter |
| $\hat{\mu}_{t,k}$ | Empirical estimate of the mean reward for LLM $k$ at round $t$ |
| $\hat{c}_{t,k}$ | Empirical estimate of the cost for LLM $k$ at round $t$ |
| $\alpha_\mu$ | Control parameter for adjusting the reward confidence bound |
| $\alpha_c$ | Control parameter for adjusting the cost confidence bound |
| $\rho_{t,\mu_k}$ | Confidence radius for the reward of LLM $k$ at round $t$ |
| $\rho_{t,c_k}$ | Confidence radius for the cost of LLM $k$ at round $t$ |
| $T_{t,\mu_k}$ | Count of selections of LLM $k$ concerning the reward up to round $t$ |
| $\sigma(\cdot)$ | Discretization rounding procedure for LLM Selection |
| $T_{t,c_k}$ | Count of selections of LLM $k$ concerning the cost up to round $t$ |
| $\tilde{z}_k$ | Continuous variable representing the selection status of LLM $k$ |
| $\tilde{Z}$ | Vector of $\tilde{z}_k$ for all LLMs |
| $o_{t,S}$ | Probability all LLMs within action $S$ are observed at round $t$ |
| $r^*$ | Maximum reward for the $L$-Lipschitz reward function |
| $o^*$ | Minimum observation probability across all LLM combinations |
| $L$ | Lipschitz constant for the reward function |

## A.2 MORE LITERATURE ON BANDIT WITH KNAPSACK CONSTRAINTS

Another line of research considers resource consumption and budget constraints in stochastic MAB/CMAB settings, with prominent examples including bandits with budgets Ding et al. (2013); Wu et al. (2015); Xia et al. (2016) and bandits with knapsacks (BwK) Badanidiyuru et al. (2013); Sankararaman & Slivkins (2018); Agrawal & Devanur (2019). In these studies, optimal stopping time is relevant because the learning process halts when resources are exhausted. Our problem differs in that our model's long-term constraint does not enforce such limitations, allowing the arm selection process to continue indefinitely. Among BwK studies, Sankararaman & Slivkins (2018) is most closely related to our work, as their proposed algorithm also employs a high-level combination of UCB/LCB, linear programming, and randomized rounding. However, their algorithm is limited to the linear CMAB setting with linear rewards and semi-bandit feedback. In the linear case, our

Table 3: Summary of the related constrained-effective MAB works.

| Algorithm | Combinatorial Arms? | Non-linear Reward? | Unknown Stochastic Cost? | Cost Constraint | Partial Feedback?* |
|---|---|---|---|---|---|
| POND, Liu et al. (2020) | × | × | ✓ | Long term | NA |
| OptPess-LP, Liu et al. (2021a) | × | × | ✓ | Long term | NA |
| Pessimistic-Optimistic, Liu et al. (2021b) | × | × | ✓ | Long term | NA |
| BwK† and its variants, Badanidiyuru et al. (2013); Agrawal & Devanur (2019) | × | × | ✓ | BwK† | NA |
| CUCB-CRA, Zuo & Joe-Wong (2021) | ✓ | ✓ | × | NA | × |
| CUCB-T and its variants, Chen et al. (2016); Li et al. (2016); Wang & Chen (2017) | ✓ | ✓ | × | NA | ✓ |
| SemiBwK-RRS, Sankararaman & Slivkins (2018) | ✓ | × | ✓ | BwK† | × |
| Constrained UCB, Chen et al. (2018) | ✓ | × | × | Long term | × |
| AFSM-UCB, Takemori et al. (2020) | ✓ | ✓ | × | Per round‡ | × |
| CA-CUCB, ours | ✓ | ✓ | ✓ | Long term | ✓ |

† BwK means the bandit with knapsacks;   * Partial feedback can cover applications with partially observed arms, e.g., cascading bandits;   ‡ Per round means the action satisfies the cost constraint in each round;

algorithm can encompass their approach by utilizing a linear relaxation function and a dependent rounding procedure under the SUC reward model. A key technical distinction between Sankararaman & Slivkins (2018) and our work is that we do not rely on the specific negative correlation property of randomized rounding (RR) for regret or violation analysis. Without assuming a negative correlation, our proof technique is more general, providing greater flexibility in choosing relaxation functions and rounding procedures for a broad class of non-linear reward functions.

### A.3 TACKLING NEW CHALLENGES IN COMBINATORIAL BANDITS

The key innovation of our approach lies in its formulation of *constraint-effective combinatorial multi-armed bandits*, a new variant of combinatorial semi-bandits. While previous CMAB research has made significant progress in minimizing regret, many existing approaches overlook the critical aspect of *cost constraints*, often allowing for excessive violations. Our work directly addresses this gap by adopting a constraint-effective perspective, where we aim to minimize both *regret* and *constraint violation* simultaneously. Specifically, our approach combines traditional regret minimization techniques with the novel challenge of managing cost constraints effectively.

A distinguishing feature of our work is the way we tackle the constrained optimization problem. Traditional methods typically treat the constraint as a *discrete* optimization problem, which is often NP-hard and computationally inefficient. Direct solutions to this problem tend to yield suboptimal approximation ratios. In contrast, we propose a *relaxation and rounding (RR)* technique, which transforms the problem into a *continuous* optimization problem. This transformation not only improves approximation guarantees but also offers better time complexity compared to traditional methods. However, implementing RR techniques in an *online setting* presents new challenges, as RR methods were originally designed for *offline* optimization problems. The flexibility in choosing relaxation functions and rounding procedures complicates the guarantee of low regret and low violation in the online case. To overcome this challenge, we propose a detailed analysis that identifies the conditions under which the RR approach can effectively perform in online scenarios. This involves establishing a connection between regret and over-estimation terms, which can be bounded using standard CMAB analysis. Additionally, we introduce a novel *martingale construction* technique that allows us to bound long-term violations of the constraints, a critical aspect in addressing online combinatorial optimization problems.

Furthermore, our approach handles non-linear reward functions and partial feedback, which are essential for addressing real-world LLM task scenarios. These challenges have not been comprehensively tackled in existing CMAB works.

## B DISCRETIZATION ROUNDING ALGORITHMS

For the AWC reward, as detailed in Algorithm 2, the scheduling cloud initiates the process by setting up a vector $v$ and identifying $K$ sets $\{\mathcal{B}_1, \cdots, \mathcal{B}_K\}$ (line 1). Following this initialization, Algorithm 2 enters an iterative phase where it fine-tunes the composition of the sets $\mathcal{B}_1$ and $\mathcal{B}_2$, aiming to keep the evolving solution within acceptable bounds while also seeking to optimize the reward (lines 3-9). Subsequent steps involve refining $\mathcal{B}_1$ by excluding all elements found in set $\mathcal{G}$, and updating $\mathcal{A}_{i+1}$ to reflect the current state of $\mathcal{B}_1$ (line 10), which are designed to gradually construct the final integrated solution. Finally, Algorithm 2 designates $\mathcal{A}_K$ as the final action of selected LLM $S_t$ (line 12). The validity of the resultant solution for LLM selection is underpinned by the lemma that follows (more discussions are in Appendix C.1).

---

**Algorithm 2** Discretization Rounding for LLM Selection with AWC Reward Model

---

**Input:** Relaxed continuous data $\tilde{\boldsymbol{Z}}_t$ from the local server.
1: **Initialize:** Find $\boldsymbol{v} = (v_1, \ldots, v_K)$ where $v_k \geq 0$, $\sum_{k \in \mathcal{K}} v_k = 1$ and sets $\mathcal{B}_1, \cdots, \mathcal{B}_K$ satisfying
    $\tilde{z}_{t,k} = \sum_{k \in \mathcal{K}} v_k \mathbb{I}_{\mathcal{B}_k}$; Set $\mathcal{A}_1 = \mathcal{B}_1$.
2: **for** $i = 1, \ldots, K - 1$ **do**
3:     $p_1 \leftarrow \sum_{j=1}^i v_j$, $p_2 \leftarrow v_{i+1}$, $\mathcal{B}_1 \leftarrow \mathcal{A}_i$, $\mathcal{B}_2 \leftarrow \mathcal{B}_{i+1}$;
4:     **if** $|\mathcal{B}_1| < |\mathcal{B}_2|$ **then** swap $p_1$ and $p_2$, $\mathcal{B}_1$ and $\mathcal{B}_2$;
5:     Find a set $\mathcal{G} \subseteq \mathcal{B}_1 \setminus \mathcal{B}_2$: cardinality $|\mathcal{G}| = |\mathcal{B}_1| - |\mathcal{B}_2|$ and $\mathcal{B}_2 \cup \mathcal{G} \subseteq \mathcal{S}$, and set $\mathcal{B}_2 \leftarrow \mathcal{B}_2 \cup \mathcal{G}$;
6:     **while** $\mathcal{B}_1 \neq \mathcal{B}_2$ **do**
7:         Find $i \in \mathcal{B}_1 \setminus \mathcal{B}_2$ and $j \in \mathcal{B}_2 \setminus \mathcal{B}_1$ satisfing $(\mathcal{B}_1 \setminus \{i\}) \cup \{j\} \in \mathcal{S}$ and $(\mathcal{B}_2 \setminus \{j\}) \cup \{i\} \in \mathcal{S}$;
8:         Set $\mathcal{B}_1 \leftarrow (\mathcal{B}_1 \setminus \{i\}) \cup \{j\}$ with probability $p_2/(p_1 + p_2)$, otherwise $\mathcal{B}_2 \leftarrow (\mathcal{B}_2 \setminus \{j\}) \cup \{i\}$;
9:     **end while**
10:    Set $\mathcal{B}_1 \leftarrow \mathcal{B}_1 \setminus \{i\}, \forall i \in \mathcal{G}$ with probability $p_2/(p_1 + p_2)$; $\mathcal{A}_{i+1} \leftarrow \mathcal{B}_1$;
11: **end for**
12: Return the final output $\mathcal{A}_K$ as the combinatorial LLM action $S_t$.

---

**Algorithm 3** Discretization Rounding for LLM Selection with SUC/AIC Reward Models

---

**Input:** Relaxed continuous data $\tilde{\boldsymbol{Z}}_t$ from the local server.
1: **while** exists $k \in \mathcal{K}$ such that $0 < \tilde{z}_{t,k} < 1$ **do**
2:     Identify distinct $k \neq j \in \mathcal{K}$, such that $0 < \tilde{z}_{t,k} < 1, 0 < \tilde{z}_{t,j} < 1$;
3:     Let $p = \min\{1 - \tilde{z}_{t,k}, \tilde{z}_{t,j}\}, q = \min\{\tilde{z}_{t,k}, 1 - \tilde{z}_{t,j}\}$;
4:     Update the pair $(\tilde{z}_{t,k}, \tilde{z}_{t,j}) \leftarrow \begin{cases} (\tilde{z}_{t,k} + p, \tilde{z}_{t,j} - p), & \text{with probability } \frac{q}{p+q}, \\ (\tilde{z}_{t,k} - q, \tilde{z}_{t,j} + q), & \text{with probability } \frac{p}{p+q}; \end{cases}$
5: **end while**
6: Return $\{k \in \mathcal{K} : \tilde{z}_{t,k} = 1\}$ as the selected LLM as the combinatorial LLM action $S_t$.

---

**Lemma 2** (Theorem 2.1 in Chekuri et al. (2009)). *For a matroid $\mathcal{M} = (K, \mathcal{S})$ with a rank function $r : 2^N \to Z_+$, the matroid polytope $P(\mathcal{M})$ is defined as the convex hull of the characteristic vectors of the independent sets $\mathcal{S}$, and the base polytope $B(\mathcal{M})$ as the convex hull of the characteristic vectors of the bases $\mathcal{B}$. For any two bases $\mathcal{B}_1, \mathcal{B}_2 \in \mathcal{B}$ and an element $i \in \mathcal{B}_1 \setminus \mathcal{B}_2$, there exists an element $j \in \mathcal{B}_2 \setminus \mathcal{B}_1$ such that $(\mathcal{B}_1 \setminus \{i\}) \cup \{j\}$ and $(\mathcal{B}_2 \setminus \{j\}) \cup \{i\}$ also belong to $\mathcal{B}$.*

For the SUC and AIC reward, as elucidated in Algorithm 3, the process commences with the examination of set $\mathcal{K}$ to identify LLM $k$ such that their corresponding value $\tilde{z}_{t,k}$ lies strictly between 0 and 1 (line 1). Subsequently, Algorithm 3 progresses by identifying a pair of distinct LLM, $k$ and $j$, within $\mathcal{K}$, both of which satisfy the criterion $0 < \tilde{z}_{t,k}, \tilde{z}_{t,j} < 1$ (line 2). The core of the discretization procedure involves computing the probabilities for adjusting the values of $\tilde{z}_{t,k}$ and $\tilde{z}_{t,j}$. This is achieved by determining $p$ and $q$, which represent the minimum increments and decrements needed for the adjustment, ensuring the discretization stays within bounds (line 3). With these parameters, Algorithm 3 probabilistically updates the pair $(\tilde{z}_{t,k}, \tilde{z}_{t,j})$ to either increase $\tilde{z}_{t,k}$ and decrease $\tilde{z}_{t,j}$, or vice versa, thus balancing the overall distribution (line 4). Finally, Algorithm 3 concludes by assembling the set $S_t$ comprised of all $k \in \mathcal{K}$ for which $\tilde{z}_{t,k} = 1$, marking them as the selected LLM for the given timestep (line 6). This discrete selection process, grounded in the probabilistic adjustments of the $\tilde{z}$ values, systematically refines the selection of LLM.

One fundamental aspect of the reward $r(S; \boldsymbol{\mu})$ is its demonstration of the "diminishing marginal" property. Specifically, when a new LLM $k \in \mathcal{K}$ is added into action $S_j$ with a relatively larger set size, the resultant increase in reward is less than or equal to the increase observed when the same LLM $k$ is added to a smaller set $S_i$, given $S_i \subseteq S_j \subseteq \mathcal{S}, i, j \in \mathcal{T}$. This behavior exemplifies a submodular function, formally expressed as follows:

$$r(S_i \cup \{k\}; \boldsymbol{\mu}) - r(S_i; \boldsymbol{\mu}) \geq r(S_j \cup \{k\}; \boldsymbol{\mu}) - r(S_j; \boldsymbol{\mu}). \tag{9}$$

By applying the common greedy algorithm to solve the maximization of submodular function problem, we have the following lemma:

**Lemma 3** (Theorem 1 in Sun et al. (2023)). *The problem of maximizing a submodular function can be efficiently solved by a greedy algorithm, achieving an approximate ratio of $\alpha = (1 - 1/e)$ relative to the theoretical optimum.*

This lemma indicates a significant efficiency in approximating the optimal solution of the submodular reward function $r(S; \boldsymbol{\mu})$. Attentive readers will recognize that the previously discussed three different reward functions (AWC, SUC, and AIC reward) are actually special cases of submodular functions, all with monotonicity and Lipschitz coninuity condition satisfied. However, we focus on designing and discussing different algorithmic components that offer better performance guarantees with $\alpha = 1$ for SUC and AIC, rather than merely categorizing them as submodular functions.

## C  CONSTRAINT AND REWARD ANALYSIS

### C.1  DISCUSSIONS ON CONSTRAINT TYPE AND EXTENDED TASK TYPE

In the context of *C2MAB-V*, we define ground base arms as $\mathcal{K} = \{1, ..., K\}$ and denote $\mathcal{S}$ as combinatorially feasible sets, structured by specific combinatorial frameworks. A set $\mathcal{S}$ is *linearizable* if its convex hull forms a polytope in $\mathbb{R}^K$, meaning we can describe it using a finite number of linear constraints in $\mathbb{R}^K$, with $\mathcal{S}$ representing the integral solutions. *Matroids* represent a prominent category of linearizable combinatorial sets, characterized by:

- **Containment of the empty set:** $\emptyset \in \mathcal{S}$.
- **Downward closure:** For any $S \in \mathcal{S}$ and $S' \subset S$, it follows that $S' \in \mathcal{S}$.
- **Exchange property:** For $S, S' \in \mathcal{S}$ with $|S'| > |S|$, there exists an $k \in S' \setminus S$ such that $S \cup \{i\} \in \mathcal{S}$.

For any $S \in \mathcal{S}$, the rank function $rank(S)$ is defined as the maximal size of independent subsets in $S$, and $\mathbb{I}_S$ indicates a vector in $\{0,1\}^K$ representing the membership of elements in $S$. The polytope induced by $\mathcal{S}$, denoted as $P(\mathcal{S})$, is defined as the convex hull of these characteristic vectors, encapsulating the feasible integral solutions for $\mathcal{S}$.

The *base* of a matroid, $\mathcal{B}$, consists of independent sets of maximum size $N$, and the base polytope, $B(\mathcal{S})$, is the intersection of $P(\mathcal{S})$ with a hyperplane defined by $\sum_{k \in \mathcal{K}} \tilde{z}_k = N$. Examples of matroids include:

1. **Cardinality-constrained subsets:** Where $\mathcal{S} = \{S \subseteq \mathcal{K} : |S| \leq N\}$.

2. **Partition matroids:** Defined over disjoint subsets $\mathcal{D}_1, ..., \mathcal{D}_M$ of $\mathcal{K}$ with cardinality constraints $d_1, ..., d_M$.

3. **Spanning trees:** For a graph $G = (V, \mathcal{K})$, $\mathcal{S}$ includes all subsets of $\mathcal{K}$ that form a tree covering all vertices in $V$.

For each matroid type, the corresponding polytope $P(\mathcal{S})$ is detailed by a linear program, reflecting its combinatorial structure and constraints.

Attentive readers may have noticed that our main text's discussion of three different types of tasks involving LLM collaboration, namely the AWC, SUC, and AIC rewards, as well as their corresponding constraints, fundamentally relates to concepts associated with matroids and their bases constrained by cardinality. To facilitate better understanding, we have organized this information as follows:

**Matroids and Their Bases Subject to Cardinality Constraints.** For a given fixed $N$, a subset $S \subset \mathcal{K}$ can belong to $\mathcal{S}$ either if $|S| \leq N$ or $|S| = N$. The former is considered in the AWC application to guarantee user satisfaction and experience, where each feasible action selects at most $N$ LLMs. The latter case is used in settings of SUC on independently tackling tasks and AIC on developing a whole project, where exactly $N$ LLMs are selected.

The corresponding induced polytopes $P(\mathcal{S})$ can be described by a vector $\tilde{\boldsymbol{Z}} \in \mathbb{R}^K$ that adheres to the following linear programs:

1. *For subsets with size at most N (Inclusive Matroids):*

$$\sum_{k \in \mathcal{K}} \tilde{z}_k \leq N, \tag{10}$$

$$\tilde{z}_k \in [0, 1], \text{ for } \forall\, k \in \mathcal{K}. \tag{11}$$

2. *For subsets with size exactly N (Base Matroids):*

$$\sum_{k \in \mathcal{K}} \tilde{z}_k = N, \tag{12}$$

$$\tilde{z}_k \in [0, 1], \text{ for } \forall\, k \in \mathcal{K}. \tag{13}$$

This structure clarifies the types of constraints applied to different combination of multiple LLMs for different task types.

However, the generalization of our framework design extends beyond this; it is also applicable to other types of settings. To illustrate this, consider a specific analytical example that employs Partition matroids as the foundational structure. Suppose we have a collection of disjoint subsets $\mathcal{D}_1, \ldots, \mathcal{D}_M$ of $\mathcal{K}$, with $\mathcal{M} = \{1, 2, ..., M\}$. This can be interpreted as further domain-specific divisions within a complete set of LLMs, $\mathcal{K}$, such as dedicating groups of non-overlapping LLMs specialized in different subjects like mathematics and physics, or each LLM being responsible for a specific submodule within a large project. Additionally, the constraints can be expanded further; for example, each submodule may have different budget requirements. Formally, this involves $M$ cardinality constraints $d_1, \ldots, d_M$. A subset $S \subset \mathcal{K}$ belongs to $\mathcal{S}$ if and only if $|S \cap \mathcal{D}_i| \leq d_i$ for some fixed $N$, where $\mathcal{S} = \{S \subseteq \mathcal{K} : |S \cap \mathcal{D}_i| \leq d_i, i \in \mathcal{M}\}$. This combinatorial feasible set $\mathcal{S}$ can model products that belong to mutually exclusive categories.

The corresponding induced polytope $P(\mathcal{S})$ can be described by $\tilde{Z} \in \mathbb{R}^K$ that follows the linear program:

$$\sum_{i \in \mathcal{D}_j} \tilde{z}_k \leq d_j,\ \forall\, j \in \mathcal{M}$$

$$\tilde{z}_k \in [0, 1], \text{ for } \forall\, k \in \mathcal{K}.$$

For instance, utilizing the constraints outlined above, combined with the AIC reward structure, we can easily apply our model to coordinate tasks like a development project where each LLM is responsible for different non-overlapping modules. This allows for online selection of an optimal combination of multiple LLMs tailored to the specific needs of the project. Alternatively, by combining partition matroids and SUC reward, it is possible to address a unified teaching task with different LLMs focusing on independent but complementary academic disciplines. Thus, within the *S* framework, based on our defined reward structure and various types of constraints, we can effectively handle a wide range of collaborative tasks among multiple LLMs.

Based on the above analysis, regarding extended tasks, such as identifying the set of LLMs to be deployed, are addressed, the strategy lacks specificity on which LLM should be assigned to a particular task (as in the SUC case) or which LLM should handle a given sub-module (as in the AIC case). This issue can be mitigated by transitioning from cardinality-constrained subsets to partition matroids. Specifically, a smaller and more suitable subset of LLMs can be preselected for specific query tasks. For example, in mathematical problems, the selection space for LLMs could be restricted to those fine-tuned for mathematical queries, rather than choosing from the full set of LLMs. This targeted selection not only reduces the search space but also accelerates the online learning process by avoiding the misallocation of resources.

Alternatively, the selection process can rely entirely on online learning through match bandit. The match bandit problem, a variant of the maximum weight matching problem, is modeled as a bipartite graph $G = (Q, K, E)$, where $Q$ represents query tasks, $K$ represents LLMs, and $E$ represents weighted edges denoting the probability of assigning a query task $q$ to an LLM $k$. The Combinatorial MAB proposed in our paper can naturally be transformed into this bipartite graph structure based on .Liu et al. (2023b); Chen et al. (2016).

## C.2 EXPLORATION ON ATTRIBUTES OF REWARD FUNCTIONS

Next, we examine the common attributes of the three defined reward functions. All three corresponding constraint optimization problems (defined in Eq. (3), Eq. (4), and Eq. (5)) are linear programming (LP) models with $K$ variables. The optimal solutions for these models, i.e., $\alpha = 1 - e$ for the AWC reward and $\alpha = 1$ for the SUC and AIC rewards, can be determined in polynomial time Vaidya (1989); Calinescu et al. (2007); Iyer et al. (2014).

Next, we define the discretization procedure $\sigma$ used in Algorithm 2 and Algorithm 3 as $S_t = \sigma(\tilde{\boldsymbol{Z}}_t)$, where $\mathbb{I}_{S_t^*}$ represents the feasible solutions of the optimization problem. The indicator variable $\mathbb{I}_S = \{z_1, z_2, \cdots, z_K\} \in \{0,1\}^K$ denotes the selection status of LLMs in the set $\mathcal{K}$. Referencing Theorem 1.1 in Chekuri et al. (2009) for Algorithm 2 and Properties P1 and P2 in Gandhi et al. (2006) for Algorithm 3, it is established that $\mathbb{E}_{S \sim \sigma(\tilde{\boldsymbol{Z}}_t)}[\mathbb{I}_S] = \tilde{\boldsymbol{Z}}_t$ for any $S \sim \sigma(\tilde{\boldsymbol{Z}}_t)$ within the feasible set $\mathcal{S}$. Consequently, we can conclude that:

$$r(S, \boldsymbol{\mu}) = \begin{cases} \left(1 - \prod_{k \in S}(1 - \mu_k)\right) &= \tilde{r}(\mathbb{I}_S, \boldsymbol{\mu}), & \text{AWC reward,} \\ \sum_{k \in S} \mu_k &= \tilde{r}(\mathbb{I}_S, \boldsymbol{\mu}), & \text{SUC reward,} \\ \prod_{k \in S} \mu_k &= \tilde{r}(\mathbb{I}_S, \boldsymbol{\mu}), & \text{AIC reward.} \end{cases} \tag{14}$$

Following Eq. (14), we can then deduce that $r(S_t^*, \bar{\boldsymbol{\mu}}_t) = \tilde{r}(\mathbb{I}_{S_t^*}, \bar{\boldsymbol{\mu}}_t)$.

Finally, we establish a relationship between the actual reward function and its relaxed counterpart by demonstrating the inequality $\mathbb{E}[\tilde{r}(\mathbb{I}_{S_t}, \boldsymbol{\mu})] \geq \tilde{r}(\tilde{\boldsymbol{Z}}_t, \boldsymbol{\mu})$.

❶ For the AWC reward, the proof can be streamlined as follows: Based on Algorithm 2, the process transitions from an initial continuous solution $\tilde{\boldsymbol{Z}}$ to a final integral solution $\boldsymbol{X}_s = \mathbb{I}_S$ over $s$ steps, resulting in the sequence $(\boldsymbol{X}_1, ..., \boldsymbol{X}_s)$, where $\boldsymbol{X}_1 = \tilde{\boldsymbol{Z}}$ and $\boldsymbol{X}_s = \mathbb{I}_S$. Utilizing Lemma B.1 and Lemma 4.1 from Chekuri et al. (2009), at any round $t$, we can represent $\boldsymbol{X}_t$ as a weighted sum of indicator functions for bases, $\boldsymbol{X}_t = \sum_{l=1}^k p_l \mathbb{I}_{\mathcal{B}_l}$, where $\mathcal{B}_l$ represents the bases. The state transition to $\boldsymbol{X}_{t+1}$ is modeled as follows:

$$\boldsymbol{X}_{t+1} = \begin{cases} \boldsymbol{X}_t + p_2(\mathbb{I}_{\{i\}} - \mathbb{I}_{\{j\}}), & \text{with probability } \frac{p_1}{p_1 + p_2}, \\ \boldsymbol{X}_t - p_1(\mathbb{I}_{\{i\}} - \mathbb{I}_{\{j\}}), & \text{with probability } \frac{p_2}{p_1 + p_2}. \end{cases}$$

Defining $a = \boldsymbol{X}_t$ and $\vartheta = \mathbb{I}_{\{i\}} - \mathbb{I}_{\{j\}}$, we let $h(p) = \tilde{r}(\boldsymbol{X}_t + p\vartheta, \boldsymbol{\mu})$. The expected value of $\tilde{r}$ at $t + 1$ given $\boldsymbol{X}_t$ is then

$$\mathbb{E}[\tilde{r}(\boldsymbol{X}_{t+1}, \boldsymbol{\mu})|\boldsymbol{X}_t] = \frac{p_1}{p_1 + p_2}\tilde{r}(\boldsymbol{X}_t + p_2\vartheta, \boldsymbol{\mu}) + \frac{p_2}{p_1 + p_2}\tilde{r}(\boldsymbol{X}_t - p_1\vartheta, \boldsymbol{\mu}),$$

which simplifies to $\frac{p_1}{p_1 + p_2}h(p_2) + (1 - \frac{p_1}{p_1 + p_2})h(-p_1)$. Due to the convexity of $h$, this is at least $h(0) = \tilde{r}(\boldsymbol{X}_t, \boldsymbol{\mu})$. Applying this inequality recursively from $t = 1$ to $s - 1$ via the tower rule, we conclude that $\mathbb{E}[\tilde{r}(\boldsymbol{X}_s, \boldsymbol{\mu})] \geq \tilde{r}(\boldsymbol{X}_1, \boldsymbol{\mu}) = \tilde{r}(\tilde{\boldsymbol{Z}}, \boldsymbol{\mu})$, thereby satisfying $\mathbb{E}[\tilde{r}(\mathbb{I}_{S_t}, \boldsymbol{\mu})] \geq \tilde{r}(\tilde{\boldsymbol{Z}}_t, \boldsymbol{\mu})$.

❷ For the SUC reward, recall that indicator variable $\mathbb{I}_S = \{z_1, z_2, \cdots, z_K\} \in \mathbb{R}^K$, with $z_k = 1$ indicating LLM $k$ selected, and $z_k = 0$ otherwise, and $\tilde{z}_k \in [0, 1]$ is a continuous variable , with $\tilde{\boldsymbol{Z}} = \{\tilde{z}_1, \tilde{z}_2, \cdots, \tilde{z}_K\}$. On the basis of Eq. (14), it holds that $\mathbb{E}_{S \sim \sigma(\tilde{\boldsymbol{Z}})}[\tilde{r}(\mathbb{I}_S, \boldsymbol{\mu})] = \sum_{k \in \mathcal{K}} \mathbb{E}[z_k]\mu_k = \sum_{k \in \mathcal{K}} \tilde{z}_k \mu_k = \tilde{r}(\tilde{\boldsymbol{Z}}, \boldsymbol{\mu})$. This fulfills that $\mathbb{E}[\tilde{r}(\mathbb{I}_{S_t}, \boldsymbol{\mu})] \geq \tilde{r}(\tilde{\boldsymbol{Z}}_t, \boldsymbol{\mu})$.

❸ For the AIC reward, consider the vector $\boldsymbol{Z} := (z_1, ..., z_K) \in \mathbb{R}^K$, where $\tilde{r}(\boldsymbol{Z}, \boldsymbol{\mu}) = \exp\left(\sum_{k \in \mathcal{K}} z_k \ln \mu_k\right)$ is demonstrated to be a convex function with respect to $\boldsymbol{Z}$. This is evidenced by the Hessian matrix of $\tilde{r}(\boldsymbol{Z}, \boldsymbol{\mu})$, denoted as $H_{\tilde{r}}(\boldsymbol{Z}) \in \mathbb{R}^K \times \mathbb{R}^K$. The entries of $H_{\tilde{r}}(\boldsymbol{Z})$ are given by $H_{\tilde{r}}(\boldsymbol{Z})_{i,j} = \ln \mu_i \ln \mu_j \tilde{r}(\boldsymbol{Z}, \boldsymbol{\mu})$. The Hessian can be expressed as $H_{\tilde{r}}(\boldsymbol{Z}) = \tilde{r}(\boldsymbol{Z}, \boldsymbol{\mu})\boldsymbol{W}\boldsymbol{W}^T$, where $\boldsymbol{W}$ is the column vector $\boldsymbol{W} := (\ln \mu_1, ..., \ln \mu_K)$. Consequently, $H_{\tilde{r}}(\boldsymbol{Z})$ is positive semidefinite because for any vector $\boldsymbol{x} \in \mathbb{R}^K$, we have $\boldsymbol{x}^T H_{\tilde{r}}(\boldsymbol{Z})\boldsymbol{x} = \tilde{r}(\boldsymbol{Z}, \boldsymbol{\mu})(\boldsymbol{W}^T x)^2 \geq 0$, given that $\tilde{r}(\boldsymbol{Z}, \boldsymbol{\mu}) \geq 0$ and $(\boldsymbol{W}^T x)^2 \geq 0$. Applying Jensen's inequality and acknowledging that $\mathbb{E}_{\boldsymbol{Z}}[z_k] = \tilde{z}_k$ as per Eq. (14), it follows that $\mathbb{E}_{\boldsymbol{Z}}[\exp\left(\sum_{k \in \mathcal{K}} z_k \ln \mu_k\right)] \geq \exp\left(\sum_{k \in \mathcal{K}} \mathbb{E}_{\boldsymbol{Z}}[z_k] \ln \mu_k\right) = \tilde{r}(\tilde{\boldsymbol{Z}}, \boldsymbol{\mu})$, thereby ensuring that $\mathbb{E}[\tilde{r}(\mathbb{I}_{S_t}, \boldsymbol{\mu})] \geq \tilde{r}(\tilde{\boldsymbol{Z}}_t, \boldsymbol{\mu})$.

In addition to the above observations, it is crucial to evaluate the sensitivity of the reward functions with respect to the input parameter $\boldsymbol{\mu}$. This sensitivity is commonly measured using the concept of

"*Lipschitz continuity*", which ensures that small changes in $\boldsymbol{\mu}$ result in proportionally small changes in the reward function.

**Lipschitz continuity.** A reward function $r(S; \boldsymbol{\mu})$ is said to be $L$-Lipschitz continuous with respect to $\boldsymbol{\mu}$ if, for any $\boldsymbol{\mu}, \boldsymbol{\mu}' \in [0,1]^K$ and any action $S$, it satisfies: $|r(S; \boldsymbol{\mu}) - r(S; \boldsymbol{\mu}')| \leq L \sum_{k \in S} |\boldsymbol{\mu}_k - \boldsymbol{\mu}'_k|$.

For the reward models AIC, SUC, and AWC, we observe that all are 1-Lipschitz continuous.

**AIC:** $|r(S; \boldsymbol{\mu}) - r(S; \boldsymbol{\mu}')| = |\prod_{k \in S} \mu_k - \prod_{k \in S} \mu'_k| = \left| \sum_{k=1}^{K} \left( \left( \prod_{j=1}^{k-1} \mu_j \right) (\boldsymbol{\mu}_k - \mu'_k) \left( \prod_{j=k+1}^{K} \mu_j \right) \right) \right| \leq \sum_{k=1}^{K} \left| \left( \left( \prod_{j=1}^{k-1} \mu'_j \right) (\boldsymbol{\mu}_k - \mu'_k) \left( \prod_{j=k+1}^{K} \mu_j \right) \right) \right| \leq \sum_{k=1}^{K} |\mu_k - \mu'_k|$.

**SUC:** $|r(S; \boldsymbol{\mu}) - r(S; \boldsymbol{\mu}')| = |\sum_{k \in S} \mu_k - \sum_{k \in S} \mu'_k| \leq \sum_{k=1}^{K} |\mu_k - \mu'_k|$.

**AWC:** $|r(S; \boldsymbol{\mu}) - r(S; \boldsymbol{\mu}')| = |\prod_{k \in S} (1 - \mu_k)) - \prod_{k \in S} (1 - \mu_k)')|$, which follows the same derivation as AIC by substituting $\lambda = \mu$.

# D PROOF APPENDIX

## D.1 PROOF OF LEMMA 1

*Proof.* We aim to establish a bound for the probability given by:

$$\Pr\{\neg \mathcal{K}_\mu\} = \Pr \left\{ \exists t \in \mathcal{T}, k \in \mathcal{K}, |\hat{\mu}_{t,k} - \mu_k| \geq \sqrt{\frac{\ln(\frac{2\pi^2 K t^3}{3\delta})}{2T_{t,\mu_k}}} \right\}$$

which can be expressed as follows:

$$\leq \sum_{k=1}^{K} \sum_{t=1}^{T} \sum_{s=1}^{t} \Pr \left\{ |\hat{\mu}_{t,k} - \mu_k| \geq \sqrt{\frac{\ln(\frac{2\pi^2 K t^3}{3\delta})}{2T_{t,\mu_k}}}, T_{t,\mu_k} = s \right\}$$

$$\leq \sum_{k=1}^{K} \sum_{t=1}^{T} \sum_{s=1}^{t} \frac{3\delta}{\pi^2 K} \frac{1}{t^3}$$

$$\leq \frac{\delta}{2}.$$

The initial inequality employs the union bound over the indices $k, t$, and $s$. The subsequent inequality leverages the Chernoff-Hoeffding inequality (see Lemma 5) when $T_{s,\mu_k} = s$ and $\hat{\mu}_{t,k} = \frac{1}{s} \sum_{j=1}^{s} \mu_k^j$ is the sample mean of $s$ independent and identically distributed random variables $\mu_k^1, \ldots, \mu_k^s$, each representing the $j$-th observation of index $k$. The final inequality is justified by the series sum $\sum_{k=1}^{\infty} \frac{1}{k^2} = \frac{\pi^2}{6}$.

Analogously, for the probability concerning $\neg \mathcal{K}_c$, we derive:

$$\Pr\{\neg \mathcal{K}_c\} = \Pr \left\{ \exists t \in \mathcal{T}, k \in \mathcal{K}, |\hat{c}_{t,k} - c_k| \geq \sqrt{\frac{\ln(\frac{2\pi^2 K t^3}{3\delta})}{2T_{t,c_k}}} \right\}$$

$$\leq \sum_{k=1}^{K} \sum_{t=1}^{T} \sum_{s=1}^{t} \Pr \left\{ |\hat{c}_{t,k} - c_k| \geq \sqrt{\frac{\ln(\frac{2\pi^2 K t^3}{3\delta})}{2T_{t,c_k}}}, T_{t,c_k} = s \right\}$$

$$\leq \sum_{k=1}^{K} \sum_{t=1}^{T} \sum_{s=1}^{t} \frac{3\delta}{\pi^2 K} \frac{1}{t^3}$$

$$\leq \frac{\delta}{2}.$$

Through consideration of the complementary events, we ascertain that the events $\mathcal{K}_\mu$ and $\mathcal{K}_c$ each occur with a probability of at least $1 - \frac{\delta}{2}$, respectively. $\square$

## D.2  COMPREHENSIVE PROOF OF THEOREM 1

**General regret proof analysis**

The proof of Theorem 1 is structured into three pivotal segments. Initially, it focuses on transforming the total regret into over-estimation regret (**Step 1: Reduce to the over-estimation regret**). Subsequently, it addresses the conversion of partially observed actions to fully observed ones, leveraging the Lipschitz condition to dissect the regret associated with an action into the regrets of its constituent base arms (**Step 2: Deal with the partial observation and apply the $L$-Lipschitz condition**). The final segment synthesizes these elements through meticulous mathematical manipulations (**Step 3: Derivation to get the final regret bound**).

We begin with a sketch of the proof to enhance comprehension. From Lemma 1, for every round $t \in \mathcal{T}$ and LLM $k \in \mathcal{K}$, the inequality $|\hat{\mu}_{t,k} - \mu_k| < \sqrt{\ln\left(\frac{2\pi^2 K t^3}{3\delta}\right)/2T_{t,\mu_k}}$ is satisfied with probability at least $1 - \delta/2$. By setting $\delta = 1/T$, we can assert with high probability that $\mu_k \leq \hat{\mu}_{t,k} + \rho_{\mu,t,k} = \bar{\mu}_{t,k}$ and $c_k \geq \hat{c}_{t,k} - \rho_{c,t,k} = \underline{c}_{t,k}$.

Given the monotonicity property of reward function $r(S; \boldsymbol{\mu})$ and $r(S_t^*, \bar{\boldsymbol{\mu}}_t) = \tilde{r}\left(\mathbb{I}_{S_t^*}, \bar{\boldsymbol{\mu}}_t\right)$, we derive the following inequality:

$$R(T) \leq \mathbb{E}\left[\alpha r\left(S_t^*, \bar{\boldsymbol{\mu}}_t\right) - r\left(S_t, \boldsymbol{\mu}\right)\right] \leq \mathbb{E}\left[\tilde{r}\left(\tilde{\boldsymbol{Z}}_t, \bar{\boldsymbol{\mu}}_t\right) - r\left(S_t, \boldsymbol{\mu}\right)\right]. \tag{15}$$

Substituting $\mathbb{E}\left[\mathbb{E}\left[\tilde{r}\left(\mathbb{I}_{S_t}, \bar{\boldsymbol{\mu}}_t\right)\right]\right] = \mathbb{E}\left[\tilde{r}\left(\mathbb{I}_{S_t}, \bar{\boldsymbol{\mu}}_t\right)\right] = \mathbb{E}\left[R\left(S_t, \bar{\boldsymbol{\mu}}_t\right)\right]$ into Eq. (15), we obtain:

$$R(T) \leq \mathbb{E}\underbrace{\left[\tilde{r}\left(\tilde{\boldsymbol{Z}}_t, \bar{\boldsymbol{\mu}}_t\right) - \mathbb{E}\left[\tilde{r}\left(\mathbb{I}_{S_t}, \bar{\boldsymbol{\mu}}_t\right)\right]\right]}_{\text{regret (a)}} + \mathbb{E}\underbrace{\left[r\left(S_t, \bar{\boldsymbol{\mu}}_t\right) - r\left(S_t, \boldsymbol{\mu}\right)\right]}_{\text{regret (b)}}. \tag{16}$$

Regret (a) arises from the relaxation and discretization, which is no greater than zero due to the inequality $\mathbb{E}\left[\tilde{r}\left(\mathbb{I}_{S_t}, \bar{\boldsymbol{\mu}}_t\right)\right] \geq \tilde{r}\left(\tilde{\boldsymbol{Z}}_t, \bar{\boldsymbol{\mu}}_t\right)$. Regret (b) is a result of the overestimating of rewards. Following the analysis idea in Wang & Chen (2017) to bound this over-estimation regret, the observation probability $o^*$ is used to derive:

$$\mathbb{E}\left[r\left(S_t, \bar{\boldsymbol{\mu}}_t\right) - r\left(S_t, \boldsymbol{\mu}\right)\right]$$

$$\leq \frac{1}{o^*}\mathbb{E}\left[\sum_{t=1}^{T} r\left(S_t, \bar{\boldsymbol{\mu}}_t\right) - r\left(S_t, \boldsymbol{\mu}\right)\mathbb{I}\left\{F_t = |S_t|\right\}\right]$$

$$\leq \frac{L}{o^*}\mathbb{E}\left[\sum_{t=1}^{T}\sum_{k\in\mathcal{F}_t}|\bar{\mu}_{t,k} - \mu_{t,k}|\right]$$

$$\leq \frac{L}{o^*}\mathbb{E}\left[\sum_{t=1}^{T}\sum_{k\in\mathcal{F}_t} 2\rho_{t,\mu_k}\right].$$

By synthesizing the bounds of regret components (a) and (b), we can establish an upper bound for the regret.

We proceed with a comprehensive proof. This detailed analysis is predicated on the concurrent occurrence of both $\mathcal{K}_\mu$ and $\mathcal{K}_c$ events, with probability at least $1 - \delta$ (by Lemma 1). In scenarios where $\mathcal{K}_\mu$ and $\mathcal{K}_c$ fail to occur, the expected additional regret is bounded by $r^*\delta T$.

**Step 1: Reduce to the over-estimation regret**

We begin by recalling the definition of the optimal action $S_t^* \triangleq \arg\max_{S\in\mathcal{S}} r(S; \boldsymbol{\mu}_t)$ for each round $t$, where $\mathcal{S}$ encompasses all feasible actions. And $\tilde{\boldsymbol{Z}}_t$ is the $\alpha$-approximate solution of $\{\max \tilde{r}(\tilde{\boldsymbol{Z}}, \bar{\boldsymbol{\mu}}) : \tilde{\boldsymbol{Z}} \in P(\mathcal{S}), \sum_{k\in\mathcal{K}} \underline{c}_{t,k}\tilde{z}_k \leq \rho\}$, where $P(\mathcal{S})$ is defined as the convex hull induced by $\mathcal{S}$:

$$P(\mathcal{S}) := \text{conv}\{\mathbb{I}_S : S \in \mathcal{S}\} = \{\tilde{\boldsymbol{Z}} \in \mathbb{R}^K : \tilde{z}_k \in [0,1], \sum_{k\in S}\tilde{z}_k \leq rank(S), \text{ for } \forall k\in\mathcal{K}, \forall S\subseteq\mathcal{K}\}.$$

Denote the discretization procedure $\sigma$ for Algorithm 2 and Algorithm 3 as $S_t = \sigma(\tilde{\boldsymbol{Z}}_t)$. The feasible solution $S_t = \sigma(\tilde{\boldsymbol{Z}}_t)$ guarantees $r(S; \boldsymbol{\mu}) \geq \alpha \cdot r(S^*; \boldsymbol{\mu})$. Under high probability events $\mathcal{K}_\mu$ and $\mathcal{K}_c$, the constraints $\mu_k \leq \hat{\mu}_{t,k} + \rho_{t,\mu_k} = \bar{\mu}_{t,k}$ and $c_k \geq \hat{c}_{t,k} - \rho_{t,c_k} = \underline{c}_{t,k}$ are satisfied. We first apply the monotonicity of the reward function and the property of the multi-linear extension to reduce the total regret to the over-estimation regret,

$$\mathbb{E}[(\alpha r(S_t^*, \boldsymbol{\mu}) - r(S_t, \boldsymbol{\mu}))]$$

$$\leq \mathbb{E}[(\alpha r(S_t^*, \bar{\boldsymbol{\mu}}_t) - r(S_t, \boldsymbol{\mu}))] \tag{17}$$

$$\leq \mathbb{E}[(\tilde{r}(\tilde{\boldsymbol{Z}}_t, \bar{\boldsymbol{\mu}}_t) - r(S_t, \boldsymbol{\mu}))] \tag{18}$$

$$= \mathbb{E}[(\underbrace{\tilde{r}(\tilde{\boldsymbol{Z}}_t, \bar{\boldsymbol{\mu}}_t) - \mathbb{E}_{S_t \sim \sigma(\tilde{\boldsymbol{Z}}_t)}[\tilde{r}(\mathbb{I}_{S_t}, \bar{\boldsymbol{\mu}}_t)]}_{\text{regret (a)}}) + \underbrace{(r(S_t, \bar{\boldsymbol{\mu}}_t) - r(S_t, \boldsymbol{\mu}))}_{\text{regret (b)}}] \tag{19}$$

$$\leq \mathbb{E}[\underbrace{r(S_t, \bar{\boldsymbol{\mu}}_t) - r(S_t, \boldsymbol{\mu})}_{\text{Over-estimation regret}}] \tag{20}$$

Here, Eq. (17) exploits the action reward function's monotonicity, while Eq. (18) is due to the equivalency of $r(S_t^*, \bar{\boldsymbol{\mu}}_t)$ and $\tilde{r}(\mathbb{I}_{S_t^*}, \bar{\boldsymbol{\mu}}_t)$ for any feasible action $S \in \mathcal{S}$, i.e., $r(S_t^*, \bar{\boldsymbol{\mu}}_t) = \tilde{r}(\mathbb{I}_{S_t^*}, \bar{\boldsymbol{\mu}}_t)$. Recall that $\mathbb{I}_{S_t^*} = \{z_1, z_2, \cdots, z_K\} \in \{0, 1\}^K$ denotes one feasible solution for the optimal selection status of base arms in $\mathcal{K}$ at round $t$ of the following three relaxed continuous optimization problems:

1. **Any Win Combination (AWC):**

$$\max \tilde{r}(\tilde{\boldsymbol{Z}}, \bar{\boldsymbol{\mu}}) = \left(1 - \prod_{k \in \mathcal{K}} (1 - \bar{\mu}_k \tilde{z}_k)\right)$$

$$\text{s.t.} \sum_{k \in \mathcal{K}} \tilde{z}_k \leq N,$$

$$\sum_{k \in \mathcal{K}} \underline{c}_{t,k} \tilde{z}_k \leq \rho,$$

$$0 \leq \tilde{z}_k \leq 1, \forall k \in \mathcal{K},$$

where $\tilde{r}\left(\tilde{\boldsymbol{Z}}, \bar{\boldsymbol{\mu}}\right) = \left(1 - \prod_{k \in \mathcal{K}}(1 - \bar{\mu}_k \tilde{z}_k)\right)$ is the closed form of the multi-linear extension $\tilde{r}\left(\tilde{\boldsymbol{Z}}, \bar{\boldsymbol{\mu}}\right) = \sum_{S \subseteq \mathcal{K}} \prod_{k \in S} \tilde{z}_k \prod_{k \notin S}(1 - \tilde{z}_k)$.

2. **Sum Up Combination (SUC):**

$$\max \tilde{r}(\tilde{\boldsymbol{Z}}, \bar{\boldsymbol{\mu}}) = \sum_{k \in \mathcal{K}} \bar{\mu}_k \tilde{z}_k$$

$$\text{s.t.} \sum_{k \in \mathcal{K}} \tilde{z}_k = N,$$

$$\sum_{k \in \mathcal{K}} \underline{c}_{t,k} \tilde{z}_k \leq \rho,$$

$$0 \leq \tilde{z}_k \leq 1, \forall k \in \mathcal{K}.$$

3. **All In Combination (AIC):**

$$\max \tilde{r}(\tilde{\boldsymbol{Z}}, \bar{\boldsymbol{\mu}}) = \prod_{k \in \mathcal{K}} \bar{\mu}_k^{\tilde{z}_k}$$

$$\text{s.t.} \sum_{k \in \mathcal{K}} \tilde{z}_k = N,$$

$$\sum_{k \in \mathcal{K}} \underline{c}_{t,k} \tilde{z}_k \leq \rho,$$

$$0 \leq \tilde{z}_k \leq 1, \forall k \in \mathcal{K},$$

whose optimal solution is equivalent to solving the LP program $\arg\max\{\sum_{k \in \mathcal{K}} \tilde{z}_k \ln \bar{\mu}_k : \sum_{k \in \mathcal{K}} \tilde{z}_k = N, \sum_{k \in \mathcal{K}} \underline{c}_{t,k} \tilde{z}_k \leq \rho, 0 \leq \tilde{z}_k \leq 1\}$ by taking $\ln r(S; \boldsymbol{\mu})$ as our objective.

Here, $\tilde{\boldsymbol{Z}} = \{\tilde{z}_1, \tilde{z}_2, \cdots, \tilde{z}_K\}$ represents the selected probability for each base arm, with $\tilde{z}_k \in [0, 1]$ reflecting the probability of selecting the $k$-th base arm.

Moreover, Eq. (19) is because $\mathbb{E}[\mathbb{E}_{S_t \sim \sigma(\tilde{\boldsymbol{Z}}_t)}[\tilde{r}(\mathbb{I}_{S_t}, \bar{\boldsymbol{\mu}}_t)]] = \mathbb{E}[\tilde{r}(\mathbb{I}_{S_t}, \bar{\boldsymbol{\mu}}_t)] = \mathbb{E}[r(S_t, \bar{\boldsymbol{\mu}}_t)]$ ($S_t \sim \sigma(\tilde{\boldsymbol{Z}}_t)$ denotes selecting $S_t$ from Algorithm 2 and Algorithm 3), and Eq. (20) holds because of the convex-preserving property: $\mathbb{E}[\tilde{r}(\mathbb{I}_{S_t}, \bar{\boldsymbol{\mu}}_t)] \geq \tilde{r}(\tilde{\boldsymbol{Z}}_t, \bar{\boldsymbol{\mu}}_t)$. Here we can see paralleling approaches in Kveton et al. (2015c;b); Wang & Chen (2017) that bounds the over-estimation regret $\mathbb{E}[r(S_t, \bar{\boldsymbol{\mu}}_t) - r(S_t, \boldsymbol{\mu})]$. We will follow the idea in Wang & Chen (2017) but a more simplified proof to deal with this over-estimation regret.

**Step 2: Deal with the partial observation and apply the $L$-Lipschitz condition**

Define the over-estimation regret for C2MAB-V at round $t$ as $r\bar{e}g(S_t, \bar{\boldsymbol{\mu}}_t) = r(S_t, \bar{\boldsymbol{\mu}}_t) - r(S_t, \boldsymbol{\mu})$, where $S_t$ denotes the action and $\bar{\boldsymbol{\mu}}_t$ represents the upper confidence bound (UCB) value of base arms at round $t$. The history of C2MAB-V prior to selecting action $S_t$, denoted as $\mathcal{H}_t$, encompasses the sequence of actions and observations up to round $t - 1$, expressed as $\mathcal{H}_t = ((S_1, \mathcal{F}_1, \boldsymbol{\mu}_{1,\mathcal{F}_1}, \boldsymbol{c}_{1,S_1}), \ldots, (S_{t-1}, \mathcal{F}_{t-1}, \boldsymbol{\mu}_{t-1,\mathcal{F}_{t-1}}, \boldsymbol{c}_{t-1,S_{t-1}}))$. Here, $\boldsymbol{\mu}_{t,\mathcal{F}_t} = (\boldsymbol{\mu}_{t,S_{t,1}}, ..., \boldsymbol{\mu}_{t,S_{t,F_t}})$ and $\boldsymbol{c}_{t,S_t} = (c_{t,S_{t,1}}, ..., c_{t,S_{t,|S_t|}})$ detail the partial observed rewards and full costs of LLM for the $t^{th}$ action, respectively.

Introduce $\Omega_t$ as the random seed influencing the discretization procedure $\sigma$ from Algorithm 2 and Algorithm 3 at round $t$, ensuring that $S_t = \sigma(\tilde{\boldsymbol{Z}}_t)$ is deterministic given $\Omega_t$ and $\tilde{\boldsymbol{Z}}_t$. The notation $\mathbb{E}[\cdot|\mathcal{H}_t, \Omega_t]$ specifies the conditional expectation given the historical context $\mathcal{H}_t$ and the random seed $\Omega_t$.

Our analysis proceeds under the premise that all base arms (i.e., LLM) in the selected set $S_t$ are observed, yielding the expected regret at round $t$, conditioned on history $\mathcal{H}_t$ and random seed $\Omega_t$, as follows:

$$\mathbb{E}[r\bar{e}g(S_t, \bar{\boldsymbol{\mu}}_t)|\mathcal{H}_t, \Omega_t] \tag{21}$$

$$= \mathbb{E}\left[r\bar{e}g(S_t, \bar{\boldsymbol{\mu}}_t)\mathbb{E}\left[\frac{1}{o_{t,S_t}}\mathbb{I}\{F_t = |S_t|\}|S_t\right]|\mathcal{H}_t, \Omega_t\right] \tag{22}$$

$$= \mathbb{E}\left[r\bar{e}g(S_t, \bar{\boldsymbol{\mu}}_t)\frac{1}{o_{t,S_t}}\mathbb{I}\{F_t = |S_t|\}|\mathcal{H}_t, \Omega_t\right] \tag{23}$$

$$\leq \frac{1}{o^*}\mathbb{E}[r\bar{e}g(S_t, \bar{\boldsymbol{\mu}}_t)\mathbb{I}\{F_t = |S_t|\}|\mathcal{H}_t, \Omega_t] \tag{24}$$

where Eq. (22) utilizes the fact that, with $S_t$ determined, $o_{t,S_t}$ represents the probability of observing $F_t = |S_t|$, Eq. (23) simplifies the expression by considering the conditions where $\mathcal{H}_t$ and $\Omega_t$ are fixed, thus fixing $S_t$. And the last inequality in Eq. (24) emerges from defining $o^* = \min_{t \in \mathcal{T}, S \in \mathcal{S}} o_{t,S}$ as the minimum observation probability, highlighting the analysis within the framework of partial feedback mechanisms and its implications for LLM selections.

Integrating the results from Eq. (2), Eq. (20), and Eq. (24), the expected cumulative regret $R(T)$ can be expressed and bounded as follows:

$$R(T) = \mathbb{E}\left[\sum_{t=1}^{T} \mathbb{E}[\mathbb{E}[r(S_t, \bar{\boldsymbol{\mu}}_t) - r(S_t, \boldsymbol{\mu})]|\mathcal{H}_t, \Omega_t]\right]$$

$$\leq \mathbb{E}\left[\sum_{t=1}^{T} \mathbb{E}[r\bar{e}g(S_t, \bar{\boldsymbol{\mu}}_t)|\mathcal{H}_t, \Omega_t]\right] \tag{25}$$

$$\leq \frac{1}{o^*}\mathbb{E}\left[\mathbb{E}[\sum_{t=1}^{T} r\bar{e}g(S_t, \bar{\boldsymbol{\mu}}_t)\mathbb{I}\{F_t = |S_t|\}|\mathcal{H}_t, \Omega_t]\right] \tag{26}$$

$$= \frac{1}{o^*}\mathbb{E}\left[\sum_{t=1}^{T} r(S_t, \bar{\boldsymbol{\mu}}_t) - r(S_t, \boldsymbol{\mu})\mathbb{I}\{F_t = |S_t|\}\right] \tag{27}$$

$$\leq \frac{L}{o^*}\mathbb{E}\left[\sum_{t=1}^{T}\sum_{k=1}^{F_t} |\bar{\mu}_{t,S_{t,k}} - \mu_{t,S_{t,k}}|\right] \tag{28}$$

$$\leq \frac{L}{o^*}\mathbb{E}\left[\sum_{t=1}^{T}\sum_{k=1}^{F_t} 2\sqrt{\frac{\ln(\frac{2\pi^2 Kt^3}{3\delta})}{2T_{t,\mu_k}}}\right], k \in S_t \tag{29}$$

$$= \frac{L}{o^*}\mathbb{E}\left[\sum_{k=1}^{K}\sum_{s=1}^{T_{T+1,\mu_k}} \sqrt{\frac{2\ln(\frac{2\pi^2 KT^3}{3\delta})}{s}}\right] \tag{30}$$

where Eq. (25) follows from Eq. (20), Eq. (26) is due to Eq. (24), Eq. (27) is by the tower rule of expectation ($\mathbb{E}[X] = \mathbb{E}[\mathbb{E}[X|Y]]$ for two random variables $\tilde{\boldsymbol{Z}}$ and $Y$) and the definition $r\bar{e}g(S_t, \bar{\boldsymbol{\mu}}_t)$, Eq. (28) leverages the $L$-Lipschitz continuity of $r(S; \boldsymbol{\mu})$, Eq. (29) employs the bound $0 \leq \bar{\mu}_{t,k} - \mu_{t,k} \leq 2\rho_{t,\mu_k}$ $k \in S_t$, given event $\mathcal{K}_\mu$, Eq. (30) accounts for the observation counter $T_{t,\mu_k}$ increasing by one whenever the weight of base arm $k$ is observed.

**Step 3: Derivation to get the final regret bound**

To get the final regret bound, we proceed with the following mathematical derivation:

$$R(T) \leq \frac{L}{o^*}\mathbb{E}\left[\sum_{k=1}^{K}\sum_{s=1}^{T_{T+1,\mu_k}} \sqrt{2\frac{\ln(\frac{2\pi^2 KT^3}{3\delta})}{s}}\right]$$

$$\leq \frac{L}{o^*}\mathbb{E}\left[\sum_{k=1}^{K}\int_{s=0}^{T_{T+1,\mu_k}} \sqrt{\frac{2\ln(\frac{2\pi^2 KT^3}{3\delta})}{s}}ds\right] \tag{31}$$

$$= \frac{2L}{o^*}\mathbb{E}\left[\sum_{k=1}^{K} \sqrt{T_{T+1,\mu_k} 2\ln(\frac{2\pi^2 KT^3}{3\delta})}\right]$$

$$\leq \frac{2L}{o^*}\mathbb{E}\left[\sqrt{2K\sum_{k=1}^{K} T_{T+1,\mu_k}\ln(\frac{2\pi^2 KT^3}{3\delta})}\right] \tag{32}$$

$$\leq \frac{2L}{o^*}\sqrt{2NKT\ln(\frac{2\pi^2 KT^3}{3\delta})}, \tag{33}$$

where Eq. (31) approximates the discrete sum with an integral for a smoother upper bound, Eq. (32) applies the Cauchy-Schwarz inequality to transition from the sum of square roots to the square root of a sum, and Eq. (33) recognizes that the total number of times base arms are observed is bounded by $NT$, considering $N$ as the maximum selection size, i.e., $\sum_{k=1}^{K}\sum_{k=1}^{K} T_{T+1,\mu_k} \leq NT$.

The proof concludes by acknowledging that the initial $K$ rounds contribute at most $r^*K$ to the regret, combined with the bound established in Eq. (33) and setting $\delta = 1/T$ for the regret analysis.

### D.3 COMPREHENSIVE PROOF OF THEOREM 2

**General violation proof analysis**

To prove the violation bound, we focus on the scenario where event $\mathcal{K}_c$ occurs with a probability of at least $1 - \delta/2$ (by Lemma 1), specifically:

$$\Pr\{\neg\mathcal{K}_c\} = \Pr\left\{\exists t \in \mathcal{T}, k \in \mathcal{K}, |\hat{c}_{t,k} - c_k| \geq \sqrt{\frac{\ln(\frac{2\pi^2 K t^3}{3\delta})}{2T_{t,c_k}}}\right\} \leq \frac{\delta}{2}.$$

Given the occurrence of $\mathcal{K}_c$, it follows that $0 \leq c_k - \underline{c}_{t,k} \leq 2\rho_{t,c_k}$. From Algorithm 2 and Algorithm 3, we know that $\rho \geq \sum_{k \in \mathcal{K}} \underline{c}_{t,k}\tilde{z}_{t,k} = \underline{c}_t \cdot \tilde{Z}_t$, where $\underline{c}_t := \{\underline{c}_{t,1}, \underline{c}_{t,2}, \ldots, \underline{c}_{t,K}\}$, and we also denote $c_t := \{c_{t,1}, c_{t,2}, \ldots, c_{t,K}\}$. This setup allows us to estimate the per-round violation as follows:

$$V(T) \leq \left|\frac{1}{T}\sum_{t=1}^{T}\sum_{k \in S_t} y_{t,k} - \rho\right| \tag{34}$$

$$\leq \frac{1}{T}\left|\sum_{t=1}^{T}\left(\sum_{k \in S_t} y_{t,k} - \underline{c}_t \cdot \tilde{Z}_t\right)\right| \tag{35}$$

$$\leq \frac{1}{T}\left(\underbrace{\left|\sum_{t=1}^{T}\left(\sum_{k \in S_t} y_{t,k} - \sum_{k \in S_t} c_k\right)\right|}_{\text{violation (a)}} + \underbrace{\left|\sum_{t=1}^{T}\left(\sum_{k \in S_t} c_k - \sum_{k \in S_t} \underline{c}_{t,k}\right)\right|}_{\text{violation (b)}}\right. \tag{36}$$

$$\left. + \underbrace{\left|\sum_{t=1}^{T}\left(\sum_{k \in S_t} \underline{c}_{t,k} - \underline{c}_t \cdot \tilde{Z}_t\right)\right|}_{\text{violation (c)}}\right) \tag{37}$$

where Eq. (34) originates from the violation definition, Eq. (35) leverages the inequality $\rho \geq \underline{c}_t \cdot \tilde{Z}_t$ illustrated above, and Eq. (37) utilizes the inequality $|a + b + c| \leq |a| + |b| + |c|$ for separation of terms.

Similarly, we begin with a sketch of the proof to enhance comprehension. For every round $t \in \mathcal{T}$ and LLM $k \in \mathcal{K}$, the event $|\hat{c}_{t,k} - c_k| < \sqrt{\ln\left(\frac{2\pi^2 K t^3}{3\delta}\right)/2T_{c,t,k}}$ occurs with probability at least $1 - \delta/2$, from Lemma 1, thus resulting in $c_k \geq \hat{c}_{t,k} - \rho_{c,t,k} = \underline{y}_{t,k}$ via setting $\delta = 1 - 1/T$. From Eq. (3), we have that $\rho \geq \sum_{k \in \mathcal{K}} \underline{c}_{t,k}\tilde{z}_{t,k}, \forall t \in \mathcal{T}$. Then $V(T) = \left[\frac{1}{T}\sum_{t=1}^{T}\sum_{k \in S_t} y_{t,k} - \rho\right]^+$ is bounded by:

$$V(T) \leq \frac{1}{T}\left(\underbrace{\left|\sum_{t=1}^{T}\sum_{k \in S_t}(y_{t,k} - c_k)\right|}_{\text{violation (a)}} + \underbrace{\left|\sum_{t=1}^{T}\sum_{k \in S_t}(c_k - \underline{c}_{t,k})\right|}_{\text{violation (b)}} + \underbrace{\left|\sum_{t=1}^{T}\sum_{k \in S_t}(\underline{c}_{t,k} - \underline{c}_{t,k}\tilde{z}_{t,k})\right|}_{\text{violation (c)}}\right). \tag{38}$$

By constructing martingale sequences and applying Azuma-Hoeffding inequality, we can bound violation (a) by $\left(N\sqrt{2T\ln(8T)}\right)$ with probability at least $1 - 1/4T$. Similarly, violation (c) is bounded by $\left(N\sqrt{2T\ln(8T)}\right)$ with probability at least $1 - 1/4T$. With $c_k \geq \hat{c}_{t,k} - \rho_{c,t,k} = \underline{c}_{t,k}$ satisfied at least $1 - 1/2T$ probability, violation (b) is bounded by $4\sqrt{2NKT\ln\left(\frac{2\pi^2 KT}{3}\right)}$. Utilizing the union bounds, we get the desired results in Eq. (8).

Next, we individually bound violations (a), (b), and (c) to further quantify the violation and finally use the union bounds for violation (a), (b), (c) with the initialization process that causes at most $NK$ violation to prove the desired terms in Eq. (8).

**Step 1: Bound violation (a)**

To address violation (a), we introduce a series of random variables $\{Y_1, Y_2, \ldots, Y_T\}$, constructing a martingale sequence relative to the historical data of C2MAB-V, denoted $\mathcal{H}_t$. The history $\mathcal{H}_t$ encompasses the sequence of actions and observed outcomes up to round $t - 1$, encapsulating both the actions taken and the corresponding observed rewards and full costs. Formally, $\mathcal{H}_t$ includes tuples $((S_1, \mathcal{F}_1, \boldsymbol{\mu}_{1,\mathcal{F}_1}, \boldsymbol{c}_{1,S_1}), \ldots, (S_{t-1}, \mathcal{F}_{t-1}, \boldsymbol{\mu}_{t-1,\mathcal{F}_{t-1}}, \boldsymbol{c}_{t-1,S_{t-1}}))$, where $\boldsymbol{\mu}_{t,\mathcal{F}_t} = (\boldsymbol{\mu}_{t,S_{t,1}}, \ldots, \boldsymbol{\mu}_{t,S_{t,F_t}})$ and $\boldsymbol{c}_{t,S_t} = (c_{t,S_{t,1}}, \ldots, c_{t,S_{t,|S_t|}})$, representing the observed rewards and costs, respectively, for the action set $S_t$.

The martingale is recursively defined starting with $\boldsymbol{Y}_0 = 0$, and for each round $t = 1, \ldots, T$,

$$\boldsymbol{Y}_t - \boldsymbol{Y}_{t-1} = \sum_{k \in S_t} y_{t,k} - \sum_{k \in S_t} c_k \tag{39}$$

where $\boldsymbol{\mu}_t = \boldsymbol{Y}_t - \boldsymbol{Y}_{t-1}$ denotes the change in the martingale value from round $t - 1$ to $t$. Given the construction of $\boldsymbol{\mu}_t$, we assert that $\mathbb{E}[\boldsymbol{\mu}_t | \mathcal{H}_t] = 0$ and the absolute change $|\boldsymbol{\mu}_t|$ is bounded by the maximum size of the action set, denoted as $N$, i.e., $|\boldsymbol{\mu}_t| \leq N$. Therefore, applying the Azuma-Hoeffding inequality (Lemma 6), we derive that

$$\left| \sum_{t=1}^T \left( \sum_{k \in S_t} y_{t,k} - \sum_{k \in S_t} c_k \right) \right| \leq \sqrt{2N^2 T \ln \frac{8}{\delta}} \tag{40}$$

holds with a probability of at least $1 - \delta/4$. This step bounds the expected deviation arising from the selection of actions and their associated costs.

**Step 2: Bound violation (b)**

In addressing violation (b), we focus on the scenario where event $\mathcal{K}_c$ occurs, which holds with a probability of at least $1 - \delta/2$. Under this condition, we estimate the cumulative difference between the actual costs $c_k$ and their lower confidence bounds $\underline{c}_{t,k}$ for all actions in $S_t$ across all rounds $t$. The bound is derived as follows:

$$\left| \sum_{t=1}^T \left( \sum_{k \in S_t} c_k - \sum_{i \in S_t} \underline{c}_{t,k} \right) \right| \leq \sum_{t=1}^T \sum_{k \in S_t} 2 \sqrt{\frac{\ln(\frac{2\pi^2 K t^3}{3\delta})}{2\boldsymbol{T}_{c,t,i}}} \tag{41}$$

$$= \sum_{k=1}^K \sum_{s=1}^{T_{T+1,c_k}} \sqrt{\frac{2 \ln(\frac{2\pi^2 K T^3}{3\delta})}{s}} \tag{42}$$

$$\leq \sum_{k=1}^K \int_{s=0}^{T_{T+1,c_k}} \sqrt{\frac{2 \ln(\frac{2\pi^2 K T^3}{3\delta})}{2s}} ds \tag{43}$$

$$= 2 \sum_{k=1}^K \sqrt{T_{T+1,c_k} 2 \ln(\frac{2\pi^2 K T^3}{3\delta})} \tag{44}$$

$$\leq 2 \sqrt{2K \sum_{k=1}^K T_{T+1,c_k} \ln(\frac{2\pi^2 K T^3}{3\delta})} \tag{44}$$

$$\leq 2 \sqrt{2NKT \ln(\frac{2\pi^2 K T^3}{3\delta})} \tag{45}$$

where Eq. (41) utilizes the confidence interval for cost estimation under event $\mathcal{K}_c$, i.e., $0 \leq c_k - \underline{c}_{t,k} \leq 2\rho_{t,c_k}$, Eq. (42) rearranges the summation to account for the total observations of cost $c_k$ for each action based on the fact that the counter $T_{t,c_k}$ increase by 1 if and only if base arm $k$'s cost has been observed, Eq. (43) replaces the summation with an integral to provide an upper bound, given the increasing sequence of $\frac{1}{\sqrt{s}}$, the final two inequalities, Eq. (44) and Eq. (45), apply the Cauchy-Schwarz inequality and the total number of selections to conclude the bound $\sum_{k=1}^K \sum_{k=1}^K T_{T+1,c_k} \leq NT$. This derivation quantifies the discrepancy between the actual costs and their estimated lower bounds over all rounds.

**Step 3: Bound violation (c)**

Similar to how we prove violation (a), we will define a series of random variables $\{Y_1, \ldots, Y_T\}$ to form a martingale with respect to history of C2MAB-V $\mathcal{H}_t$, defined as $\mathcal{H}_t = ((S_1, \mathcal{F}_1, \boldsymbol{\mu}_{1,\mathcal{F}_1}, \boldsymbol{c}_{1,\mathcal{F}_1}), \ldots, (S_{t-1}, \mathcal{F}_{t-1}, \boldsymbol{\mu}_{t-1,\mathcal{F}_{t-1}}, c_{t-1,\mathcal{F}_{t-1}}))$ to encompasses the sequence of actions and observations up to round $t-1$, before choosing the action $S_t$ which includes the first $t-1$ actions and first $t-1$ partial observed rewards and full costs, where $\boldsymbol{\mu}_{t,\mathcal{F}_t} = (\boldsymbol{\mu}_{t,S_{t,1}}, ..., \boldsymbol{\mu}_{t,S_{t,F_t}})$ and $\boldsymbol{c}_{t,S_t} = (c_{t,S_{t,1}}, ..., \boldsymbol{c}_{t,S_{t,F_t}})$. The martingale sequence is defined recursively starting with $\boldsymbol{Y}_0 = 0$. For each round $t = 1, \ldots, T$, the incremental change is given by:

$$\boldsymbol{Y}_t - \boldsymbol{Y}_{t-1} = \sum_{k \in S_t} \underline{c}_{t,k} - \underline{\boldsymbol{c}}_t \cdot \tilde{\boldsymbol{Z}}_t, \tag{46}$$

where $\boldsymbol{\mu}_t = \boldsymbol{Y}_t - \boldsymbol{Y}_{t-1}$ denotes the change in the martingale value from round $t-1$ to $t$. Recall that the discretization procedure $\sigma$ is denoted for Algorithm 2 and Algorithm 3 as $S_t = \sigma(\tilde{\boldsymbol{Z}}_t)$ and $N = \max_{S \in \mathcal{S}} |S|$. Given that $\mathbb{E}[\boldsymbol{\mu}_t | \mathcal{H}_t] = 0$ follows from the expectation $\mathbb{E}_{S_t \sim \sigma(\tilde{\boldsymbol{z}}_t)}[\mathbb{I}_{S_t}] = \tilde{\boldsymbol{Z}}_t$ and and the bound $|\boldsymbol{\mu}_t| \leq N$. Therefore, we can apply the Azuma-Hoeffding inequality (Lemma 6) to obtain:

$$\left| \sum_{t=1}^{T} \left( \sum_{k \in S_t} \underline{c}_{t,k} - \underline{\boldsymbol{c}}_t \cdot \tilde{\boldsymbol{Z}}_t \right) \right| \leq \sqrt{2K^2 T \ln \frac{8}{\delta}} \tag{47}$$

with probability at least $1 - \delta/4$.

Employing the union bound across violations (a), (b), and (c), and accounting for the initial setup contributing a maximum of $KN$ to the violation, we affirm the desired bounds as specified in Eq. (8) with a collective probability of at least $1 - \delta$. The proof concludes by setting $\delta = 1/T$ for the violation analysis.

### D.4 INSTANCE-DEPENDENT REGRET BOUND

Our instance-independent regret reduction holds in expectation, where the action-dependent term is effectively smoothed out by averaging. However, we now require a regret reduction that holds even without relying on this expectation. To achieve this, we introduce a new condition and derive an instance-dependent bound accordingly. Before proceeding, we define several gap-related terms. Consider a performance distribution $P_{\boldsymbol{\mu}}$ and its expectation performance vector $\boldsymbol{\mu}$. For each action $S$, let the gap $\Delta_S = \max\{0, \alpha r(S^*, \boldsymbol{\mu}) - r(S, \boldsymbol{\mu})\}$. For each arm $k \in \mathcal{K}$, let $\Delta_{\min}^k = \min_{S \in \mathcal{S}: \Delta_S > 0, k \in S} \Delta_S$ and we define $\Delta_{\min} = \min_{k \in \mathcal{K}} \Delta_{\min}$.

**Condition 1** (Sampling Quality Condition). *For any relaxed solution $\tilde{\boldsymbol{Z}} \in \mathbb{R}^K$ of the LLM combination and any performance vector $\boldsymbol{\mu}$, we say that the sampling quality condition is satisfied if, for any sampled action $S \sim \sigma(\tilde{\boldsymbol{Z}})$, the inequality $r(S, \boldsymbol{\mu}) + \gamma \geq \tilde{r}(\tilde{\boldsymbol{Z}}, \boldsymbol{\mu})$, where $\gamma \leq \Delta_{\min}$.*

Intuitively, this condition means that the reward of any sampled action is no worse than the relaxed reward by at most $\gamma$, which places a lower bound on the quality of the sampling procedure in each round, rather than just in expectation.

Instead of considering the instantaneous regret $\bar{r}(S_t, \boldsymbol{\mu}) = \alpha f(S^*, \boldsymbol{\mu}) - f(S_t, \boldsymbol{\mu}) = \Delta_{S_t}$, we use a surrogate regret $\Delta'_{S_t}$ defined as follows: $\Delta'_{S_t} = \Delta_{S_t} + r(S_t, \bar{\boldsymbol{\mu}}_t) - \tilde{r}(\tilde{\boldsymbol{Z}}_t, \bar{\boldsymbol{\mu}}_t) \geq \Delta_{S_t} - \gamma \geq 0$. On one hand, using the relation $\mathbb{E}[\tilde{r}(\mathbb{I}_{S_t}, \boldsymbol{\mu})] \geq \tilde{r}(\tilde{\boldsymbol{Z}}_t, \boldsymbol{\mu})$ from Appendix C.2, we have

$$\mathbb{E}[r\bar{e}g(S_t, \boldsymbol{\mu}_t)] \leq \mathbb{E}[\Delta'_{S_t}]. \tag{48}$$

Thus, if we can bound the right-hand side, we also bound the left-hand side. On the other hand, using the similar poof of Step 1 in Appendix D.2, which reduces to over-estimation regret, we obtain $\tilde{r}(\tilde{\boldsymbol{Z}}_t, \bar{\boldsymbol{\mu}}_t) \geq \alpha r(S^*, \boldsymbol{\mu})$. Consequently, we have

$$\Delta'_{S_t} \leq r(S_t, \bar{\boldsymbol{\mu}}_t) - r(S_t, \boldsymbol{\mu}) \leq \sum_{k \in S_t} 2L \sqrt{\frac{\ln(\frac{2\pi^2 K t^3}{3\delta})}{2T_{t,\mu_k}}}. \tag{49}$$

Importantly, this bound holds both with and without taking the expectation, meaning it retains its instance-dependence even without smoothing.

We denote the history of C2MAB-V prior to selecting action $S_t$ by $\mathcal{H}_t$, which includes all actions and observations up to round $t-1$. Specifically, $\mathcal{H}_t = ((S_1, \mathcal{F}_1, \boldsymbol{\mu}_{1,\mathcal{F}_1}, \boldsymbol{c}_{1,S_1}), \ldots, (S_{t-1}, \mathcal{F}_{t-1}, \boldsymbol{\mu}_{t-1,\mathcal{F}_{t-1}}, \boldsymbol{c}_{t-1,S_{t-1}}))$, where $\boldsymbol{\mu}_{t,\mathcal{F}_t} = (\mu_{t,S_{t,1}}, \ldots, \mu_{t,S_{t,F_t}})$ and $\boldsymbol{c}_{t,S_t} = (c_{t,S_{t,1}}, \ldots, c_{t,S_{t,|S_t|}})$ represent the partially observed rewards and full costs of the LLM for the $t^{th}$ action, respectively. Let $\Omega_t$ represent the random seed that affects the discretization procedure $\sigma$ used in Algorithm 2 and Algorithm 3 at round $t$. The notation $\mathbb{E}[\cdot|\mathcal{H}_t, \Omega_t]$ specifies the conditional expectation given the historical context $\mathcal{H}_t$ and the random seed $\Omega_t$. Define the set $\mathcal{S}_t$ as

$$\mathcal{S}_t = \{\Delta'_{S_t} \leq \sum_{k \in S_t} 2L\sqrt{\frac{\ln(\frac{2\pi^2 K t^3}{3\delta})}{2T_{t,\mu_k}}}, \Delta'_{S_t} > 0, F_t = |S_t|\}.$$

Next, we use the Lipschitz condition along with the partial observation probability to bound the regret:

$$\bar{reg}(T) = \mathbb{E}\left[\sum_{t=1}^{T} \mathbb{E}[\bar{reg}(S_t, \boldsymbol{\mu}_t)|\mathcal{H}_t, \Omega_t]\right]$$

$$\leq \mathbb{E}\left[\sum_{t=1}^{T} \mathbb{E}[\Delta'_{S_t}|\mathcal{H}_t, \Omega_t]\right] \tag{50}$$

$$\leq \frac{1}{o^*}\mathbb{E}\left[\mathbb{E}[\sum_{t=1}^{T} \Delta'_{S_t}\mathbb{I}\{F_t = |S_t|\}|\mathcal{H}_t, \Omega_t]\right] \tag{51}$$

$$= \frac{1}{o^*}\mathbb{E}\left[\sum_{t=1}^{T} \mathbb{E}\left[\Delta'_{S_t}\mathbb{I}\{F_t = |S_t|\}|\mathcal{H}_t\right]\right]$$

$$\leq \frac{1}{o^*}\mathbb{E}\left[\sum_{t=1}^{T} \mathbb{E}\left[\Delta'_{S_t}\mathbb{I}\{\mathcal{S}_t\}|\mathcal{H}_t\right]\right] \tag{52}$$

where Eq. (50) follows from Eq. (48), Eq. (51) follows from the partial observation argument in Step 2 of Appendix D.2, Eq. (52) follows from the application of Eq. (49).

Finally, we can find similar counterparts of $\mathcal{S}_t$ and Eq. (52) in Appendix A.3 of Kveton et al. (2015c), leading to an instance-dependent regret bound with $\delta = 1/T$:

$$R(T) \leq O\left(\frac{L^2 N}{o^*} \sum_{k \in \mathcal{K}} \frac{1}{\Delta_{\min}^k - \gamma} \log T + r^*(K+1)\right) \tag{53}$$

## D.5 Technical Inequalities

We finally introduce some well-known fundamental results without proofs that serve as pivotal tools in probability theory and statistical inference, particularly in the realms of concentration inequalities and martingale sequences.

**Lemma 4** (Subgaussian random variables). *Suppose that random variables $X$ is $\sigma$-subgaussian, $X_1$ and $X_2$ are independent and $\sigma_1$ and $\sigma_2$-subgaussian, respectively, then*

*1. For any $\varepsilon > 0$, $\Pr[X \geq \varepsilon] \leq \exp\left(-\frac{\varepsilon^2}{2\sigma^2}\right)$.*

*2. $X_1 + X_2$ is $\sqrt{\sigma_1^2 + \sigma_2^2}$-subgaussian.*

**Lemma 5** (Chernoff-Hoeffding inequality Dubhashi & Panconesi (2009).). *Let $Y_1, \ldots, Y_n$ be independent and identically distributed (i.e., i.i.d.) random variables with common support $[0,1]$ and mean $\mu$. Let $Z = Y_1 + \ldots + Y_n$. Then for any $\epsilon > 0$,*

$$\Pr\{|Z - n\mu| \geq \epsilon\} \leq 2e^{-\frac{2\epsilon^2}{n}}.$$

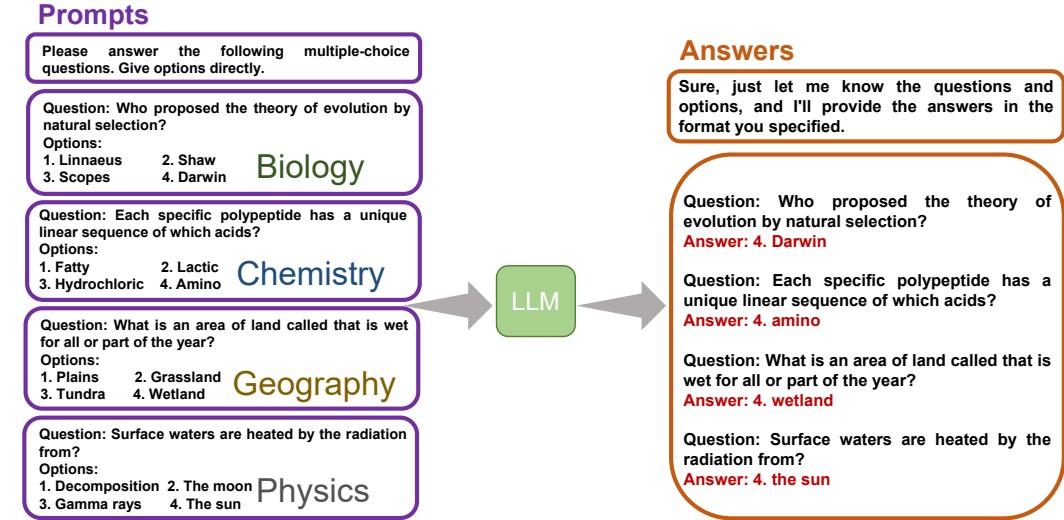

Figure 5: Sample conversation with LLM on biology, chemistry, geography, and physics.

**Lemma 6** (Azuma-Hoeffding inequality Azuma (1967).). *Let* $\{z_k : i = 0, 1, 2, \ldots, n\}$ *be a martingale and* $|Y_k - Y_{k-1}| \le c_k$ *for* $k \in [n]$ *almost surely, then for any* $\epsilon > 0$

$$\Pr\{|Y_n - Y_0| \ge \epsilon\} \le 2e^{-\frac{\epsilon^2}{2\sum_{i=1}^n c_k^2}}.$$

# E   MORE EXPERIMENTS

## E.1   EXPERIMENTAL DETAILS

Consistent with Section 6, we conduct all experiments on a device equipped with an Intel Core i5-13600KF CPU @ 3.50GHz and 32 GB of memory. Gurobi, a powerful mathematical optimization solver that supports various programming models, is utilized to solve relaxed constraint optimization problems, employing version 11.0.1 Gurobi-Optimization (2024). Fig. 5 depicts a sample interaction involving large language models (LLMs) across various scientific disciplines—Biology, Chemistry, Geography, and Physics—utilizing the SciQ dataset Welbl et al. (2017). Table 4 lists the LLMs used in the experiments (refer to Section 6), detailing each model's name with parameters, cost per 1,000 tokens in USD, and size in gigabytes. The models range from smaller configurations, such as ChatGLM2-6B-32K, to larger ones like Llama 2-70B. Considering the variability in generating answers when using LLMs, as well as potential non-responses, we specify reward allocations based on different LLM return scenarios in addition to comparisons with the SciQ dataset's labels. If the question is answered accurately and adheres to the prescribed prompt format, the reward is set at 0.5. In the event of an incorrect response, no reward is granted. However, if factors such as network congestion or potential issues with the LLM result in an empty response, a reward of 0.1 is allocated. Should the response adhere to the format guidelines yet fail to meet the specified requirements, a reward of 0.3 is assigned. We simulate user feedback for each round using the public SciQ dataset to function as a binary feedback Shuster et al. (2022), and our algorithm learns the performance of LLMs based on this feedback. Our entire framework is built on the popular bandit framework Lattimore & Szepesvári (2020), which continuously updates and adjusts decisions based on a steady stream of data (i.e., user feedback) to address the classic exploration-exploitation dilemma regarding LLM performance and cost.

## E.2   RESPECTIVE RESULTS FOR REWARD AND VIOLATION

**Per-round Reward and Violation.** Fig. 6 and Fig. 7 display the experimental rewards and violations for three different task types. In Fig. 6, *C2MAB-V* consistently outperforms or matches the $\epsilon$-*Greedy* baseline in terms of rewards per round. The parameters on $\alpha_\mu$ and $\alpha_c$ used are (0.3, 0.05), (1,

Table 4: List of utilized large language models in Section 6, associated cost per usage.

| LLM_ID | Model Name (with Parameters) | Cost (USD/1k tokens) | Size (GB) |
|--------|------------------------------|----------------------|-----------|
| 1 | **ChatGLM2-6B-32K** OpenAI (2023) | 0.005 | 12.5 |
| 2 | **ChatGPT-3.5** OpenAI (2023) | 0.02 | / |
| 3 | **Claude 2** Claude (2023) | 0.08 | / |
| 4 | **ERNIE 3.5-8K** Baidu (2024) | 0.015 | / |
| 5 | **Llama 2-7B** Ollama (2023) | 0.005 | 12.6 |
| 6 | **Llama 2-13B** Ollama (2023) | 0.008 | 24.3 |
| 7 | **Llama 2-70B** Ollama (2023) | 0.05 | 128.3 |
| 8 | **Mixtral-8x7B-Instruct** Mistral-AI (2024) | 0.05 | 93.37 |
| 9 | **ChatGPT-4** OpenAI (2023) | 0.12 | / |

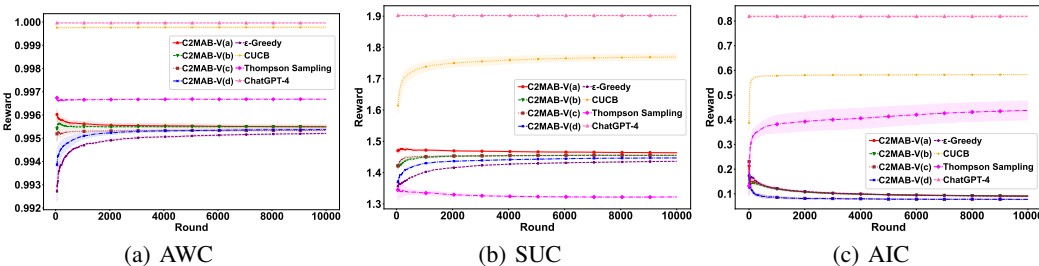

(a) AWC        (b) SUC        (c) AIC

Figure 6: Reward of three task types with nine different LLMs.

0.05), (0.3, 0.01), and (1, 0.01), respectively denoted as (a), (b), (c), and (d). Adjusting the values of $\alpha_\mu$ and $\alpha_c$ consistently results in higher rewards for *C2MAB-V* settings. These settings also significantly reduce violations per round compared to the $\epsilon$-*Greedy* approach. Although the *CUCB* algorithm achieves the highest per-round rewards, it does so completely ignoring cost constraints, leading to a per-round violation rate at least four times higher than that of *C2MAB-V*, as shown in Fig. 7. This figure also indicates that *C2MAB-V*'s violations are lower than those of the $\epsilon$-*Greedy* baseline, highlighting its ability to effectively manage constraints. Additionally, since Thompson Sampling relies on Bayesian updates, which can be highly variable, this method may lack stability and potentially cause severe oscillations, as observed in scenarios like the AIC task.

Additionally, as shown in Fig. 6(c), we can observe a convergence trend in the rewards of *C2MAB-V*. This trend suggests that the initial actions selected might have had higher rewards but also higher violations. Over time, the cost-aware online learning strategy adjusts by slightly reducing rewards to achieve greater cost savings, demonstrating its effectiveness in managing long-term constraints.

**Varying Budget Threshold.** We then examine the influence of the budget threshold $\rho$. As depicted in Fig. 8, the optimal action of combinatorial LLMs varies with the change in the budget threshold $\rho$. This variation in both arm and action selection space results in fluctuations in reward and violation. Furthermore, it is evident that across different values of $\rho$, our proposed *C2MAB-V* with $\alpha_\mu = 1, \alpha_c = 0.01$ consistently outperforms the benchmarks *CUCB* and $\epsilon$-*Greedy* by at least 356.0% and 55.7% improvment, highlighting the robustness of our algorithm.

**Performance-Driven Scenarios.** Our study explores a range of trade-offs, extending beyond the typical exploration-exploitation dichotomy to include cost and reward considerations. Given the anticipated reduction in costs over time with the development of LLMs, our analysis focuses on scenarios that have a generous budget and prioritize performance. In such performance-driven scenarios, our theoretical analysis suggests setting the cost exploration parameter $\alpha_c$ higher to better emphasize performance outcomes, which means exploring enough to have good empirical performance. Consequently, we adjust both $\alpha_\mu$ and $\alpha_c$ as follows: Performance-driven1 with $(\alpha_\mu, \alpha_c) = (0.3, 1)$, Performance-driven2 with $(\alpha_\mu, \alpha_c) = (1, 1)$, Cost-driven1 with $(\alpha_\mu, \alpha_c) = (0.3, 0.01)$, and Cost-driven2 with $(\alpha_\mu, \alpha_c) = (1, 0.01)$. As shown in Fig. 9, the performance-driven types of *C2MAB-V* yield higher rewards, whereas the cost-driven types exhibit fewer cost violations. The preferable approach varies, depending on the specific needs and constraints of the task at hand.

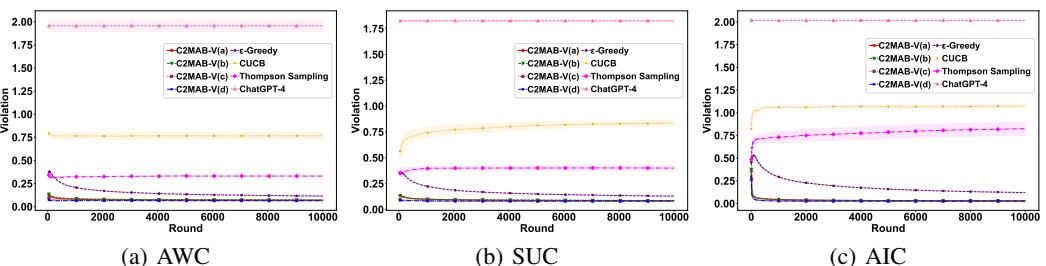

Figure 7: Violation of three task types with nine different LLMs.

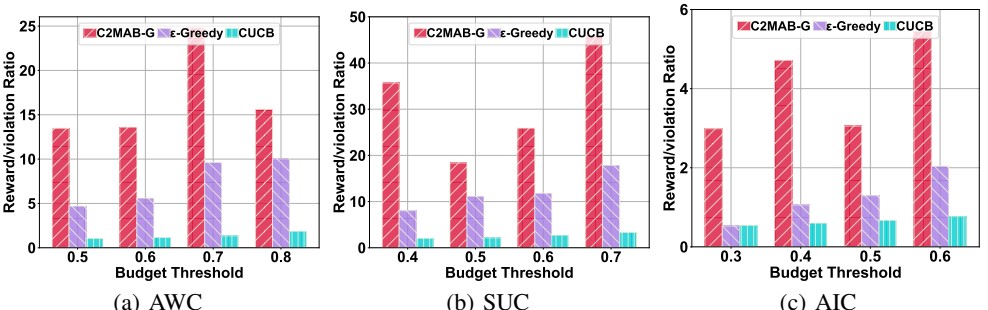

Figure 8: Reward/violation ratio for three task types across different budget thresholds $\rho$.

### E.3 IN-DEPTH ANALYSIS OF EXPLORATORY RESULTS

**Impacts of Maximum Number of LLMs.** To investigate how the maximum number of selectable LLMs, denoted as $N$, impacts performance, we present the reward/violation ratio for different $N$ values after $T = 3,000$ rounds in Fig. 10. Recognizing that different task types should not adhere to a universal budget threshold, we specifically use AWC as an example to underscore the influence of $N$, while keeping the budget threshold constant. It is important to note that as $N$ changes, the optimal combination of LLMs, i.e., the optimal action, also changes. Under the same budget threshold $\rho$, increasing $N$ expands the selection space, adding complexity and thus affecting the efficiency of all algorithms. However, we observe that with an increase in $N$, *C2MAB-V* consistently outperforms both the *CUCB* and $\epsilon$-*Greedy* baselines.

**Comparison of Computational Efficiency on Relaxation.** *C2MAB-V-Direct* is an adjusted version that finds the best feasible solution by directly solving the discrete constrained optimization Eq. (54) via enumeration, rather than using relaxation techniques:

$$\begin{cases} \max 1 - \prod_{k\in\mathcal{K}}\left(1 - \bar{\mu}_k z_k\right), & \sum_{k\in\mathcal{K}} \underline{c}_{t,k} z_k \leq \rho, z_k \in \{0,1\}, \forall k \in \mathcal{K}, \\ \max \sum_{k\in\mathcal{K}} \bar{\mu}_k z_k, & \sum_{k\in\mathcal{K}} \underline{c}_{t,k} z_k \leq \rho, z_k \in \{0,1\}, \forall k \in \mathcal{K}, \\ \max \prod_{k\in\mathcal{K}} \bar{\mu}_k^{z_k}, & \sum_{k\in\mathcal{K}} \underline{c}_{t,k} z_k \leq \rho, z_k \in \{0,1\}, \forall k \in \mathcal{K}. \end{cases} \quad (54)$$

The objective of this study is to compare the computational efficiency between *C2MAB-V*, and *C2MAB-V-Direct*, which directly solves the constrained integer optimization problem without this approach. As illustrated in Fig. 11, using the AwC reward function as an example, the comparison of reward and violation shows that *C2MAB-V-Direct* almost completely avoids violations, while the reward for *C2MAB-V* at four different parameter pair settings $(\alpha_\mu, \alpha_c)$ $((0.3, 0.05) = (a), (1, 0.05) = (b), (0.3, 0.01) = (c), \text{ and } (1, 0.01) = (d))$ is higher than that of *C2MAB-V-Direct*.

Table 5: Comparison of runtime between *C2MAB-V* and *C2MAB-V-Direct* methods.

| **Runtime** (s) | **C2MAB-V** | **C2MAB-V-Direct** |
|---|---|---|
| **AWC** | 221.78 | 1464.33 |
| **SUC** | 267.20 | 16890.91 |
| **AIC** | 283.06 | 17320.47 |

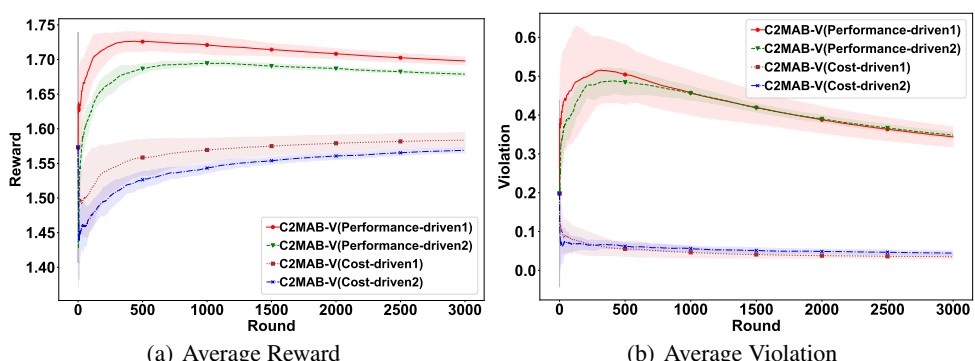

Figure 9: Analysis of reward and violation metrics across two task-driven variants of *C2MAB-V*.

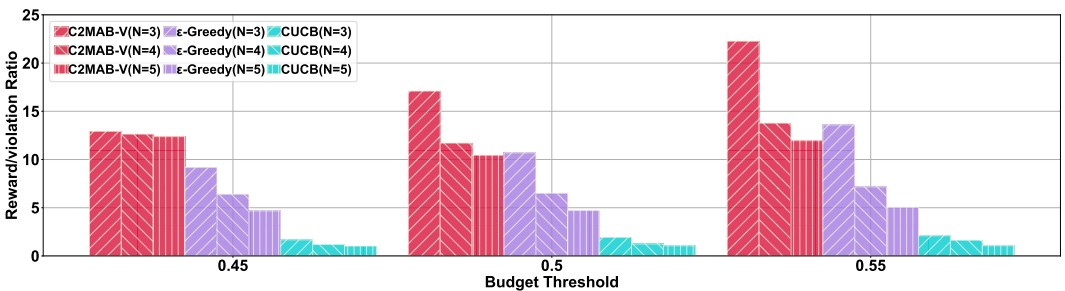

Figure 10: Impact of varying the maximum number $N$ on performance metrics over 3000 rounds.

Next, we compare the runtime of the two methods. Due to response latency fluctuations when directly invoking the LLM, which may be influenced by network conditions or query tokens (with observed delays ranging from 1 ms to 4 s), we employ a synthetic setting to focus solely on the comparative analysis of algorithmic runtime. The specifics are as follows: For each arm $k$, we independently simulate $\mu_i$ and $c_k$ from a uniform distribution $U[0, 1]$. All tasks are recorded over 10,000 rounds. Regarding parameter settings: in the AWC reward, $K = 16$, $N = 8$, and $\rho = 2.5$; in the SUC reward, $K = 25$, $N = 8$, and $\rho = 1.4$; and in the AIC reward, $K = 25$, $N = 8$, and $\rho = 1.6$.

As shown in Table 5, the computational efficiency of *C2MAB-V-Direct* is significantly lower than that of *C2MAB-V*. Specifically, *C2MAB-V* is at least 6 times faster across three different reward scenarios. Due to our relaxation and discretization design, the time complexity of *C2MAB-V* is polynomial, whereas *C2MAB-V-Direct* likely requires exponential time for the constrained integer problem, particularly when the search space is large. Nevertheless, for scenarios requiring stricter adherence to constraints, *C2MAB-V-Direct* remains a viable option. In summary, in practical applications, *C2MAB-V* and *C2MAB-V-Direct* each serve distinct purposes, offering targeted choices between computational efficiency and strict constraint compliance.

**Enhancing LLM Selection: From Two-Tier to Multi-Tier.** We further demonstrate the benefits of transitioning from a traditional "two-tier" to a more complex "multi-tier" LLM selection strategy in online settings. In the simpler two-tier system, selections are limited to just two levels of LLM engagement, which might not adequately address the diverse range of queries typically encountered in dynamic environments. By adopting a multi-tier strategy, our system can more precisely match the complexity and specificity of incoming queries with an appropriate level of computational resources and LLM expertise. The ratio comparison is not displayed in Fig. 12, as the denominator may be zero. Fig. 12 underscores the necessity and effectiveness of our advanced multi-tier strategy in enhancing response quality and system adaptability in real-time online interactions.

**Necessity of Online Learning for LLM Selection.** As discussed in Section 6, our approach emphasizes online learning methodologies that incorporate continuous feedback. To highlight the significance of ongoing online adjustments in LLM selection, we present a comparison. Initially, we pre-learned a fixed combination of multiple LLMs offline, which was then utilized to manage online queries. As illustrated in Fig. 13, our feedback-driven adjustments significantly enhance

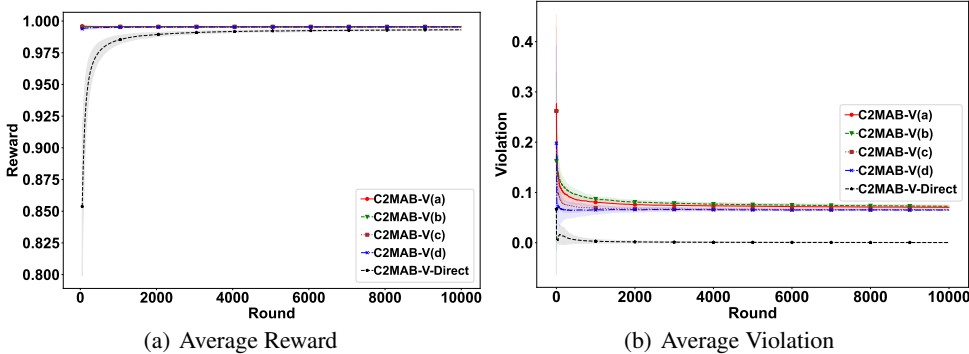

(a) Average Reward        (b) Average Violation

Figure 11: Comparison of reward and violation between *C2MAB-V* and *C2MAB-V-Direct*.

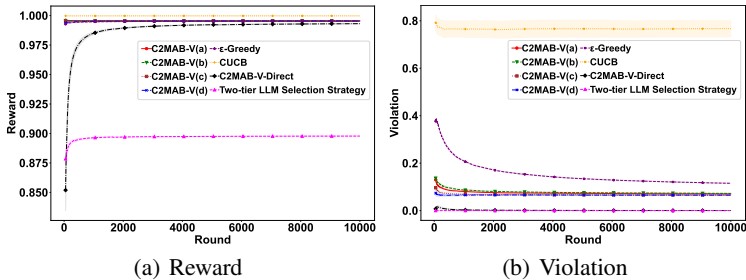

(a) Reward        (b) Violation

Figure 12: Comparison with a "two-tier LLM selection strategy", which only considers one larger and one smaller LLM, against a multi-tier approach under the AWC task type.

the effectiveness of pre-set LLM selections. By comparing our method with purely offline-learned, cascaded combinations of multi-LLMs for the AWC task type, we demonstrate the essential role of continuous online adaptations in optimizing LLM selection for real-time queries.

**Exploring Asynchronous Local-Cloud Architecture.** Recall the flow of the *C2MAB-V* algorithm: Based on our online learning-based multi-LLM selection algorithm *C2MAB-V*, multiple LLMs are coordinated and selected through a scheduling cloud. When new user data feedback is received, the local server adjusts the performance evaluation of the corresponding LLMs and notifies the scheduling cloud to update the multi-LLM selection strategy.

In the main text, we primarily demonstrate a synchronized local-cloud setting. However, in practice, the local server may not receive user feedback every round, and local-cloud communication may not be entirely synchronous. Therefore, we explore an asynchronous local-cloud architecture with a batch size. Specifically, when the batch size $B$ is reached, the local server sends new relaxed continuous data to the scheduling cloud after storing $B$ pieces of user feedback. This prompts the cloud to re-coordinate multiple LLMs and adjust the multi-model selection strategy. Until then, the previous multi-LLM selection strategy remains in use. We investigate the changes in reward and violation for batch sizes of 10, 50, 100, and 200. As shown in Fig. 14, the largest batch size of 200 exhibits relatively lower rewards and higher violations, which is intuitive. However, there is no significant difference in the reward and violation values of *C2MAB-V* under different batch sizes. This indicates that the choice of batch size may not critically impact the overall performance within the tested range for our proposed *C2MAB-V* algorithm.

### E.4 EXPANDED EVALUATION ACROSS DIVERSE DATASETS

We extended our empirical evaluation to include multiple publicly available datasets, incorporating the newly released LLaMA 3 model. The parameter $K$ is fixed at 4, with $\rho$ consistently set to 0.4. Table 6 compares the performance of various algorithms on three datasets—PIQA, OpenBookQA, and MMLU—using the AWC reward model. Detailed descriptions of these datasets are provided in Tables 7, 8, and 9, highlighting their focus on physical commonsense reasoning, multi-step reasoning

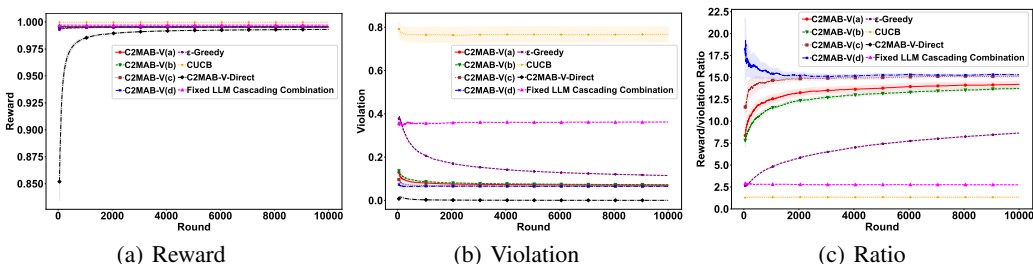

Figure 13: Comparison with "offline-learned fixed sets of multi-LLM cascaded combinations" for "online queries" under the AWC task type.

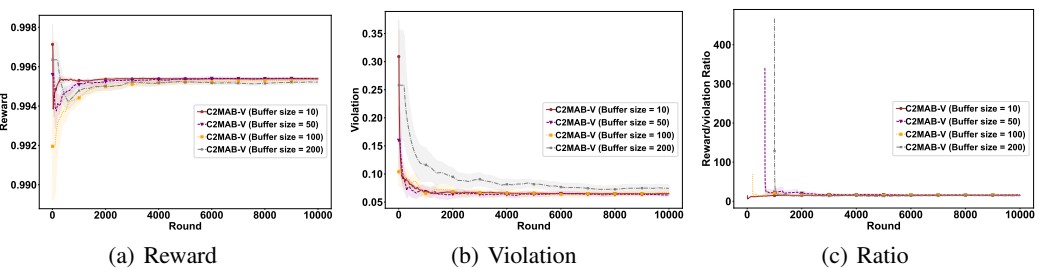

Figure 14: Comparison on asynchronous local-cloud architecture.

with commonsense knowledge, and broad domain knowledge assessment, respectively. The evaluation uses three key metrics: Reward, Violation, and Ratio. Notably, the proposed C2MAB-V algorithm outperforms others across all datasets, achieving the highest ratio and demonstrating a superior balance between performance and cost constraints. Table 10 further analyzes algorithm performance across distinct task types within the MMLU dataset: AWC, AIC, and SUC. Unlike the cross-dataset comparison, this analysis underscores task-specific variations and adaptability. C2MAB-V not only maintains the highest ratio but also exhibits consistent performance across diverse tasks, reinforcing its robustness and versatility.

# F OTHER APPLICATION EXAMPLES OF THE FRAMEWORK

In this section, we explore three representative applications that fit within our framework and summarize the results in Table 11, providing comparisons with existing works.

**Dynamic Assortment**: In this problem Sankararaman & Slivkins (2018), an agent selects $N$ products from $K$ available options over $T$ rounds to maximize total sales, subject to a cost constraint. Each product $k$ has an unknown probability $\mu_i$ of being purchased and an expected resource consumption $c_k$. The average resource consumption over $T$ rounds must remain below a threshold $\rho$. For this application, the goal is to maximize the total reward, making it suitable for the SUC reward model.

**Network Routing**: In this scenario Kveton et al. (2015b), an agent selects a path in a network with $K$ routers to maximize the probability of successful packet delivery. Each router $k$ has an unknown uptime probability $\mu_k$ and incurs a random cost with an expected value $c_k$. The agent observes whether the packet is delivered and the costs of the chosen routers, aiming to keep the average cost below $\rho$ over $T$ rounds. For this application, if any router in the selected path fails, the entire connection fails, making it suitable for the AIC reward model.

**Movie Recommendations**: In this setting Kveton et al. (2015a), an agent (e.g., Netflix) recommends a list of up to $N$ movies from $K$ options over $T$ rounds. Each movie $k$ has an unknown probability $\mu_k$ of being found attractive and incurs a random royalty cost with an expected value $c_k$. Users sequentially scan the list and click on the first attractive movie, which generates a reward. The agent aims to maximize the click probability while keeping the average cost below $\rho$. For this application, users stop browsing as soon as they find a satisfactory movie, making it suitable for the AWC reward model.

Table 6: **Comparison on different datasets on AWC reward model**.

| Dataset | PIQA Bisk et al. (2020) | | | OpenBookQA Mihaylov et al. (2018) | | | MMLU | | |
|---|---|---|---|---|---|---|---|---|---|
| | Reward | Violation | Ratio | Reward | Violation | Ratio | Reward | Violation | Ratio |
| $\epsilon$-Greedy | 0.9878 | 0.040 | 24.60 | 0.9977 | 0.041 | 24.33 | 0.9926 | 0.040 | 24.71 |
| Thompson Sampling | 0.9696 | 0.2083 | 5.151 | 0.9917 | 0.1961 | 5.697 | 0.9681 | 0.1737 | 6.104 |
| CUCB | 0.9804 | 0.2554 | 3.839 | 0.9949 | 0.2566 | 3.877 | 0.9739 | 0.2566 | 3.796 |
| **C2MAB-V** | 0.9960 | 0.0140 | **75.00** | 0.9999 | 0.0096 | **104.22** | 0.9980 | 0.0122 | **82.09** |

Table 7: **Description of PIQA** Bisk et al. (2020).

Physical commonsense knowledge is a major challenge on the road to true AI-completeness, including robots that interact with the world and understand natural language. PIQA focuses on everyday situations with a preference for atypical solutions. The dataset is inspired by instructables.com, which provides users with instructions on how to build, craft, bake, or manipulate objects using everyday materials.

Table 8: **Description of OpenBookQA** Mihaylov et al. (2018).

OpenBookQA aims to promote research in advanced question-answering, probing a deeper understanding of both the topic (with salient facts summarized as an open book, also provided with the dataset) and the language it is expressed in. In particular, it contains questions that require multi-step reasoning, use of additional common and commonsense knowledge, and rich text comprehension. OpenBookQA is a new kind of question-answering dataset modeled after open book exams for assessing human understanding of a subject.

Table 9: **Description of MMLU** Hendrycks et al. (2021).

This is a massive multitask test consisting of multiple-choice questions from various branches of knowledge. The test spans subjects in the humanities, social sciences, hard sciences, and other areas that are important for some people to learn. This covers 57 tasks including elementary mathematics, US history, computer science, law, and more. To attain high accuracy on this test, models must possess extensive world knowledge and problem solving ability.

Table 10: **Different task types on the MMLU dataset** Hendrycks et al. (2021).

| Task | AWC | | | AIC | | | SUC | | |
|---|---|---|---|---|---|---|---|---|---|
| | Reward | Violation | Ratio | Reward | Violation | Ratio | Reward | Violation | Ratio |
| $\epsilon$-Greedy | 0.9926 | 0.040 | 24.71 | 0.2242 | 0.047 | 4.748 | 1.435 | 0.0475 | 30.22 |
| Thompson Sampling | 0.9681 | 0.1737 | 6.104 | 0.3322 | 0.2346 | 1.416 | 0.9158 | 0.0186 | 60.82 |
| CUCB | 0.9739 | 0.2566 | 3.796 | 0.3297 | 0.1792 | 10.89 | 1.546 | 0.2709 | 5.811 |
| **C2MAB-V** | 0.9980 | 0.0122 | **82.09** | 0.3189 | 0.01347 | **24.13** | 1.525 | 0.01260 | **121.9** |

Table 11: Summary of other applications examples with reward Models and approximation guarantees.

| Application | Reward Function | Offline Oracle | Approximation |
|---|---|---|---|
| Dynamic Assortments | SUC | 0-1 Knapsack Chan (2018) | $\alpha = 1 - 1/K$ |
| Dynamic Assortments | SUC | Our Method | $\alpha = 1$ |
| Network Routing | AIC | 0-1 Knapsack Chan (2018) | $\alpha = 1 - 1/K$ |
| Network Routing | AIC | Our Method | $\alpha = 1$ |
| Movie Recommendations | AWC | Submodular Maximization Takemori et al. (2020) | $\alpha = \frac{1}{4(1+\varepsilon)}, \varepsilon > 0$ |
| Movie Recommendations | AWC | Our Method | $\alpha = 1 - 1/e$ |

