# OpenReview forum: "Cost-Effective Online Multi-LLM Selection with Versatile Reward Models"
_ICLR.cc/2025/Conference — Submitted to ICLR 2025_

### Official Review · Reviewer_XJC9 · 2024-11-01

**Soundness:** 2
**Presentation:** 2
**Contribution:** 2
**Rating:** 5
**Confidence:** 4

**Summary:**

This paper investigates the problem of the selection of multiple large language models (LLMs) with versatile reward models in an online manner. A framework named C2MAB-V, i.e., a cost-effective combinatorial online model with a versatile reward structure, has been proposed to select multiple LLMs based on specific task and requirements while adhering to the budget constraint. The performance of C2MAB-V has been both analyzed theoretically and evaluated via experiments. On the theoretical side, both the regret and violation bounds are provided. On the numerical side, the performance of C2MAB-V has been evaluated with nine LLMs under various settings.

**Strengths:**

- This paper studies a timely and interesting problem in the context of LLMs, in particular considering the selection of multiple LLMs with versatile reward models and in an online manner
- A new framework named C2MAB-V, i.e., a cost-effective combinatorial online model with a versatile reward structure, has been proposed to select multiple LLMs based on specific task and requirements while adhering to the budget constraint.
- Both the regret and violation bounds for C2MAB-V have been provided.
- The performance of C2MAB-V is further evaluated via extensive experiments, e.g., including nine LLMs under various settings.

**Weaknesses:**

- Despite the studied problem is timely and interesting, the intrinsic nature of the problem is not quite new. In other words, it is still a combinatorial online optimization problem but simply framed in a new LLM setting.
- Likewise, the proposed online algorithm C2MAB-V is not much in difference with the exiting methods as mentioned in the related works.
- The paper is very dense and sometimes hard to understand. For instance, it introduces three reward models, i.e., AWC, SUC, AIC. What are the fundamental differences between these models? What are the impacts on the algorithm designs and analysis? It is not quite clear and the analysis seems to only focus on one model, and not sure if you need to present all of them in the main paper. For instance, it may be better to just present AWC which has the regret analysis in Section 5, and then has a new section just to discuss the extension to SUC and AIC models with numerical evaluations.

Minor:
- This paper is framed very densely and many spaces are squeezed. For instance, there is almost no space between lines 243 and 244.

**Questions:**

- In section 3, a subset $S_t$ is selected from the available LLMs set $\mathcal{K}$. This is termed as an "action" and satisfies $|S_t|\leq K$. Right after that, $N=\max_{S\in\mathcal{S}}|S|$ is defined as the maximum number of LLMs that can be simultaneously active. What are the differences or the relations between $K$ and $N$?
- This paper introduces three reward models, i.e., AWC, SUC, AIC. What are the fundamental differences between these models? What are the impacts on the algorithm designs and analysis? The performance analysis in Section 5 is only for AWC. What about the performance guarantees under SUC and AIC. If there is no such results, what's the point to present these models in Sections 3 and 4, which make the paper very dense and hard to follow.
- For a budget constraint $\rho$, why does the definition of $V(T)$ require a max operator? In many constrained RL or constrained MDPs, the violation constraint is naturally defined without a max operator. Can you elaborate on this?
- In Section 4.1, several relaxation strategies for LLM selection have been discussed. However, are the solutions to these relaxations feasible to the "original" problem defined in Section 3 with the constraint that must be satisfied? This may be obviously infeasible. Can you clarify how to further design the solutions on top of the solutions to these relaxation problems?
- For the violation bound in Theorem 2, the right hand size of (8) has two terms, one in the order of $\mathcal{O}(\sqrt{K/T})$ and the other $\mathcal{O}(K/T)$. The authors claimed that the violation decreases at a rate of $\mathcal{O}(\sqrt{K/T})$. In addition, why "the overall violation is shown to be $\mathcal{O}(\sqrt{KT})$ as claimed in lines 457-458?
- How tight is the upper bound in Theorem 1? The coefficient can be extremely large on the right hand size of (7) associated with the $T$ terms. similarly for the constant term $(K+1)r^*$ that is in the order of $(K+1)NL$.

**Details Of Ethics Concerns:**

N/A.

---

> ### Author Response · Authors · 2024-11-19
> **Response to Reviewer XJC9**
>
> Thank you for your insightful comments! Here are our responses:
>
> ---
>
> ## Weaknesses
>
> 1. **Regarding  the intrinsic nature of the problem on combinatorial online optimization**
>
> Thank you for your thoughtful feedback. We acknowledge that the problem we study falls within the broader context of combinatorial online optimization, but it presents several novel aspects that distinguish it from existing work. To our knowledge, we are the first to study the CMAB problem with **long-term constraints** for **general reward functions** and **partial feedback models**. Additionally, we propose a unified and computationally efficient solution using the relaxation and rounding (RR) approach that provides both regret and violation guarantees.
> **For more discussions about the new challenges and techniques, please refer to our general response**.
>
> ---
>
> 2. **Regarding the Similarity Between C2MAB-V and Existing Methods:**
>
> We appreciate your feedback and would like to clarify the key differences that set C2MAB-V apart from previous **Combinatorial MAB (CMAB)** approaches.
>
> First, the central innovation of C2MAB-V lies in its approach to solving the constrained optimization problem. Traditional methods typically treat the constraint as a **discrete** optimization problem, where the objective is to maximize the total reward while ensuring that the cost does not exceed a certain threshold. However, this problem is often NP-hard, and directly solving it is either computationally inefficient or yields an approximate solution with a suboptimal approximation ratio. To overcome this challenge, we propose a novel strategy that avoids the difficulties of the discrete optimization problem by leveraging the **relaxation and rounding (RR)** technique, which transforms the problem into a **continuous** optimization problem. This approach offers improved approximation guarantees and better time complexity compared to direct methods.
>
> However, RR techniques were originally designed for offline settings, and due to the flexibility in choosing relaxation functions and rounding procedures, it is not immediately clear how to ensure that the RR approach will yield good online solutions that simultaneously achieve low regret and low violation. To address this, we identify general, unified properties for selecting appropriate relaxation functions and rounding procedures that are tailored for different task types. These properties establish a key connection between the regret in C2MAB-V and the over-estimation terms, which can be bounded through CMAB analysis. Furthermore, our novel **martingale construction** technique allows us to bound the long-term violation of the constraints.
>
> In our analysis, we tackle several technical challenges, including handling non-linear reward functions and partial feedback, which are critical for addressing various **LLM task scenarios**. These challenges are not covered by existing CMAB work, making our contributions both novel and valuable to a wide range of related problems.
>
> We have added a detailed discussion of the challenges and techniques used in the algorithm and analysis in Appendix A.3.
>
> ---
>
> 3. **Regarding the differences and analysis of reward models:**
>
> Thank you for your valuable comment.
>
> Firstly, the three reward models (AWC, SUC, and AIC) are distinguished based on the characteristics of the multi-LLM collaboration tasks they are designed to address. Each model reflects a different way of measuring success and failure. For example, the AWC model is designed to maximize user satisfaction in scenarios where multiple LLMs generate answers to a user query. The goal is to provide diverse solutions to enhance user experience; For example, the AIC model is typically applied in development tasks, where multiple LLMs collaborate to build different sub-modules of a larger system, where failure of any LLM (sub-module) leads to the failure of the entire task. Success relies on all models working together.
>
> Secondly, we would like to clarify that the **performance analysis in Section 5 applies to all proposed reward models**, which is also why they are presented in parallel. This is possible because, despite their differences, the models **share several key properties**, which we analyze and leverage to provide a unified theoretical guarantee. Specifically, we highlight these shared properties in **Appendix C.2**.
>
> ---
>
> 4. **Regarding the minor:**
>
> We have addressed the issue of overly tight spacing between lines, such as the lack of space between lines 243 and 244.
>
> ---

---

> ### Author Response · Authors · 2024-11-19
>
> ## Questions
>
> 1. **Regarding the differences between $K$ and $N$:**
>
> Thank you for your question. $K$ represents the **total number of LLMs deployed** in the scheduling cloud. $N$ represents the **maximum number of LLMs actually used** across all rounds, which is primarily used in the **theoretical analysis**. Obviously, $N$ may be less than or equal to $K$, since not all deployed models are necessarily used in every round.
>
> ---
>
> 2. **Regarding the versatile reward models:**
>
> Thank you for your valuable comment.  We would like to clarify that the **performance guarantees** discussed in **Section 5** are **applicable to all three reward models**: AWC, SUC, and AIC.  **Sections 3 and 4** provide the necessary context for understanding the task-specific characteristics of each reward model. The detailed discussion of these models is crucial for clarifying how the guarantees extend across different task types.
>
> For a more specific answer, please refer to our response to your  **Weakness 3**.
>
> ---
>
>
> 3. **Regarding  the definition of $V(T)$ :**
>
> The definition of \( V(T) \) follows a commonly used approach in the literature [1, 2, 3], which specifically focuses on capturing **budget overruns** rather than instances where costs fall below the budget.
>
> The **max operator** serves a practical purpose here: without it, negative values could imply "negative violations," which are not meaningful in this context. By using the max operator, the violation metric only captures the **cost exceeding the budget**, while ignoring situations where the cost remains below or within the threshold. This definition does not affect any of our theoretical analysis.
>
> [1]. Bernasconi M, Cacciamani F, Castiglioni M, et al. Safe learning in tree-form sequential decision making: Handling hard and soft constraints[C]//International Conference on Machine Learning. PMLR. 2022: 1854-1873.
>
> [2]. Chen K, Cai K, Huang L, et al. Beyond the click-through rate: web link selection with multi-level feedback[C]//Proceedings of the 27th International Joint Conference on Artificial Intelligence. 2018: 3308-3314.
>
> [3]. Saxena V, Jalden J, Gonzalez J. Thompson sampling for linearly constrained bandits[C]//International Conference on Artificial Intelligence and Statistics. PMLR. 2020: 1999-2009.
>
> ---
>
>
> 4. **Regarding from the relaxed problem to the original problem:**
>
> Thank you for your comment regarding the feasibility of the solutions to the relaxed problem.
>
> Our framework includes a non-trivial **discretization rounding process** (please refer to **Section 4.2** and **Appendix B**) to convert the **relaxed continuous solutions** into feasible, **discrete solutions** that satisfy the original problem constraints. Specifically:
>
> - *Relaxation to Discretization*: After solving the relaxed continuous optimization problem, where each selection variable $z_k$ is allowed to be in the range $[0, 1]$, we convert these values into binary selections 0 or 0 for each LLM.
>
> - We employ **discretization rounding algorithms**  that are tailored to each reward model (AWC, SUC, and AIC). These algorithms are designed to **ensure that the resulting selections are feasible** while maintaining an optimal balance between the **budget constraint** and the **reward**.
>
> This approach ensures that relaxed solutions are appropriately transformed into discrete, feasible actions that satisfy the original constraints. Please note that selecting different rounding algorithms tailored to various reward models, while maintaining consistency within a unified theoretical framework, is **not an easy task**. It represents a significant effort in identifying suitable methods for our specific setting.
>
> ---

---

> ### Author Response · Authors · 2024-11-19
>
> 5. **Regarding the violation bound:**
>
> Let me clarify both points about the asymptotic behavior and the cumulative violation:
>
> a. *Asymptotic Dominance of $O\left(\sqrt{\frac{K}{T}}\right)$*:
>    - In Equation (8), the right-hand side contains two terms: $O\left(\sqrt{\frac{K}{T}}\right)$ and $O\left(\frac{K}{T}\right)$.
>    - As **$T$** becomes large, the term $O\left(\frac{K}{T}\right)$ decays faster compared to $O\left(\sqrt{\frac{K}{T}}\right)$, making it asymptotically negligible. Thus, for large $T$, the violation is primarily dominated by the slower-decaying term, $O\left(\sqrt{\frac{K}{T}}\right)$.
>
>
> b. *Cumulative Violation Over $T$ Rounds*:
>    - The **per-round violation bound** derived in Equation (8) is asymptotically $O\left(\sqrt{\frac{K}{T}}\right)$. To calculate the **total cumulative violation** over $T$ rounds, we sum the per-round violation bounds across all rounds:
> $$
> V_{\text{cumulative}}(T) = \sum_{t=1}^T O\left(\sqrt{\frac{K}{T}}\right) = T \cdot O\left(\sqrt{\frac{K}{T}}\right) = O\left(\sqrt{KT}\right).
> $$
>    - Thus, the overall violation is $O\left(\sqrt{KT}\right)$.
>
> ---
>
> 6. **Regarding the regret bound:**
>
> Firstly, the reasons our regret bound is considered tight compared to other CMAB works are as follows.  In most existing CMAB works, the primary source of regret comes from the **over-estimation of unknown parameters**. In our work, we also incorporate a **relaxation and rounding procedure**, which can introduce additional regret due to the gap between the relaxed and discrete solutions. However, based on our unique exploration analysis of attributes of reward functions (see Appendix C.2), the additional regret introduced by the relaxation and rounding process is **non-positive in expectation**. This ensures that the overall regret bound remains tight relative to other existing approaches.
>
>
> Regarding the coefficient $o^*$, the coefficient $o^*$ in the regret bound is also present in other related works, such as  [1, 2]. It represents the **partial observation** probability, which is an inherent feature of dealing with **partial observations** in a combinatorial setting  [1, 2]. Improvements to this coefficient are possible by adopting a **stronger assumption** (Triggering Probability Modulated Lipschitz continuity)  [3] for the reward function. Incorporating such an adaptation could help in reducing the impact of the coefficient, and we consider this a promising direction for future work.
>
> The term $(K+1)r^*$ is a **constant** term that does not scale with the time horizon $T$. As a result, it does not dominate the regret bound asymptotically. This behavior aligns with regret bounds observed in other CMAB frameworks and arises from addressing uncertainty in the number of arms.
>
> Finally, in **Appendix D.2**, we added an analysis of the **more challenging distribution-dependent** regret bound because the regret reduction in Step 1 of Appendix D.2 holds only in expectation, which **smooths out the distribution-dependent terms**. Therefore, we needed an alternative approach that directly retains the instance-specific details of the learning process.
>
>
> [1]. Wei Chen, Yajun Wang, Yang Yuan, et al. Combinatorial multi-armed bandit and its extension to probabilistically triggered arms. The Journal of Machine Learning Research. 2016: 1746–1778.
>
> [2]. Branislav Kveton, Csaba Szepesvari, Zheng Wen, et al.  Cascading bandits: Learning to rank in the cascade model. In International Conference on Machine Learning. PMLR. 2015: 767–776.
>
> [3]. Wang Q, Chen W. Improving regret bounds for combinatorial semi-bandits with probabilistically triggered arms and its applications[J]. Advances in Neural Information Processing Systems. 2017: 30.

---

> ### Author Response · Authors · 2024-11-25
>
> Dear Reviewer XJC9,
>
> As the discussion phase is drawing to a close, we kindly ask whether our responses have resolved your concerns or if there are any remaining issues that we can further clarify. Your insights are invaluable in refining our work, and we are eager to ensure that all your concerns are fully addressed.
>
> Thank you once again for your time and effort in reviewing our manuscript.
>
> Best regards,\
> Authors

---

> > ### Comment · Reviewer_XJC9 · 2024-11-26
> > **Thank you**
> >
> > Thank you for the rebuttal and I appreciate your efforts on addressing my comments on the regret bounds etc, which I roughly missed the calculation earlier. As mentioned earlier, despite this work is framed under the current popular topic on LLM, the nature of the problem is not "too" different. I agree with other reviewers regarding the technical contributions and the novelty of the paper, although the authors put many existing techniques together to solve a hard problem. I will keep the current rating.

---

> ### Author Response · Authors · 2024-11-27
>
> Thank you for your thoughtful feedback. We respectfully disagree that our work merely puts existing techniques together:
>
> ---
> Our work **innovatively introduces relaxation and rounding (RR) techniques** into the **online learning** domain. While previous RR approaches were primarily designed for **offline problems**, using RR in an online setting presents unique challenges due to the need for both **flexibility** in choosing relaxation functions and the complexity of ensuring **simultaneously low regret and low violation**. This is particularly challenging, as RR methods are not easily combined with existing online learning frameworks.
>
> By leveraging RR, we are able to **avoid the complexities of the original NP-hard problem**, significantly **improving time complexity** compared to previous approaches (see **Appendix E.3, Table 5**). Furthermore, our approach enhances the **$\alpha$-Approximate Regret**, with the $\alpha$ values for similar reward functions in [1, 2] being $ 1 - 1/K $, $ 1 - 1/K $, and $\frac{1}{4(1+\varepsilon)}, \varepsilon > 0 $, compared to our values of **1**, **1**, and **1 - 1/e**. This improvement is highlighted in the comparison table below.
>
> ---
>
> Moreover, we formally **formulated the multi-LLM selection problem** and provided a **unified, comprehensive algorithmic design and theoretical analysis** for various multi-LLM task types by analyzing the properties of reward functions. Our proof techniques are highly general, extending to a wide class of problems that satisfy the reward properties discussed, thereby making the framework applicable beyond LLM selection.
>
> ---
>
> To demonstrate the **generality** of our approach, we have also added **three representative applications** that fit within our framework and provided comparisons with existing works in the following table:
>
> | **Application**           | **Reward Function** | **Offline Oracle**                       | **Approximation**              |
> |----------------------------|---------------------|------------------------------------------|---------------------------------|
> | Dynamic Assortments        | SUC                 | 0-1 Knapsack [1]                       | $\alpha = 1 - 1/K$             |
> | Dynamic Assortments    | SUC                 | **Our Method**                          | **$\alpha = 1$**               |
> | Network Routing            | AIC                 | 0-1 Knapsack [1]                       | $\alpha = 1 - 1/K$             |
> | Network Routing       | AIC                 | **Our Method**                          | **$\alpha = 1$**               |
> | Movie Recommendations      | AWC                 | Submodular Maximization [2]            | $\alpha = \frac{1}{4(1+\varepsilon)}, \varepsilon > 0$ |
> | Movie Recommendations  | AWC                 | **Our Method**                          | **$\alpha = 1 - 1/e$**         |
>
> - a. *Dynamic Assortment*: An agent selects some products from K options to maximize total sales, subject to a cost constraint. For this application, the goal is to **maximize the total reward**, making it suitable for the **SUC reward model**.
>
> - b. *Network Routing*: An agent selects a path in a network of routers to maximize the probability of successful packet delivery while maintaining average cost constraints. For this application, if **any router in the selected path fails, the entire connection fails**, making it suitable for the **AIC reward model**.
>
> - c. *Movie Recommendations*: An agent recommends movies to users, aiming to maximize the click probability while adhering to budget constraints. For this application, **users stop browsing as soon as they find a satisfactory movie**, making it suitable for the **AWC reward model**.
>
> These examples demonstrate that our framework is applicable to a wide range of **general combinatorial bandit problems** beyond the specific LLM selection task and effectively maps each scenario to a suitable reward model. We have included these more detailed application descriptions and the comparison table in our revised version.
>
> ---
>
> [1]. Timothy M Chan. 2018. Approximation schemes for 0-1 knapsack. In 1st Symposium on Simplicity in Algorithms (SOSA 2018). Schloss Dagstuhl-Leibniz-Zentrum fuer Informatik.
>
> [2]. Sho Takemori, Masahiro Sato, Takashi Sonoda, Janmajay Singh, and Tomoko Ohkuma. 2020. Submodular bandit problem under multiple constraints. In Conference on Uncertainty in Artificial Intelligence. PMLR, 191–200.

---

### Official Review · Reviewer_hptZ · 2024-11-05

**Soundness:** 3
**Presentation:** 3
**Contribution:** 3
**Rating:** 6
**Confidence:** 4

**Summary:**

This paper introduces a model of Cost-effective Combinatorial Multi-armed Bandit with Versatile reward for the optimal selection of multiple large language models (LLMs). The proposed approach aims to manage the exploration-exploitation trade-off while balancing costs and rewards in a collaborative LLM setting, using an online learning framework. The paper provides a theoretical analysis of regret and budget violation bounds, demonstrating that the proposed approach C2MAB-V matches state-of-the-art results in certain degenerate cases. Empirical experiments validate its effectiveness in balancing cost-efficiency and performance across three scenarios with nine LLMs.

**Strengths:**

1. The formulation of the problem as a cost-effective combinatorial multi-armed bandit with versatile reward models is interesting and applicable in real-world scenarios where cost-efficiency is crucial. The paper reads well and includes examples/intuition to help the reader.

2.  The authors provide a rigorous theoretical analysis of the regret and budget violations of the proposed algorithm, achieving results that match the state-of-the-art in several degenerate cases. Plus, the case-by-case analysis for different reward models provides enough insights about the discretization rounding procedure, which is crucial for understanding the proposed method.

3. The paper presents extensive empirical evaluations using various LLMs across three application scenarios, providing strong evidence for the proposed method's effectiveness.

**Weaknesses:**

1. The algorithms and theoretical analysis presented are natural extensions of previous approaches, specifically in the combinatorial multi-armed bandit literature. Also, I do not believe that non-linear rewards pose a significant challenge for the analysis, as the paper assumes that the reward function is Monotone and Lipschitz. Therefore, the regret analysis can be converted to the accumulated impact of overestimating rewards, just as most of the literature does. Hence, I cannot give a score higher than 6 (accept).

2. Lines 1240-1241 contain content that seems unrelated to the context and should be revised or removed.

3. The literature on Combinatorial Multi-Armed Bandit with Knapsack Constraints is highly relevant to this paper's content, yet it has not been formally reviewed in Section 2.1 (Related Work). I highly recommend the authors Incorporate a formal review of these works in the revised paper.

**Questions:**

1. I noticed that LLaMA 2 was used for comparison in the experiments, but LLaMA 3 has also been released. I am curious why there was no comparison with LLaMA 3 in the experiments.

2. When the reward function is linear, can instance-dependent performance bounds be derived based on a sub-optimality gap defined by the extreme points?

---

> ### Author Response · Authors · 2024-11-19
> **Response to Reviewer hptZ**
>
> Thank you for your insightful comments! Here are our responses:
>
> ---
>
> ## Weaknesses
>
> 1. **Regarding theoretical analysis, especially on the complexity of non-linear reward analysis:**
>
> Thank you for your thoughtful feedback. We acknowledge that the problem we study falls within the broader context of combinatorial online optimization, but it presents several novel aspects that distinguish it from existing work. To our knowledge, we are the first to study the CMAB problem with **long-term constraints** for **general reward functions** and **partial feedback models**. Additionally, we propose a unified and computationally efficient solution using the relaxation and rounding (RR) approach that provides both regret and violation guarantees.
> **For more discussions about the new challenges and techniques, please refer to our general response**.
>
> Moreover, we respectfully disagree with the assertion that non-linear reward analysis is straightforward in our setting. As mentioned, unlike the traditional approach of treating the constraint as an NP-hard **discrete optimization problem**, our approach uses RR techniques to avoid these challenges. However, the analysis of **non-linear rewards**  brings its own complexities:
>
> a. *Relaxation Challenges*:
>
>    Unlike linear reward functions, where a simple concave relaxed function can be used, a similar approach is not possible for non-linear rewards. For instance, the relaxed function for non-linear rewards cannot be defined as $ 1 - \prod\_{k \in \mathcal{K}} (1 - \tilde{z}\_k) $. To overcome this, we treat the non-linear reward as a **submodular function** and use its **multi-linear extension**:
>    $$
>    \tilde{r} \left(\tilde{\boldsymbol{Z}} , \bar{\boldsymbol{\mu}}\right) = \sum\_{S \subseteq \mathcal{K}} \prod\_{k\in S}\tilde{z}\_k \prod\_{k\notin S}(1-\tilde{z}\_k).
>    $$
>    This allows our solution to be applicable to a wider class of  **submodular reward functions**.
>
> b. *Rounding Procedure*:
>
>    In terms of rounding, we need to ensure that the quality of any action $S$ with respect to the reward objective ($ \mathbb{E}\_{S \sim \sigma(\boldsymbol{Z})}[r(S,\boldsymbol{\mu})] $) is no worse than the value of the relaxed function in expectation. The rounding algorithm designed for linear rewards (Algorithm 3) is not applicable in this case. This requires us to introduce a new **randomized swap rounding procedure** (Algorithm 2), which rounds the relaxed solution $\tilde{\boldsymbol{Z}}$ while preserving convexity properties, i.e., $\mathbb{E}\left[\tilde{r}\left(\mathbb{I}\_{S\_t}, \boldsymbol{\mu}\right)\right] \geq \tilde{r}(\tilde{\boldsymbol{Z}}\_t, \boldsymbol{\mu})$.
>
> These adjustments show the additional complexities involved in handling non-linear rewards, necessitating significant modifications to both relaxation and rounding techniques.
>
> ---
>
> 2. **Regarding the unrelated content:**
>
> Thank you for pointing this out. We have removed the unrelated content.
>
> ---
>
> **Regarding the  literature on bandits with knapsack constraints:**
>
> Thank you for your suggestion. In the initial version of our manuscript, we did provide a comparison with works on bandits with knapsack constraints in the remark of Section 5 of the theoretical analysis. Additionally, we have added a formal review of relevant works on **Bandits with Knapsack Constraints** in **Appendix A.2**, comparing them with our approach.
>
> Key points of comparison are as follows:
>
> - Several works consider **resource consumption and budget constraints** in stochastic MAB/CMAB settings, such as **bandits with budgets** and **bandits with knapsacks (BwK)**. These works typically focus on **optimal stopping times**, where the learning process ends once resources are exhausted. In contrast, our model employs a **long-term constraint**, allowing the arm selection process to continue for an arbitrary duration.
>
> - Among BwK studies, "Combinatorial semi-bandits with knapsacks" is closest to our work. Both use a combination of **UCB/LCB, linear programming, and randomized rounding**. However, their approach is restricted to **linear CMAB** settings with linear rewards and **semi-bandit feedback**. Our framework generalizes this approach, as our algorithm also applies to non-linear settings using different relaxation and rounding techniques.
>
> - A notable difference is that our proof technique does not rely on the **negative correlation property** of randomized rounding, providing more flexibility in the choice of relaxation functions and rounding procedures for a broader class of non-linear reward functions.
>
> ---

---

> > ### Author Response · Authors · 2024-11-24
> >
> > We would like to clarify that the **L-Lipschitz continuity** is **not assumed** but **derived as an intrinsic property** of our reward models. Specifically, we establish that all three reward models are **1-Lipschitz continuous** with respect to LLM performance $\boldsymbol{\mu}$.
> >
> > - *AIC*:
> >
> > $ |r(S;\boldsymbol{\mu})-r(S;\boldsymbol{\mu}')|=|\prod\_{k \in S} \mu\_k - \prod\_{k \in S} \mu'\_k|= |\sum\_{k=1}^K (\prod\_{j=1}^{k-1}{\mu\_{j}}) (\boldsymbol{\mu}\_{k} - \mu'\_{k})(\prod\_{j=k+1}^K\mu\_{j})|
> > \le  \sum\_{k=1}^K | (\prod\_{j=1}^{k-1}{\mu'\_{j}}) (\boldsymbol{\mu}\_{k} - \mu'\_{k})(\prod_{j=k+1}^K \mu\_{j})|
> > \le  \sum\_{k=1}^K |\mu\_{k} - \mu'\_{k}|.
> > $
> >
> > - *SUC*:
> >
> > $|r(S;\boldsymbol{\mu})-r(S;\boldsymbol{\mu}')|=|\sum_{k \in S} \mu_{k} - \sum_{k \in S} \mu_{k}'| \le  \sum_{k=1}^K |\mu_{k} - \mu'_{k}|$.
> >
> > - *AWC*:
> >
> > $|r(S;\boldsymbol{\mu})-r(S;\boldsymbol{\mu}')|=|\prod_{k \in S} (1-\mu_k)) - \prod_{k \in S} (1-\mu_k)')|$, and we can follow the same derivation as AIC by introducing a new variable $\lambda = \mu$.
> >
> >
> >  Thus, our theoretical analysis extends to the more general **L-Lipschitz setting**. We have clarified these in  **Section 5** and **Appendix C.2**.

---

> ### Author Response · Authors · 2024-11-19
>
> ## Questions
>
> 1. **Regarding the addition of LLaMA 3:**
>
> Thank you for your comment. At the time we initially conducted our experiments, LLaMA 3 had not yet been released. In our revised version, we are currently running experiments to include LLaMA 3 in additional evaluations on other public datasets to demonstrate the robustness of our approach. **We will strive to include these results in a revised version of the manuscript** before the discussion phase ends.
>
> ---
>
> 2. **Regarding instance-dependent bound for linear reward function:**
>
> Thank you for your question. We have provided a unified **instance-dependent performance bound** for not only the **linear reward function** but also the **versatile reward models** discussed in our paper.
>
> Proving a **distribution-dependent bound** in our setting is more challenging because the **regret reduction** in Step 1 of **Appendix D.2** holds only **in expectation**. This expectation smooths out the **action-dependent terms** that are critical for deriving a distribution-dependent bound. Therefore, we needed an alternative approach that directly retains the instance-specific details of the learning process.
>
> To achieve this, we firstly introduced a new condition—the **Sampling Quality Condition**—which ensures that the reward of any sampled action is no worse than the relaxed reward by at most a small margin $\gamma$. This condition places a lower bound on the sampling quality in each round, rather than just in expectation.
>
> We also used a **surrogate regret** $
> \Delta'\_{S\_t} := \Delta\_{S\_t} + r(S\_t, \bar{\boldsymbol{\mu}}\_t) - \tilde{r}(\tilde{\boldsymbol{Z}}\_t, \bar{\boldsymbol{\mu}}\_t)
> $ to facilitate the analysis. This surrogate regret accounts for both the discrepancy between the relaxed and actual rewards, as well as the error introduced by the rounding procedure, ensuring it remains non-negative.
>
> By leveraging these techniques, we derived an instance-dependent regret bound that holds for all versatile reward models in our setting. The complete proof of the instance-dependent performance bound can be found in **Appendix D.4** of the revised version.

---

> ### Author Response · Authors · 2024-11-25
>
> Dear Reviewer hptZ,
>
> As the discussion phase is drawing to a close, we kindly ask whether our responses have resolved your concerns or if there are any remaining issues that we can further clarify. Your insights are invaluable in refining our work, and we are eager to ensure that all your concerns are fully addressed.
>
> Thank you once again for your time and effort in reviewing our manuscript.
>
> Best regards,\
> Authors

---

> > ### Comment · Reviewer_hptZ · 2024-11-26
> > **Re: Author Response**
> >
> > Thank you for your detailed response and most of my concerns have been addressed. However, I still think the technical contributions of the paper not sufficiently novel for ICLR. As such, I will maintain my current score.

---

> ### Author Response · Authors · 2024-11-27
>
> Thank you for your detailed response. We respectfully disagree with the assessment that our technical contributions lack sufficient novelty.
>
> ---
>
> First, we formally formulated the **multi-LLM selection problem**, taking into account the diverse reward requirements and cost constraints across different types of multi-LLM tasks. This is a novel and systematic approach that has not been fully addressed in prior works. To validate the practicality and feasibility of our algorithm, we conducted experiments on **four publicly available datasets** and tested the performance of **10 LLMs (including LLaMA 3)**, demonstrating the robustness and real-world applicability of our method.
>
> ---
>
> Second, the theoretical techniques we employ are far from trivial. A straightforward approach to selecting an action that **maximizes reward while satisfying cost constraints** would be to solve a **discrete constrained optimization (CO) problem**, aiming to maximize the total UCB values while ensuring the total LCB values remain below a threshold. However, such CO problems are typically **NP-hard**, and solving them directly is either computationally inefficient or leads to unsatisfactory approximation ratios.
>
> To overcome these challenges, we propose a novel approach by leveraging the **relaxation and rounding (RR) technique**. Instead of directly solving the hard CO problem, we **transform** it into a **continuous optimization problem**, which allows us to achieve better **approximation guarantees** and **improved time complexity**.
>
> The use of RR in this context introduces a new challenge, as RR techniques were originally designed for **offline problems**. Given the flexibility required in choosing relaxation functions and the **randomized rounding procedure**, it is not immediately clear how to ensure the RR approach can provide good **online solutions**, achieving **low regret** and **low constraint violation** simultaneously. Developing a robust RR framework that works well in the online setting with these guarantees is a significant technical contribution and an important advancement over previous approaches.
>
> In our analysis, we address several technical challenges to handle **non-linear reward functions** and **partial feedback**, both of which are essential to cover different application scenarios. Additionally, we consider **distribution-dependent regret bounds**, extending the applicability and robustness of our method.

---

### Official Review · Reviewer_RM4A · 2024-11-07

**Soundness:** 2
**Presentation:** 2
**Contribution:** 2
**Rating:** 5
**Confidence:** 3

**Summary:**

This paper formulates the multi-LLM selection problem as a combinatorial bandit problem. The authors develop an online bandit algorithm for online multi-LLM selection where different types of reward modes are applied. Also, the authors consider a computational cost constraint on the LLM selection. The paper provides analysis for regret and constraint violation.  The performance is evaluated based on  the SciQ dataset.

**Strengths:**

This paper has the following strengths:
+ This paper consider an interesting problem which selects LLM models on cloud.
+ This paper provides good formulations for online LLM selection problem. Different reward models are considered. Budget constraint of the scheduling cloud is also modeled.
+ Performance analysis is provided for the combinatorial bandit problem.

**Weaknesses:**

The paper can be improved in the following aspects.
- This paper considers the LLM selection problem. However the reward are modeled as some simple equations in page 5 and look like some toy models. These models cannot accurately reflect the real evaluation of combinatorial LLMs.
- The algorithm design is a simple extension of combinatorial bandits. The analysis of the algorithm also loos similar as the previous results on combinatorial bandits. It would be better if the authors can present the new challenges and techniques used in the algorithm and analysis.
- The empirical results are carried out for only one dataset.  Also, the choices of the parameters like budget threshold, $\alpha$, etc are not explained. This is not enough to verify the proposed algorithms.

**Questions:**

Please see the questions above.

---

> ### Author Response · Authors · 2024-11-19
> **Response to Reviewer RM4A**
>
> Thank you for your insightful comments! Here are our responses:
>
> ---
> ## Weaknesses
> 1. **Regarding the design of the reward models:**
>
> Thank you for your comment. The versatile reward models presented are designed to ensure that our approach can be applied broadly across different task types, capturing the core requirements of multi-LLM collaboration.
> Specifically, we define three types of reward models:
>
> a. *Any Win Combination (AWC):*  $r(S;\boldsymbol{\mu}) = 1 - \prod_{k \in S} (1 - \mu_{k}).$ This model ensures user satisfaction by selecting multiple LLMs, where success occurs if any LLM meets the user’s requirements.
>
> b. *Sum Up Combination (SUC):*   $r(S;\boldsymbol{\mu}) = \sum_{k \in S} \mu_{k}. $ Domain-specific LLMs work in parallel, earning rewards for answering questions in their domain, thus speeding up task completion.
>
> c. *All In Combination (AIC):*  $ r(S;\boldsymbol{\mu}) = \prod_{k \in S} \mu_{k}.$ Used in development tasks, this model requires all LLMs to succeed—failure of any LLM results in failure of the task.
>
> These reward models are explicitly defined and formulated to provide rigorous theoretical analysis while reflecting the core needs of multi-LLM collaboration.
>
> Note that while prior research on multi-LLM collaboration has often focused on a single task type (e.g., AWC), our work provides a **unified theoretical framework** that applies across multiple task types, each with its own reward structure.
>
> Additionally, in **Appendix C.2**, we provide a more detailed **discussion on constraint types and extended task types**. We welcome your specific suggestions for improving the reward model, and we would be happy to incorporate any recommendations to enhance the discussion further.
>
> ---
>
> 2. **Regarding the new challenges and techniques used in the algorithm and analysis:**
>
> Thank you for your thoughtful feedback. We acknowledge that the problem we study falls within the broader context of combinatorial online optimization, but it presents several novel aspects that distinguish it from existing work. To our knowledge, we are the first to study the CMAB problem with **long-term constraints** for **general reward functions** and **partial feedback models**. Additionally, we propose a unified and computationally efficient solution using the relaxation and rounding (RR) approach that provides both regret and violation guarantees.
>
> **For more discussions about the new challenges and techniques, please refer to our general response**.
>
> ---
>
> 3. **Regarding experimental results and parameter clarification:**
>
> Thank you for your feedback. We are currently running experiments to include **multiple publicly available datasets** and **we will strive to include these results in a revised version of the manuscript** before the discussion phase ends.
>
> Regarding the **budget threshold**, it is executor-defined and **not determined by the algorithm**. Executors are responsible for specifying their desired budget, and our experiments **evaluate the algorithm’s performance under different budget thresholds**. In **Figure 8 of Appendix E.2**, we present results showing how varying the budget threshold affects the algorithm's performance, highlighting its versatility.
>
> As for **$\alpha$**, please note that it is **not a hyperparameter**.  Instead, $\alpha$ represents the **$\alpha$-regret**, a standard concept in **combinatorial MAB**, as described in Section 3, under **$\alpha$-Approximate Regret**.  The value of $\alpha$ depends on the theoretical properties of the reward models.  Specifically, for the reward models defined in our paper (AWC, SUC, and AIC), the corresponding values of $\alpha$  are **1-1/e, 1, and 1**, respectively, as detailed in **Appendix B**.

---

> ### Comment · Reviewer_RM4A · 2024-11-23
>
> I appreciate the authors' response which addresses some of my concerns.
>
> The technical contribution in terms of combinatorial bandits with long term constraints and partial feedback looks important and interesting. I recommend the authors to include a literature review about bandits with long term constraints and include their comparison table. Also, the authors need to clarify the assumptions for the reward and cost functions. Do we need other assumptions except for L-Lipschitz continuous?
>
> My remaining concern is that the theory is not strongly supporting the application. From the title and abstract, I feel that the main contribution of the paper is to solve a concrete application of selecting LLMs. However, the analysis is performed on a general Lipschitz continuous reward function, but this does not show the performance of different reward models in LLM selection. Are the reward modes in LLM selection L-Lipschitz continuous? What is the Lipschitz constant for each reward model? Moreover, when selecting LLMs, do we really need an unreliable online exploration? And, can we include some data-driven reward models learned from human preference?

---

> > ### Author Response · Authors · 2024-11-24
> > **Response to Remaining Concerns**
> >
> > Thank you for your comments and for appreciating our technical contributions. We have made the following updates and clarifications to address your remaining concerns.
> >
> > ---
> >
> > 1. **Literature Review on Bandits with Long-Term Constraints**:
> >
> > We have now added a **formal review of relevant works on Bandits with Constraints** in **Appendix A.2**, which includes a comparison table with existing works on **bandits with long-term constraints**. While prior works use a combination of UCB/LCB, they are typically restricted to **linear CMAB settings** with **semi-bandit feedback**. In contrast, our framework extends this approach to handle **non-linear reward functions** through different **relaxation and rounding techniques**, making it applicable to broader scenarios.
> >
> > ---
> >
> > 2. **Clarification on Assumptions for Reward and Cost Functions**:
> >
> > We would like to clarify that we **do not** make prior assumptions about the **cost** and **reward functions** of individual LLMs; instead, they are learned through **online feedback**. The **overall reward definitions for multi-LLM tasks** (AWC, SUC, AIC) are drawn from common multi-LLM collaboration scenarios and are validated by existing literature [1, 2, 3].
> >
> > The **L-Lipschitz continuity** is **not assumed** but **derived as an intrinsic property** of our reward models. Specifically, we establish that all three reward models are **1-Lipschitz continuous** with respect to LLM performance $\boldsymbol{\mu}$.
> >
> > - *AIC*:
> >
> > $ |r(S;\boldsymbol{\mu})-r(S;\boldsymbol{\mu}')|=|\prod\_{k \in S} \mu\_k - \prod\_{k \in S} \mu'\_k|= |\sum\_{k=1}^K (\prod\_{j=1}^{k-1}{\mu\_{j}}) (\boldsymbol{\mu}\_{k} - \mu'\_{k})(\prod\_{j=k+1}^K\mu\_{j})|
> > \le  \sum\_{k=1}^K | (\prod\_{j=1}^{k-1}{\mu'\_{j}}) (\boldsymbol{\mu}\_{k} - \mu'\_{k})(\prod_{j=k+1}^K \mu\_{j})|
> > \le  \sum\_{k=1}^K |\mu\_{k} - \mu'\_{k}|.
> > $
> >
> > - *SUC*:
> >
> > $|r(S;\boldsymbol{\mu})-r(S;\boldsymbol{\mu}')|=|\sum_{k \in S} \mu_{k} - \sum_{k \in S} \mu_{k}'| \le  \sum_{k=1}^K |\mu_{k} - \mu'_{k}|$.
> >
> > - *AWC*:
> >
> > $|r(S;\boldsymbol{\mu})-r(S;\boldsymbol{\mu}')|=|\prod_{k \in S} (1-\mu_k)) - \prod_{k \in S} (1-\mu_k)')|$, and we can follow the same derivation as AIC by introducing a new variable $\lambda = \mu$.
> >
> >
> >  Thus, our theoretical analysis extends to the more general **L-Lipschitz setting**. We have clarified these in  **Section 5** and **Appendix C.2**. Beyond the contribution of solving the multi-LLM selection problem, our general theoretical analysis itself is novel and of independent interest.
> >
> > ---
> >
> > 3. **Exploration in LLM Selection**:
> >
> > Regarding the **exploration** aspect of LLM selection, it is not "unreliable" as suggested. Rather, our exploration strategy is **carefully designed**, relying on **Lemma 1** to ensure **high-probability** events where empirical estimates for both rewards and costs align closely with their true means. Without this exploration component, the approach would **reduce to a greedy algorithm**, which may be unreliable and could lead to **sub-optimal** choices without considering potentially better LLMs.
> >
> > ---
> >
> > 4. **Contextual Bandit for Human Preferences**:
> >
> > Our current implementation focuses on a **non-contextual setup**, but we recognize the potential of extending our framework to include **contextual combinatorial multi-armed bandits** to better capture human preferences. We have highlighted our intention to explore this extension in **Section 7** as part of future research directions.
> >
> > ---
> >
> > [1]. Zijun Liu, Yanzhe Zhang, Peng Li, Yang Liu, and Diyi Yang. Dynamic llm-agent network: An llm-agent collaboration framework with agent team optimization. *arXiv preprint arXiv:2310.02170*, 2023c.
> >
> > [2]. Lingjiao Chen, Matei Zaharia, and James Zou. Frugalgpt: How to use large language models while reducing cost and improving performance. *arXiv preprint arXiv:2305.05176*, 2023.
> >
> >
> > [3]. Sirui Hong, Xiawu Zheng, Jonathan Chen, Yuheng Cheng, Jinlin Wang, Ceyao Zhang, Zili Wang, Steven Ka Shing Yau, Zijuan Lin, Liyang Zhou, et al. Metagpt: Meta programming for multi-agent collaborative framework. *arXiv preprint arXiv:2308.00352*, 2023.

---

> > > ### Comment · Reviewer_RM4A · 2024-11-24
> > >
> > > Thank you for your response. I increased the score.
> > >
> > > I am still not sure about some technical points provided by the theory, so I hope to discuss more with the authors regarding the following points.
> > >
> > > 1. The paper considers $\alpha-$approximate regret in (2). The regret is largely affected by the parameter  $\alpha$. If $\alpha=0$, we can even get a negative regret. Did you choose a concrete $\alpha$ in your analysis? If so, $\alpha$ is important to evaluate the performance of the algorithm.
> > >
> > > 2. The parameters $\alpha_u$ and $\alpha_c$ are key parameters for the explorations of reward and cost. How do they affect the regret and constraint violation? How do you choose them?
> > >
> > > 3. The authors claim that they study a partial feedback setting. How does the partial feedback subset or its cardinality $F_t$ affect the regret bound and constraint violation?
> > >
> > > 4. How does the analysis indicate the trade-off between reward and cost constraint violation? Are there some parameters that control the trade-off?
> > >
> > > 5. The 'unreliable' concern comes from the exploration at the early stage. As shown in Figure 4, the reward is low and the violation ratio is high at the early rounds. But, I understand this issue as this paper studies online setting.
> > >
> > > 6. As is also mentioned by other reviewers, the algorithm and theory are proposed for a general combinatorial bandit problem instead of a specific LLM selection problem. I would recommend the authors to include some other application examples in motivation or formulation sections to highlight that the algorithm can be used for more general problems.

---

> > > > ### Author Response · Authors · 2024-11-25
> > > >
> > > > 5. **Regarding the Early-Stage Exploration**:
> > > >
> > > > Thank you for your understanding of the challenges associated with early-stage exploration in the online setting. As you observed in Figure 4, the performance is temporarily suboptimal in the early rounds due to **insufficient feedback**.
> > > >
> > > > Our study **completely focuses on online learning** from continuous feedback, which naturally faces challenges in the initial stages. Incorporating **offline pre-training** or addressing **cold-start issues** like [3] is a promising approach for accelerating online learning early on and could be seamlessly **integrated into** our framework.
> > > >
> > > > ---
> > > >
> > > > 6. **Regarding General Application Examples Beyond LLM Selection**:
> > > >
> > > > Thank you for the suggestion. We agree that highlighting broader applications is important to show the generality of our approach. To that end, we have added **three representative applications** that fit within our framework and provided comparisons with existing works in the following table.
> > > >
> > > > | **Application**           | **Reward Function** | **Offline Oracle**                       | **Approximation**              |
> > > > |----------------------------|---------------------|------------------------------------------|---------------------------------|
> > > > | Dynamic Assortments        | SUC                 | 0-1 Knapsack [1]                       | $\alpha = 1 - 1/K$             |
> > > > | Dynamic Assortments    | SUC                 | **Our Method**                          | **$\alpha = 1$**               |
> > > > | Network Routing            | AIC                 | 0-1 Knapsack [1]                       | $\alpha = 1 - 1/K$             |
> > > > | Network Routing       | AIC                 | **Our Method**                          | **$\alpha = 1$**               |
> > > > | Movie Recommendations      | AWC                 | Submodular Maximization [2]            | $\alpha = \frac{1}{4(1+\varepsilon)}, \varepsilon > 0$ |
> > > > | Movie Recommendations  | AWC                 | **Our Method**                          | **$\alpha = 1 - 1/e$**         |
> > > >
> > > >
> > > >
> > > > - a. *Dynamic Assortment*:
> > > >
> > > > An agent selects some products from $ K $ options to maximize total sales, subject to a cost constraint. For this application, the goal is to *maximize the total reward*, making it suitable for the **SUC reward model**.
> > > >
> > > > - b. *Network Routing*:
> > > >
> > > > An agent selects a path in a network of routers to maximize the probability of successful packet delivery while maintaining average cost constraints. For this application, if *any router in the selected path fails, the entire connection fails*, making it suitable for the **AIC reward model**.
> > > >
> > > > - c. *Movie Recommendations*:
> > > >
> > > > An agent recommends movies to users, aiming to maximize the click probability while adhering to budget constraints. For this application, *users stop browsing as soon as they find a satisfactory movie*, making it suitable for the **AWC reward model**.
> > > >
> > > > These examples demonstrate that our framework is applicable to a wide range of **general combinatorial bandit problems** beyond the specific LLM selection task, and effectively maps each scenario to a suitable reward model. We have included these more detailed application descriptions and the comparison table in our revised version.
> > > >
> > > > ---
> > > >
> > > > [1]. Timothy M Chan. 2018. Approximation schemes for 0-1 knapsack. In 1st Symposium on Simplicity in Algorithms (SOSA 2018). Schloss Dagstuhl-Leibniz-Zentrum fuer Informatik.
> > > >
> > > > [2]. Sho Takemori, Masahiro Sato, Takashi Sonoda, Janmajay Singh, and Tomoko Ohkuma. 2020. Submodular bandit problem under multiple constraints. In Conference on Uncertainty in Artificial Intelligence. PMLR, 191–200.
> > > >
> > > > [3]. Xiaoying Zhang, Hong Xie, Hang Li, and John C.S. Lui. Conversational contextual bandit: Algorithm and application. In Proceedings of the web conference 2020, pp. 662–672, 2020.

---

> > > > ### Author Response · Authors · 2024-12-02
> > > >
> > > > Dear Reviewer RM4A,
> > > >
> > > > As the discussion phase is drawing to a close, we kindly ask whether our responses have resolved your concerns or if there are any remaining issues that we can further clarify. Your insights are invaluable in refining our work, and we are eager to ensure that all your concerns are fully addressed.
> > > >
> > > > Thank you once again for your time and effort in reviewing our manuscript.
> > > >
> > > > Best regards,\
> > > > Authors

---

> ### Author Response · Authors · 2024-11-25
> **Response to Remaining Concerns**
>
> Thank you for your detailed questions and for increasing the score of our work. Below, we address each of your remaining concerns.
>
> ---
> 1. **Regarding \(\alpha\)-Approximate Regret**
>
> Thank you for your insightful question. In combinatorial multi-armed bandits (CMAB), the introduction of $\alpha$ addresses a key challenge: selecting multiple arms is often **NP-hard**, even in the **offline setting** (where $\boldsymbol{\mu}$ is known). Comparing regret to a solution that assumes **unlimited time and resources** to solve this NP-hard problem is impractical and uninformative for real-world applications.
>
> To provide a meaningful benchmark, CMAB frameworks typically compare **online learning performance** against an offline solution that can be computed in **polynomial time** when $\boldsymbol{\mu}$ is known. This offline solution is often $\alpha$-approximate, meaning it achieves a theoretical performance ratio $\alpha$ relative to the true optimum. For many well-studied NP-hard problems, $\alpha$ is less than 1, reflecting the inherent difficulty of finding exact optimal solutions efficiently. Importantly, $\alpha = 0$ is typically unrealistic, as even suboptimal solutions are typically achievable when $\boldsymbol{\mu}$ is known.
>
> In our paper, the values of $\alpha$ for the different reward models—AWC, SUC, and AIC—are **1 - 1/e**, **1**, and **1**, respectively, as detailed in **Appendix B**. Notably, for the original **NP-hard integer problem**, we propose a **relaxation and rounding technique**, which transforms the problem into a **continuous** optimization problem, thereby improving upon previous works. Specifically, in those works [1, 2], the $\alpha$ values for similar reward functions were $ 1 - 1/K $, $ 1 - 1/K $, and $\frac{1}{4(1+\varepsilon)}, \varepsilon > 0 $, compared to our values of **1**, **1**, and **1 - 1/e**. We will highlight this improvement in the table in our subsequent response.
>
> ---
>
> 2. **Regarding the Parameters $\alpha_{\mu}$ and $\alpha_{c}$ for Exploration**:
>
> Thank you for your question. The parameters $\alpha_{\mu}$ and $\alpha_{c}$ control the **exploration** of reward and cost, respectively.
> In general, for online learning scenarios with high uncertainty, we recommend setting exploration parameters **higher** to encourage more exploration and **avoid local optima**. Conversely, in less uncertain settings, a **lower** value can suffice. In our work, due to the black-box nature of LLMs and their performance uncertainty, we set $\alpha_{\mu}$ higher than $\alpha_{c}$.
>
> In **Appendix E.2, Figure 7** (lines 1718-1727), we explore this topic further. Given the anticipated reduction in costs as LLMs develop, we focus on scenarios with generous budgets that prioritize performance. Our theoretical analysis suggests setting $\alpha_c$ higher in performance-driven contexts to ensure sufficient exploration for empirical improvement.
>
> We used the following configurations:
> - *Performance-driven1*: $(\alpha_{\mu}, \alpha_{c}) = (0.3, 1)$
> - *Performance-driven2*: $(\alpha_{\mu}, \alpha_{c}) = (1, 1)$
> - *Cost-driven1*: $(\alpha_{\mu}, \alpha_{c}) = (0.3, 0.01)$
> - *Cost-driven2*: $(\alpha_{\mu}, \alpha_{c}) = (1, 0.01)$
>
> The results in  Figure 7 indicate that **performance-driven settings** yield higher rewards, while **cost-driven settings** show fewer violations. The choice of approach depends on the specific **needs and constraints** of the task at hand.
>
> ---
>
> 3. **Regarding the Effect of Partial Feedback**:
>
> Thank you for your question. The **partial feedback setting** introduces **incompleteness in feedback observability**, as characterized by the minimum observation probability ($ p^* $) in our upper bound. This limited feedback increases the **uncertainty** in decision-making for LLM selection..
>
> In this context, standard techniques for CMAB with **semi-bandit feedback** are not directly applicable because we cannot observe feedback from all arms within an action in each round. To address this, we developed a novel approach to recharacterize and analyze the required **pull counts** under conditions of **incomplete feedback**. Additionally, we reconstructed the martingale to account for the partial observability, enabling us to derive a **tight upper bound**. For detailed derivations, please refer to **Appendix 1305–1378**.
>
>
> ---
> 4. **Regarding the Trade-Off Between Reward and Cost Constraint Violation**:
>
> Thank you for your question. In our **theoretical analysis**, the regret and constraint violation are **analyzed in parallel**. On the **experimental side**, the trade-off between reward and cost violation can be controlled using the parameters $\alpha_{\mu}$ and $\alpha_{c}$. For further details, please refer to our response to your second question.
>
> ---

---

### Official Review · Reviewer_WQ9G · 2024-11-07

**Soundness:** 3
**Presentation:** 3
**Contribution:** 3
**Rating:** 6
**Confidence:** 2

**Summary:**

Main problem statement: the problem statement is clear and well-motivated (i.e. the setup of selecting a subset of LLMs/experts to answer a given query, in a cost and reward-sensitive environment. I do think combinatorial bandits are a reasonable way to model this problem, although I do have some nits about it which I’ll mention below).
Idea - the technical/intellectual contribution of the paper isn’t inherently about LLMs, in my opinion.I think the main idea is a new combinatorial bandit algorithm (which they dub C2MAB-V), which attempts to do exploration/exploitation and bandit-feedback updates in tandem with solving the submodular optimization described in Eq.3. I certainly have not seen this algorithm in the literature before, but I should caveat by saying that I am not an expert on combinatorial bandits. (As an aside, I do see a paper of Karthik’s referenced here on comb. bandits, so it may be worth getting his gut-check?) In addition to the new algorithm, the authors also derive a regret bound.
Novelty - Both the C2MAB-V and its regret analysis (Theorem 1) are novel as far as I (a non-expert) know.
Evaluation - the empirical evaluations are interesting and suggest that C2MAB-V does empirically outperform other “naive” alternatives such as always going with the best/most expensive model. I am curious to know why the y-axis in Fig 4 is reward / budget violation. The presence of the budget violation term there is a bit puzzling to me. Any feasible solution of (3) should be budget feasible and so budget should never really be violated (i.e. the denominator should always be zero)?
Benchmark - In Fig 4., evaluation is done against a good set of benchmarks including eps-greedy, Thompson sampling, and combinatorial UCB
Practical deployability?
I have some “weak” concerns about the practical “deployability” of the ideas proposed here in the context of LLMs. Some alternative ways that cost and performance tradeoffs may potentially be navigated for practical large-scale systems involve avoiding the appeal to MAB altogether, and its unclear whether this paper can prescribe into any insights about the relative merits of leveraging MAB versus other approaches. One such approach is Mixture-of-Experts (MoE) where the experts are directly absorbed as “sub-models” inside a larger model and the task of identifying which expert to direct a task to is also captured inside the model and learnt in the pre-training phase. Another approach is to appeal to “Composition of Experts (https://arxiv.org/pdf/2405.07518)”, where given some expert models, a new model is trained to learn to route a given task to the best set of experts. Based on this paper, can we draw any conclusions vis-a-vis MoE vs CoE vs MAB-combination of different LLMs?
One other deployability challenge that I see (and I don’t think was adequately addressed in the paper) is the contextual nature of the problem. Given a task, it seems to me that the context for that task (e.g. the required expertise e.g. medicine vs biology vs math) is needed to meaningfully identify the potential reward and also cost, and we’d need the bandit feedback to be contextual. In the method described by the authors, it is unclear whether and how the context is captured. I do believe the regret analysis itsef would also change substantially based on whether the algorithm describes a non-contextual vs a contextual implementation (since uncertainty quantification is far more nuanced in the contextual setup)

**Strengths:**

I think the core explore-exploit algorithm that was introduced is new (to this reviewer), interesting, and also comes with potentially new regret guarantees.

**Weaknesses:**

I have some concerns about the practical deployability of the approach.

**Questions:**

No direct questions, but I welcome responses from the authors on the deployability and other aspects raised above.

---

> ### Author Response · Authors · 2024-11-19
> **Response to Reviewer WQ9G**
>
> Thank you for your insightful comments! Here are our responses:
>
> ---
>
> (Since your questions are mainly in the summary section, we will address this part first.)
> ## Questions in Summary
>
> 1. **Regarding the new challenges and techniques used in the algorithm and analysis:**
>
> Thank you for your thoughtful feedback. To our knowledge, we are the first to study the CMAB problem with **long-term constraints** for **general reward functions** and **partial feedback models**. Additionally, we propose a unified and computationally efficient solution using the relaxation and rounding (RR) approach that provides both regret and violation guarantees.
> **For more discussions about the new challenges and techniques, please refer to our general response**.
>
> ---
>
> 2. **Regarding Comparison with Combinatorial Bandits and Karthik’s Work:**
>
> Thank you for your feedback. We would like to clarify that in the initial version of our manuscript, we did present arison with works on **bandits with knapsack constraints**, including the paper [1] by Karthik referenced in Section 5. Additionally, we have now added a **formal review of relevant works on Bandits with Knapsack Constraints** in **Appendix A.2**, where we compare these works with our approach.
>
> Key points of comparison are as follows:
>
> a. *Resource Consumption and Budget Constraints*:
>
> Several works in the literature, such as **bandits with budgets** and **bandits with knapsacks (BwK)**, consider **resource consumption and budget constraints** in stochastic MAB/CMAB settings. These works typically focus on **optimal stopping times**, where the learning process terminates when resources are depleted. In contrast, our model addresses a **long-term constraint**, allowing the arm selection process to continue for an arbitrary number of rounds, without requiring an early termination when resources are used up.
>
> b. *Comparison with BwK Studies*:
>
> Among the BwK studies, the work on **Combinatorial Semi-Bandits with Knapsacks** is closest to our approach. Both frameworks utilize a combination of UCB/LCB, linear programming, and rounding procedures. However, their approach is restricted to the **linear CMAB** setting, where rewards are linear and feedback is semi-bandit. In contrast, our framework generalizes this approach to accommodate **non-linear reward functions** by applying different **relaxation** and **rounding techniques**, allowing for a broader set of applications.
>
> c. *Key Difference in Proof Technique*:
>
>  A significant difference between our work and previous BwK studies is that our proof technique does not rely on the **negative correlation property** of randomized rounding. This enables greater flexibility in choosing **relaxation functions** and **rounding procedures**, making our framework applicable to a wider class of **non-linear reward functions**, which is a key distinction in our approach.
>
>
> [1]. Sankararaman K A, Slivkins A. Combinatorial semi-bandits with knapsacks[C]//International Conference on Artificial Intelligence and Statistics. PMLR, 2018: 1760-1770.
>
> ---
>
> 3. **Regarding the reward/violation ratio:**
>
> Thank you for your comment. The **reward/violation ratio** is used to measure the **reward-violation tradeoff** and the overall performance of the algorithm. It is defined as the **per-round reward** divided by the **per-round violation**, which allows us to understand how much reward the algorithm gains for each unit of budget violation.
>
> Although the budget constraint is **not always violated**, as discussed in **Theorem 2**, the **violation** is not always zero, where **violation decays at a rate of $\tilde{O}\left(\sqrt{\frac{K}{T}}\right) $**. This is because the cost distribution also needs to be learned online, and there may be some **discrepancies between the learning process and the real-world cost structure**. While we expect the violation to remain small, there can occasionally be slight budget overruns. Importantly, **significant violations are not permitted**, so this metric ensures that we are capturing any small breaches of the budget constraint, which is commonly used in related works, such as [1, 2, 3].
>
> Additionally, as shown in **Appendix E.2 (Figures 6 and 7)**, we provide further visualizations of the reward and violation results across the three task types with nine different LLMs.
>
> ---
>
> [1]. Bernasconi M, Cacciamani F, Castiglioni M, et al. Safe learning in tree-form sequential decision making: Handling hard and soft constraints[C]//International Conference on Machine Learning. PMLR. 2022: 1854-1873.
>
> [2]. Chen K, Cai K, Huang L, et al. Beyond the click-through rate: web link selection with multi-level feedback[C]//Proceedings of the 27th International Joint Conference on Artificial Intelligence. 2018: 3308-3314.
>
> [3]. Saxena V, Jalden J, Gonzalez J. Thompson sampling for linearly constrained bandits[C]//International Conference on Artificial Intelligence and Statistics. PMLR. 2020: 1999-2009.
>
> ---

---

> ### Author Response · Authors · 2024-11-19
>
> 4. **Regarding comparison of MAB, MoE, and CoE:**
>
> We understand the importance of considering alternative strategies like MoE and CoE, and we appreciate the opportunity to compare them in terms of their practical applicability.
>
> a. *Practical Deployment: MAB vs. MoE/CoE*
>
> The most suitable approach depends on the **specific application scenario** and **deployment requirements**:
>
> - MAB methods excel in **real-time feedback-driven model selection** to maximize performance while controlling costs, making them ideal for tasks with changing demands over time.
>
> - MoE is good at **inference efficiency** and **large-scale deployment** scenarios.
>
> - CoE is especially useful for enabling **multi-expert collaboration** in handling complex tasks, ensuring optimal utilization of expert capabilities.
>
> b.  *Complementarity of MAB, MoE, and CoE*:
>
> While MAB, MoE, and CoE are distinct strategies, they can also be **complementary**.  For example, one can combine **MAB and MoE** by using MAB to **dynamically select** among the expert models within an MoE framework.  This would be particularly useful when there are **shifts in task distribution** or when the system needs to optimize which expert to activate for each query. Additionally, MAB-based methods provide theoretical performance analysis, offering **rigorous explainability** of their decision-making process
>
> c.  *Empirical Evidence*:
>
>  In **Appendix E.3 (Figure 13)**, we provide an experiment where we **pre-learned a fixed combination of multi-LLMs offline** and applied it to online queries.  The results demonstrate that **online feedback-driven adjustments** significantly enhance the effectiveness of the pre-set LLM selection.  This experiment shows the crucial role of **continuous online adaptations** in optimizing LLM selection for real-time queries, indicating how MAB can complement offline pre-trained combinations in practical systems.
>
> ---
>
> 5. **Regarding the contextual bandit framework:**
>
> Thank you for your valuable feedback on the role of contextual information in multi-armed bandit methods. We agree that incorporating **context**—such as domain-specific expertise (e.g., medicine, biology, math)—can significantly enhance the algorithm’s ability to capture user preferences and improve both reward identification and cost estimation.
>
> For example, as illustrated in Theorem 1, the regret bound depends on $K$, the total number of available LLMs. In a contextual setup, such as for math queries, it might be more efficient to select an LLM that has been fine-tuned specifically for math tasks rather than choosing from the entire pool of LLMs. In this case, $K$ would be significantly reduced, narrowing the bandit’s decision space and making the selection process more efficient.
>
> While our current implementation focuses on a non-contextual setup, we believe that extending our framework to incorporate **contextual combinatorial multi-armed bandit** is a promising direction for future work. In **Section 7**, we highlight our intention to explore this extension in subsequent research.
>
> ---
>
> ## Weakness
>
>
> 1. **Regarding practical deployability:**
>
> Thank you for your concerns regarding the practical deployability of our approach. We designed our method with **practical deployment** in mind, particularly in the context of the **Local-Cloud Architecture** outlined in Section 3, and the division of tasks between the **local server** and **scheduling center** described in Section 4. This architecture is specifically intended to optimize the coordination between **local servers**, which have limited resources, and the **scheduling cloud**, which has more abundant resources.
>
> Additionally, we have explored real-world deployment scenarios in Appendix E.3, including experiments with more complex **asynchronous communication architectures** and research on transitioning from a traditional **two-tier** model to a more sophisticated **multi-tier LLM selection strategy**. These investigations demonstrate the scalability and adaptability of our framework in real-world deployments.  Furthermore, we are currently running experiments on other open-source datasets to showcase the robustness in practical deployment settings. **We will strive to include these results in a revised version of the manuscript** before the discussion phase ends.
>
> For more detailed answers on practical deployability, please refer to our responses to your **Questions 4 and 5** of **Questions in Summary**.

---

> ### Author Response · Authors · 2024-11-25
>
> Dear Reviewer WQ9G,
>
> As the discussion phase is drawing to a close, we kindly ask whether our responses have resolved your concerns or if there are any remaining issues that we can further clarify. Your insights are invaluable in refining our work, and we are eager to ensure that all your concerns are fully addressed.
>
> Thank you once again for your time and effort in reviewing our manuscript.
>
> Best regards,\
> Authors

---

### Author Response · Authors · 2024-11-19
**General Response**

We sincerely thank all the reviewers for their thoughtful feedback and valuable comments on our paper. We have addressed the suggestions by incorporating revisions, detailed in the **revised manuscript PDF**. Newly added sections and modifications are highlighted in **blue** for easy reference. Additionally, to clarify common concerns regarding the novel challenges and techniques in our combinatorial multi-armed bandit approach, we would like to provide the following statement.

---

**Regarding  the new challenges and techniques used in the algorithm and analysis**:

We acknowledge that the problem we study falls within the broader context of combinatorial online optimization, but it presents several novel aspects that distinguish it from existing work. To our knowledge, we are the first to study the CMAB problem with **long-term constraints** for **general reward functions** and **partial feedback models**. Additionally, we propose a **unified and computationally efficient solution** using the **relaxation and rounding (RR) approach** that provides both **regret** and **violation guarantees**.

The key innovation of our approach lies in its formulation of **constraint-effective combinatorial multi-armed bandits**, a new variant of combinatorial semi-bandits. While previous CMAB research has made significant progress in minimizing regret, many existing approaches overlook the critical aspect of **cost constraints**, often allowing for excessive violations. Our work directly addresses this gap by adopting a **constraint-effective** perspective, where we aim to minimize both **regret** and **constraint violation** simultaneously.

*Solving Constrained Optimization with Relaxation and Rounding*:

A distinguishing feature of our work is the way we tackle the constrained optimization problem. Traditional methods typically treat the constraint as a **discrete** optimization problem, which is often NP-hard and computationally inefficient. Direct solutions to this problem tend to yield suboptimal approximation ratios. In contrast, we propose a **relaxation and rounding (RR)** technique, which transforms the problem into a **continuous** optimization problem. This transformation not only improves approximation guarantees but also offers better time complexity compared to traditional methods.

*Tackling Online Challenges with RR Techniques*:

However, implementing RR techniques in an **online setting** presents new challenges, as RR methods were originally designed for **offline** optimization problems. The flexibility in choosing relaxation functions and rounding procedures complicates the guarantee of low regret and low violation in the online case. To overcome this challenge, we propose a detailed analysis that identifies the conditions under which the RR approach can effectively perform in online scenarios. This involves establishing a connection between regret and over-estimation terms, i.e., $ \mathbb{E}\left[\tilde{r}\left(\tilde{\boldsymbol{Z}}\_t, \bar{\boldsymbol{\mu}}\_t\right)-\mathbb{E}\left[\tilde{r}\left(\mathbb{I}\_{S\_t}, \bar{\boldsymbol{\mu}}\_t\right)\right]\right]+\mathbb{E}\left[r\left(S\_t, \bar{\boldsymbol{\mu}}\_t\right)-r\left(S\_t, \boldsymbol{\mu}\right)\right]$, where only the second term can be bounded using standard CMAB  analysis.

*Addressing Complex Reward Models in Multi-LLM Tasks*:

The complexity of multi-LLM tasks introduces additional challenges due to non-linear reward models. Specifically:  1) **Non-linear reward** cannot be relaxed using a simple concave function. Instead, we model it as a **submodular function** and use its **multi-linear extension**; 2) Ensuring the quality of an action in expectation requires us to introduce a **randomized swap rounding procedure** tailored for non-linear rewards. This ensures that the rounded solution maintains the desired **convexity properties**, i.e., $\mathbb{E}\left[\tilde{r}\left(\mathbb{I}\_{S\_t}, \boldsymbol{\mu}\right)\right] \geq \tilde{r}(\tilde{\boldsymbol{Z}}\_t, \boldsymbol{\mu})$.

*Ensuring Long-term Constraint Satisfaction*:

Additionally, we use a novel **martingale construction** technique to bound long-term violations, which is essential for addressing constraints in online combinatorial optimization.

We have added a detailed discussion of the challenges and techniques used in the algorithm and analysis in **Appendix A.3**. Additionally, we provide a unified **instance-dependent** performance bound for diverse reward models in **Appendix D.4**, addressing the key limitation where **expectation-based analysis** overlooks crucial **action-dependent terms** needed for instance-dependent guarantees.

---

> ### Author Response · Authors · 2024-11-19
>
> A summary of related multi-armed bandit (MAB) works is presented below.
>
> | **Algorithm**                          | **Combinatorial Arms?** | **Non-linear Reward?** | **Unknown Stochastic Cost?**       | **Cost Constraint**  | **Partial Feedback?\***         |
> |----------------------------------------|--------------------------|-------------------------|-------------------------------------|-----------------------|-----------------------------------|
> | POND, [1] | ✗                        | ✗                       | ✓                           | Long term             | NA                                |
> | OptPess-LP, [2] | ✗                        | ✗                       | ✓                      | Long term             | NA                                |
> | Pessimistic-Optimistic, [3] | ✗                        | ✗                       | ✓                       | Long term             | NA                                |
> | BwK\† and its variants, [4, 5] | ✗                        | ✗                       | ✓                           | BwK†                 | NA                                |
> | CUCB-CRA, [6] | ✓                        | ✓                       | ✗                               | NA                    | ✗                                |
> | CUCB-T and its variants, [7, 8] | ✓                        | ✓                       | ✗                               | NA                    | ✓                                |
> | SemiBwK-RRS, [9] | ✓                        | ✗                       | ✓                          | BwK†                 | ✗                                |
> | Constrained UCB, [10] | ✓                        | ✗                       | ✗                         | Long term             | ✗                                |
> | AFSM-UCB, [11] | ✓                        | ✓                       | ✗                         | Per round‡           | ✗                                |
> | **C2MAB-V, Ours**                      | ✓                        | ✓                       | ✓           | Long term             | ✓                                |
>
> \* Partial feedback can cover applications with partially observed arms, e.g., cascading bandits.
> † BwK means the bandit with knapsacks.
> ‡ Per round means the action satisfies the cost constraint in each round.
>
> ---
>
> [1]. Xin Liu, Bin Li, Pengyi Shi, and Lei Ying. 2020. POND: Pessimistic-Optimistic Online Dispatching. arXiv preprint arXiv:2010.09995 (2020).
>
> [2]. Tao Liu, Ruida Zhou, Dileep Kalathil, Panganamala Kumar, and Chao Tian. 2021. Learning policies with zero or bounded constraint violation for constrained mdps. Advances in Neural Information Processing Systems 34 (2021), 17183–17193.
>
> [3].  Xin Liu, Bin Li, Pengyi Shi, and Lei Ying. 2021. An efficient pessimistic-optimistic algorithm for stochastic linear bandits with general constraints. Advances in Neural Information Processing Systems 34 (2021), 24075–24086.
>
> [4]. Shipra Agrawal and Nikhil R Devanur. 2019. Bandits with global convex constraints and objective. Operations Research 67, 5 (2019), 1486–1502.
>
> [5]. Ashwinkumar Badanidiyuru, Robert Kleinberg, and Aleksandrs Slivkins. 2013. Bandits with knapsacks. In 2013 IEEE 54th Annual Symposium on Foundations of Computer Science. IEEE, 207–216.
>
> [6]. Jinhang Zuo and Carlee Joe-Wong. 2021. Combinatorial multi-armed bandits for resource allocation. In 2021 55th Annual Conference on Information Sciences and Systems (CISS). IEEE, 1–4.
>
> [7]. Wei Chen, Yajun Wang, Yang Yuan, and Qinshi Wang. 2016. Combinatorial multi-armed bandit and its extension to probabilistically triggered arms. The Journal of Machine Learning Research 17, 1 (2016), 1746–1778.
>
> [8]. Shuai Li, Baoxiang Wang, Shengyu Zhang, and Wei Chen. 2016. Contextual combinatorial cascading bandits. In International conference on machine learning. PMLR, 1245–1253.
>
> [9].  Karthik Abinav Sankararaman and Aleksandrs Slivkins. 2018. Combinatorial semi-bandits with knapsacks. In International Conference on Artificial Intelligence and Statistics. PMLR, 1760–1770.
>
> [10]. Kun Chen, Kechao Cai, Longbo Huang, and John CS Lui. 2018. Beyond the click-through rate: web link selection with multi-level feedback. In Proceedings of the 27th International Joint Conference on Artificial Intelligence. 3308–3314.
>
> [11]. Sho Takemori, Masahiro Sato, Takashi Sonoda, Janmajay Singh, and Tomoko Ohkuma. 2020. Submodular bandit problem under multiple constraints. In Conference on Uncertainty in Artificial Intelligence. PMLR, 191–200.

---

### Author Response · Authors · 2024-11-24
**Expanded Evaluation Across Diverse Datasets**

Dear Reviewers,


We have extended our empirical evaluation to include multiple publicly available datasets and the LLaMA 3 model, aiming to validate the robustness of our approach across diverse deployment settings. The latest experimental results are detailed in **Appendix E.4**.

For your convenience, all changes have been highlighted in blue. We hope these revisions effectively address your comments and enhance the manuscript.

Please do not hesitate to let us know if further clarification or additional adjustments are required.

Thank you once again for your valuable and constructive feedback.

Best regards,
The Authors

---

### Meta-Review · Area_Chair_NDnj · 2024-12-31

**Metareview:**

The paper proposes a new algorithm for selecting a subset of large language models (LLMs) to answer a given query, considering both cost and reward. The algorithm, called C2MAB-V, is based on combinatorial bandits and attempts to balance exploration and exploitation while solving a submodular optimization problem. The paper also derives a regret bound for the algorithm.


The problem statement is clear and well-motivated, and the use of combinatorial bandits is a natural approach to address the trade-off between cost and reward (WQ9G, hptZ) in this setup. The paper provides a regret bound for the algorithm (WQ9G, hptZ), which involves a number of novel moving parts. The empirical evaluations are interesting and suggest that C2MAB-V outperforms other naive alternatives (WQ9G, hptZ).

However, while the application to LLMs is timely, reviewers found the algorithm to be not substantially different from existing methods, and the analysis is similar to previous results on combinatorial bandits (RM4A, XJC9).   RM4A points out that reward models used in the paper are simple and may not accurately reflect real-world scenarios. The paper only evaluates the algorithm on one dataset and does not provide enough information on the choice of parameters (RM4A). Moreover, the paper does not address the practical deployability of the algorithm, particularly in the context of LLMs (WQ9G, XJC9), and does not provide a clear comparison with other approaches, such as Mixture-of-Experts (MoE) and Composition of Experts (CoE) (WQ9G). One reviewer found the paper to be dense and hard to follow (XJC9), perhaps because this is more of a bandit paper.

Given the weaknesses and the lack of clarity around the contribution,  I would recommend rejecting this paper.  The authors should address whether this is solving a generalizable bandit problem that is a durable contribution to the literature, in which case framing the contribution as such (with an appropriate title, and cases beyond the LLM routing problem) would be appropriate.  If it is simply solving a specific problem with deployment of LLMs, more practical evaluation must be done (and perhaps breaking up the work into two papers may be logical).

**Additional Comments On Reviewer Discussion:**

The authors provided extensive arguments clarifying the contributions of the work.  However, this did not sway the reviewer's conclusion that this is essentially a bandit paper and the technical contribution isn't great enough to warrant publication.  This, plus the lack of connection between the theoretical work and representation learning, led me to conclude that this work is better suited to a community that can critically evaluate and appreciate this type of contributon.

---

### Decision · Program_Chairs · 2025-01-22

Reject